# Efficient and Near-Optimal Smoothed Online Learning for Generalized Linear Functions

**Adam Block**
Department of Mathematics
MIT
Cambridge, MA 02139
ablock@mit.edu

**Max Simchowitz**
CSAIL
MIT
Cambridge, MA 02139
msimchow@csail.mit.edu

## Abstract

Due to the drastic gap in complexity between sequential and batch statistical learning, recent work has studied a smoothed sequential learning setting, where Nature is constrained to select contexts with density bounded by $1/\sigma$ with respect to a known measure $\mu$. Unfortunately, for some function classes, there is an exponential gap between the statistically optimal regret and that which can be achieved efficiently. In this paper, we give a computationally efficient algorithm that is the first to enjoy the statistically optimal $\log(T/\sigma)$ regret for realizable $K$-wise linear classification. We extend our results to settings where the true classifier is linear in an over-parameterized polynomial featurization of the contexts, as well as to a realizable piecewise-regression setting assuming access to an appropriate ERM oracle. Somewhat surprisingly, standard disagreement-based analyses are insufficient to achieve regret logarithmic in $1/\sigma$. Instead, we develop a novel characterization of the geometry of the disagreement region induced by generalized linear classifiers. Along the way, we develop numerous technical tools of independent interest, including a general anti-concentration bound for the determinant of certain matrix averages.

## 1 Introduction

In batch statistical learning, a learner faces a set of independent examples drawn from a given distribution, and is tasked with generalizing to novel examples drawn from that same distribution. In sequential or *online* learning, however, Nature may adversarially select examples to thwart the learner's progress and success is defined only in comparison to the best a priori predictor. Due to the wide range of application and minimal set of assumptions, online learning has received considerable recent attention. For concreteness, consider binary classification, where a sequence of $T$ examples takes the form $(x_t, y_t) \in \mathcal{R}^d \times \{-1, +1\}$. Even in the *realizable setting*, where there exists a true $f^\star$ in a pre-specified class of functions $\mathcal{F}$ for which $f^\star(x_t) = y_t$ for all $t \in \{1, 2, \ldots, T\}$, the gap between batch and statistical learning and sequential learning can be drastic: when $d = 1$, the class of linear thresholds $f_\theta(x) = \text{sign}(x - \theta)$ has VC dimension one and is thus learnable in the PAC framework [Wainwright, 2019]. A sequential adversary, however, can select $x_t$ so as to force the linear to misclassify $\Omega(T)$ points [Littlestone, 1988].

To circumvent the pessimism of the sequential setting, recent works [Rakhlin et al., 2011, Haghtalab et al., 2020, 2021, Block et al., 2022, Haghtalab et al., 2022] have studied the *smoothed sequential learning* paradigm, where the adversary is constrained to choose $x_t$ at random from any probability distribution $p_t$ with density at most $1/\sigma$ with respect to a known measure $\mu$. The most current of these results point to a striking statistical computational gap: whereas there exist algorithms which attain regret that scales with $\sqrt{T \log(/\sigma)}$, computationally efficient algorithms can only

36th Conference on Neural Information Processing Systems (NeurIPS 2022).

hope for $\text{poly}(T/\sigma)$ regret in general, even against a realizable adversary [Haghtalab et al., 2022, Theorem 5.2]. In many $d$-dimensional settings, natural choices of $\mu$ yield $\sigma = \exp(-\Omega(d))$, and thus the exponential separation in $\sigma$ translates into an exponential separation in dimension. This gap motivates the following question: can the statistical-computational gap be eliminated in more structured settings? In this work, we answer the question affirmatively for a variety of natural function classes. A better understanding of what function classes allow computationally efficient, statistically optimal regret-minimizing algorithms remains a promising direction for future research.

**Contributions.** We show that for certain classes of realizable smoothed online classification problems, there exists a *computationally efficient* algorithm which enjoys the statistically optimal $\log(T/\sigma)$ regret scaling, when the base measure $\mu$ is uniform on the unit-ball. Specifically, we provide computationally efficient algoirthms for achieving the statistically optimal regret bound for the following function classes:

- For affine thresholds,
- For affine thresholds in nonlinear features,
- For $K$-class affine classification,
- For piecewise affine regression

We also provide lower bounds that demonstrate the statistical optimality of our algorithms. Furthermore, we apply our results to noiseless contextual bandits and get a fast algorithm that achieves optimal regret dependence on the horizon, up to logarithmic factors. Finally, we present a complementary approach based on the perceptron algorithm which is robust to adversarial corruptions of the labels $y_t$, and enjoys a polynomial regret in a "directional smoothness" parameter which interpolates between the $\log(1/\sigma)$-guarantees attained above in the realizable setting, and the $\text{poly}(1/\sigma)$ bounds from prior work. We emphasize that, though we adopt the smoothed online learning setting of Rakhlin et al. [2011], Haghtalab et al. [2021], Block et al. [2022], we use entirely different techniques involving Ville's inequality [Ville, 1939], geometric measure theory, and convex geometry. Moreover, in none of these works was the question of adapting to realizability explored; thus, we provide the first regret bounds that are *logarithmic in both the horizon* and the smoothness parameter. We now discuss some related work:

**Online Learning.** Extensions of classical learning theory to the online setting have proliferated due to the scope of application. Several works [Littlestone, 1988, Blumer et al., 1989, Ben-David et al., 2009, Rakhlin et al., 2015a] have explored the gap in statistical rates between classical and online learning settings, with Littlestone [1988], Blumer et al. [1989] showing that the class of one dimensional thresholds, which is easy to learn in the batch setting, is not learnable with adversarial data. Other works, such as Rakhlin et al. [2015a], Rakhlin and Sridharan [2013], Rakhlin et al. [2015b], Block et al. [2021], Rakhlin and Sridharan [2014] have provided sequential analogues of classical notions of complexity that characterize minimax regret, as well as providing computational separation between classical and online learning [Hazan and Koren, 2016]. Due to the statistical and computational hardness results presented in the aforementioned work, there has been great interest in finding realistic, robust assumptions, such as smoothness, that allow for efficient learning.

**Smoothed Online Learning.** Smoothed analysis was first proposed in Spielman and Teng [2004] as a way to explain the success of the simplex algorithm of Klee and Minty [1972] by combining the polynomial time bounds of an average-case analysis with the verisimilitude of a worst-case analysis. Since then, smoothed analysis has been applied to explain the empirical success of many algorithms [Roughgarden, 2021]. In the learning setting, Rakhlin et al. [2011] proposed smoothed adversaries and proved regret bounds for linear thresholds in $\mathbb{R}^d$; their proof, however, was nonconstructive and did not achieve logarithmic regret in the realizable setting. The use of smoothed adversaries was essential due to the hardness results discussed above. In a series of works Haghtalab et al. [2020, 2021] generalized Rakhlin et al. [2011] and showed the regret depending on the VC dimension was possible in the smoothed online learning setting, albeit with computationally *inefficient* algorithms.

Recently, Block et al. [2022], Haghtalab et al. [2022] generalized Haghtalab et al. [2021] to allow for continuous labels and, more importantly, provided *oracle-efficient* algorithms for achieving vanishing regret in the smoothed setting. These papers also showed that the dependence on $\sigma$ in the regret bounds of their oracle-efficient algorithms, which was polynomial, could not in general be

reduced to the logarithmic dependence achievable by the inefficient algorithms, thereby exposing a statistical-computational gap. Unlike other recent works such as Block et al. [2022], Haghtalab et al. [2022], we do not use the coupling approach [Haghtalab et al., 2021] to prove our regret bounds.

**Classification with Linear Thresholds.**    Considering the ubiquity of linear thresholds in classification, the list of relevant references is far too long to include here; as such, we highlight only those most germane to our work. The perceptron algorithm was introduced in Rosenblatt [1958] and a margin-based mistake bound was proved in Novikoff [1963]. There have been many variations on and applications of this bound, from Ben-David et al. [2009] using it to bound the Littlestone dimension of linear thresholds with margin to dealing with non-realizable samples [Crammer et al., 2006, Freund and Schapire, 1999]. To the best of our knowledge our work constitutes the first to explore the effect that a smoothed adversary has on the perceptron algorithm.

**Disagreement Coefficient and Active Learning.**    Intuitively, our analysis is similar to works in active learning based on the disagreement coefficient [Hanneke, 2007, 2011, Hanneke et al., 2014, Wang, 2011]. Indeed, as we shall see, our regret bounds arise by bounding the probability that a point falls into the disagreement region in a similar way as, for example, Hanneke [2007] controls the label complexity of active learning. We will note in Remark 3, however, that an approach grounded purely in the disagreement coefficient cannot hope to achieve regret logarithmic in $\sigma$ in the smoothed setting. Indeed, our approach incorporates a finer understanding of the geometry, accomodated by the more limited scope of application of our techniques, which allows us to prove tight rates.

In Section 2, we setup the learning problem and introduce some necessary notions from convex geometry, as well as fixing notation. In Section 3, we highlight two technical results that form the foundation of our approach, before, as a warmup, applying them to the case of classification with linear thresholds in Section 4. In Section 5, we generalize beyond linear thresholds to allow for offset and nonlinear features. Finally, in Section 6, we move beyond binary classification by extending our results to $K$-class affine classification, piecewise affine regression, and noiseless contextual bandits.

## 2   Preliminaries

In this section, we provide basic definitions and setup the learning problem. We begin by defining a smooth distribution, as in Block et al. [2022], Haghtalab et al. [2021]:

**Definition 1.** *Let $\mu$ be a probability measure on a measurable space $\mathcal{X}$. For some $0 < \sigma \leq 1$, we say that a measure $p$ on $\mathcal{X}$ is $\sigma$-smooth with respect to $\mu$ if the likelihood $\frac{dp}{d\mu} \leq \frac{1}{\sigma}$ is uniformly bounded.*

We consider the smoothed online learning setting. First, a horizon $T \in \mathbb{N}$ is fixed and a distribution $\mu$ on $\mathcal{X}$ is chosen. For each step $1 \leq t \leq T$, Nature chooses a distribution $p_t$, possibly depending on the history, such that $p_t$ is $\sigma$-smooth with respect to $\mu$ and samples $x_t \sim p_t$ as well as choosing some $y_t \in \mathcal{Y}$. The learner sees $x_t$, chooses $\hat{y}_t$ and suffers loss $\ell(\hat{y}_t, y_t)$. Given a function class $\mathcal{F}$ of functions mapping $\mathcal{X} \to \mathcal{Y}$, the learner attempts to minimize regret, where regret is defined as:

$$\text{Reg}_T = \sum_{t=1}^{T} \ell(\hat{y}_t, y_t) - \inf_{f \in \mathcal{F}} \sum_{t=1}^{T} \ell(f(x_t), y_t). \tag{2.1}$$

In the sequel, for the sake of simplicity, we take $\mathcal{X} = \mathcal{B}_1^d$ to be the unit ball, $\mu = \mu_d$ to be the uniform measure on $\mathcal{B}_1^d$, and $\ell(\hat{y}_t, y_t) = \mathbb{I}(\hat{y}_t \neq y_t)$ to be the 0-1 loss.

**Remark 1** (Scaling of $\sigma$). *A natural example of a smoothed adversary is one that is allowed to place $\widehat{x}_t$ in a worst-case manner, which gets perturbed by some small additive noise, chosen uniform on $\varepsilon \cdot \mathcal{B}_1^d$, to become $x_t$; this adversary is $\sigma = \varepsilon^d$ smooth. For such situations, polynomial dependence on $\sigma$ in the regret translates into something exponential in dimension.*

**Remark 2** (Other measures $\mu$). *Assuming the dominating measure $\mu = \mu_d$ is not overly strong: if $\mu$ is another measure on $\mathcal{B}_1^d$ for which $\frac{d\mu}{d\mu_d} \geq c > 0$, then, because our regret bounds are logarithmic in $\sigma$, our results will still hold with an additive term of $\log\left(\frac{1}{c}\right)$.*

For much of the paper, we assume that Nature is *realizable with respect to $\mathcal{F}$*, i.e., for some $f^\star \in \mathcal{F}$, $f^\star(x_t) = y_t$ for all $1 \leq t \leq T$. In this case, $\text{Reg}_T$ is just a mistake bound: $\text{Reg}_T = \sum_{t=1}^{T} \mathbb{I}\{\hat{y}_t \neq y_t\}$.

The foundations of our analysis consider the class of linear threshold classifiers

$$\mathcal{F}_{\text{lin}}^d := \left\{ x \mapsto \text{sign}(\langle w, x \rangle) | w \in \mathcal{B}_1^d \right\}. \tag{2.2}$$

We identify $\mathcal{F}_{\text{lin}}^d$ with the set of $w$'s defining it, so that we may treat it as, itself, a subset of $\mathcal{B}_1^d$; other function classes are similarly identified with their parameters (without further comment).

At the core of our base algorithm is the computation of the *John ellipsoid* [John, 1948, Ball et al., 1997], the maximal volume ellipsoid contained in a convex body.[1] It is well-known that given a polytope in $\mathbb{R}^d$, the John ellipsoid can be computed in time polynomial in $d$ and the number of faces [Boyd and Vandenberghe, 2004]. In particular, we compute the John ellipsoid of the *version space*, $\mathcal{F}_t$, where for any time $t$, we let $\mathcal{F}_t = \{ f \in \mathcal{F}_{\text{lin}}^d | f(x_s) = y_s \text{ for all } s < t \}$, which is a polytope with $t \leq T$ faces. An important concept in our analysis is the notion of *Hausdorff measure*, which generalizes the standard notions of volume and surface area in $\mathbb{R}^d$; we will denote the $k$-dimensional Hausdorff measure (see Definition 18) by $\text{vol}_k(\cdot)$. More detail on both the John ellipsoid and the Hausdorff measure can be found in Appendix B.

**Notation.** For a set $\mathcal{U} \subset \mathbb{R}^d$, we denote by $\partial \mathcal{U}$ its boundary. We let $\mathcal{B}_r^d$ denote the ball of radius $r$ around the origin in $\mathbb{R}^d$ and let $S^{d-1} = \partial B_1^d$. Letting $\Gamma$ denote the $\Gamma$-function, let $\omega_d = \frac{\pi^{d/2}}{\Gamma(d/2+1)}$ denote the volume of $\mathcal{B}_1^d$ and let $\mu_d$ denote the uniform measure on $\mathcal{B}_1^d$ normalized to be a probability measure. If $\phi : \mathbb{R}^n \to \mathbb{R}^m$ is Lipschitz, we denote the Jacobian by $D\phi$. Lastly, we use "$\lesssim$" to denote inequality up to universal, problem-independent constants.

## 3 The Technical Workhorses

In this section we introduce the two key workhorse results that provide the technical foundation for the rest of the paper. The first result is a purely probabilistic statement that we use as a blackbox throughout the paper to turn probabilistic and geometric theorems into regret bounds in the realizable, smoothed online learning setting. The second result is a geometric statement that allows us to apply the black box regret bound to the case of classification with affine thresholds.

### 3.1 An Abstract Decay Analysis

We begin with an abstract, technical result that will form the basis for all of our regret bounds. We first introduce the following definition:

**Definition 2.** *Let $\mu$ be a measure on some set $\mathcal{Z}$ and let $\ell_t : \mathcal{Z} \to \{0,1\}$ be a sequence of loss functions. For $R > 0$ and $0 < c < 1$, we say that the sequence $(\ell_t, z_t)$ satisfies $(R,c)$-geometric decay with respect to $\mu$ if there exists a sequence of nonnegative numbers $R_t$ with $R_1 = R$ satisfying the following two properties:*

1. *For all $t$, $\mu \left( \{ z : \ell_t(z) = 1 \} \right) \leq R_t$.*

2. *For any $t$ such that $\ell_t(z_t) = 1$, we have $R_{t+1} \leq cR_t$.*

To motivate this admittedly abstract definition, consider the case of online classification with thresholds $f_\theta(x) = \text{sign}(x - \theta)$ from the introduction, with $\mu$ uniform on $[0,1] \times \{\pm 1\}$ (note that this does not precisely fit into the linear setting described above due to the offset); take $z_t = (x_t, y_t)$ and $\ell_t(z_t) = \mathbb{I}[\hat{y}_T \neq y_t]$, where the learner predicts $\hat{y}_t$ at each time $t$. By realizability, $\ell_t(z) = 1$ only when $x_t$ falls in the "region of disagreement," i.e. the interval the rightmost $x_s$ labelled $-1$ and the leftmost $x_s$ labelled 1. To see why this is true, note that the "version space," i.e., the set of thresholds that correctly classify all the data so far, is exactly this interval; for us to make a mistake, there must be two functions in the version space that disagree on $x_t$, which can only happen if $x_t$ itself is in the version space. If the learner denotes by $w_t$ the midpoint of the region of disagreement, then any mistake forces the version space, and thus the disagreement region, to shrink by a factor of 2. We see then that $(\ell_t, z_t)$ satisfy $\left(1, \frac{1}{2}\right)$-geometric decay with respect to the uniform measure.

If the adversary were constrained to choose $x_t \sim \mu$ at each time step, it is intuitive that we should not expect many mistakes to be made because, after any mistake, the probability that we make a mistake

---

[1]Some authors refer to the minimal volume ellipsoid *containing* a convex body as the John ellipsoid.

in some future interval decreases. In the following result, we show that this intuition holds in the more general smoothed setting:

**Lemma 3** (Abstract Decay Lemma). *Suppose that a sequence $(\ell_t, z_t)$ satisfies $(R, c)$-geometric decay with respect to some $\mu$ on $\mathcal{Z}$, and that for all $t$, there is some $p_t$ that is $\sigma$-smooth with respect to $\mu$ and $z_t \sim p_t$. Then for all $T \in \mathbb{N}$, with probability at least[2] $1 - \delta$,*

$$\sum_{t=1}^{T} \ell_t(z_t) \leq 4 \frac{\log\left(\frac{2TR}{\sigma\delta}\right)}{\log\left(\frac{1}{c}\right)} + \frac{e-1}{1-\sqrt{c}}. \tag{3.1}$$

*Proof Sketch.* We break our analysis into epochs whose lengths $h_m$ are tuned at the end of the proof. We then consider a sequence of stopping times $\tau_m$ that count the number of epochs of length $h_m$ we experience in between the $(m-1)^{st}$ and $m^{th}$ time that $\ell_t = 1$. We then show that if $h_m$ is not too large relative to the Probability that $\ell_t = 1$, then $\tau_m - \tau_{m-1}$ is large with high probability and apply Ville's inequality [Ville, 1939] to conclude that if $m_T$ is the maximal epoch-index $m$ such that $\tau_m \leq T$, then $m_T$ cannot be too large. We again apply Ville's inequality to show that if $h_m$ is not too large then the probability of multiple mistakes per epoch is small. Because of the geometric decay property, the probability that $\ell_t = 1$ decreases exponentially in the number of mistakes and thus we may let $h_m$ grow exponentially in $m$ and still not be too large to apply the above argument. We then conclude by noting that if $h_m$ are growing exponentially in $m$ then $m_T$ has to be logarithmic in $T$. The details can be found in Appendix C.1. ∎

If we return to the above example of online classification with thresholds, we see that Lemma 3 immediately yields the first regret bound for realizable, smoothed online learning with thresholds that is logarithmic both in the horizon $T$ and the smoothness parameter $\sigma$. The intuition gleaned from one-dimensional thresholds that geometric decay suffices to ensure logarithmic regret will be key to the more general regret bounds we exhibit below.

## 3.2 A Volumetric Lemma

In the previous section, we saw that in the setting of realizable, smoothed online classification with one-dimensional thresholds, the learner can force the indicator of a mistake at time $t$ to satisfy geometric decay; our second workhorse result will allow us to extend this fact to higher dimensions. In the case of thresholds in the unit interval, the key intuition leading to geometric decay was the fact that the disagreement region was exactly the version space and thus shrinking the version space tautologically shrank the disagreement region as well. In higher dimensions the situation is significantly more complicated. We have the following result:

**Lemma 4.** *Let $x_1, \ldots, x_t \in \mathcal{B}_1^d$ and suppose that $y_1, \ldots, y_t$ are realizable with respect to $\mathcal{F}_{\text{lin}}^d$. Define the disagreement region*

$$D_t := \left\{ x \in \mathcal{B}_1^d \mid \text{there exist } f, f' \in \mathcal{F}_t \text{ such that } f(x) \neq f'(x) \right\} \tag{3.2}$$

*where $\mathcal{F}_t$ is the version space, defined in Section 2. Then, recalling that $\partial\mathcal{F}_t$ is the boundary of $\mathcal{F}_t$,*

$$\mu_d(D_t) \leq 2 \cdot 4^{d-1} \mu_d(\mathcal{F}_t) + \frac{4^{d+1}}{\omega_d} \text{vol}_{d-1}(\partial\mathcal{F}_t). \tag{3.3}$$

Note that by controlling the size of $D_t$ by that of $\mathcal{F}_t$, Lemma 4 is a direct generalization of the one-dimensional case; however, in contradistinction to that setting, the proof is much more difficult and the bound includes an extra term corresponding to the surface area of $\mathcal{F}_t$, which is unavoidable in general. The full proof is in Appendix C.2, but we summarize the key points here. Though the conclusion of Lemma 4 is intuitive, it requires significant technical effort to prove.

*Proof Sketch of Lemma 4.* We first note that $D_t$ is contained in the set of points $x$ such that there is some $w \in \mathcal{F}_t$ with $\langle w, x \rangle = 0$; thus the conclusion of Lemma 4 reduces to a geometric statement about the volume of the set of points orthogonal to at least one point in a given set can be. It may seem like this should "obviously" be the volume of a $(d-1)$-dimensional ball multiplied by the

---

[2]Here, as in the rest of the paper, we made no effort to optimize constants. We include them only to demonstrate that they are not unreasonably large.

---

**Algorithm 1** Binary Classification with Linear Thresholds

---

1: **Initialize** $\mathcal{W}_1 = \mathcal{B}_1^d$, $w_1 = \mathbf{e_1}$
2: **for** $t = 1, 2, \ldots$ **do**
3:     **Recieve** $x_t$, and **predict** $\hat{y}_t = \text{sign}(\langle w_t, x_t \rangle)$,            (**% self.classify**$(x_t)$)
4:     **Update** $\mathcal{W}_{t+1} = \mathcal{W}_t \cap \{w \in \mathcal{B}_1^d | \langle w, x_t y_t \rangle \geq 0\}$
5:     **if** $\hat{y}_t \neq y_t$ **then**                                   (**% self.errorUpdate**$(x_t)$)
6:         $w_{t+1} \leftarrow$ **JohnEllpsoidCenter**$(\mathcal{W}_{t+1})$
7:         `% returns center of John Ellpsoid of given convex body`

---

volume of $\mathcal{F}_t$, but this is false: if $\mathcal{F}_t$ is the equator of the sphere $S^{d-1}$, then $\mu_d(\mathcal{F}_t) = 0$, but the set of points orthogonal to at least one point in $\mathcal{F}_t$ is the entirety of $\mathcal{B}_1^d$. Ruling out this pathology requires several steps, including a covering argument to reduce to the case where $\mathcal{F}_t$ is a ball, and application of (a generalized) Steiner's formula, and a deep geometric fact called Weyl's Tube Formula [Weyl, 1939, Gray, 2003] that governs how much volume we can add to $\mathcal{F}_t$ by "fattening" to include all points distance at most $\varepsilon$ from $\mathcal{F}_t$. ∎

## 4 Warmup with Linear Classification

In this section, we begin to apply our results from Section 3 to get tight regret bounds with computationally efficient algorithms for learning halfspaces in the realizable, smoothed online setting:

**Theorem 5.** *Let $\mu$ be the uniform measure on $\mathcal{B}_1$. Suppose that we are in the smoothed, realizable online learning setting, where the adversary samples $x_t$ from a distribution that is $\sigma$-smooth with respect to $\mu$. If we predict $\hat{y}_t$ according to Algorithm 1, then for all horizons $T$, with probability at least $1 - \delta$,*

$$\text{Reg}_T \leq 136 d \log(d) + 34 \log\left(\frac{T}{\sigma\delta}\right) + 56. \tag{4.1}$$

**Computational Efficiency.** The subroutine **JohnEllpsoidCenter**$(\mathcal{W}_{t+1})$ can be run in time polynomial in $T$ and $d$ by solving a Semi-definite Program (SDP) [Boyd and Vandenberghe, 2004, Primak and Kheyfets, 1995]. Note that we change our predictor $f_t$ only at the times $t$ that we make a mistake; thus, the number of calls to the SDP is also logarithmic in $T$.

*Proof Sketch of Theorem 5.* We apply Lemma 3 with $z_t = (x_t, y_t)$ and $\ell_t(z) = \mathbb{I}[\hat{y}_t \neq y_t]$. In order to do this we need to show that $\ell_t$ satisfies $(R, c)$ geometric decay, which amounts to finding a geometrically decreasing sequence of upper bounds on $\mu(D_t)$. By Lemma 4, it will suffice to provide such bounds on both $\mu(\mathcal{F}_t)$ and $\text{vol}_{d-1}(\partial \mathcal{F}_t)$, which is where the specific choice of $w_t$ becomes important. It is now classical [Tarasov et al., 1988, Khachiyan, 1990] that if a polytope is cut by a hyperplane through the center of its John ellipsoid then both halfs have John ellipsoids whose volumes are at most $\frac{8}{9}$ times the volume of the original ellipsoid; as we know that $\mathcal{F}_t \subset d \cdot \mathcal{E}_t$ [John, 1948], where $\mathcal{E}_t$ is the John ellipsoid of $\mathcal{F}_t$, we see that $\mu(d \cdot \mathcal{E}_t)$ is an upper bound on $\mu(\mathcal{F}_t)$ that decreases by $\frac{8}{9}$ every time we make a mistake. The true utility of the center of the John ellipsoid is that it also allows us to show that $\partial \mathcal{F}_t$ decreases by a constant factor. Indeed, we show that $\text{vol}_{d-1}(\partial \mathcal{F}_t) \leq \text{vol}_{d-1}(\partial \mathcal{E}_t)$ using a simple projection argument; we then apply a result of Rivin [2007] to bound the size of $\partial \mathcal{E}_t$ by $\mu(\mathcal{E}_t)$. The details are in Appendix D. ∎

**Importance of the John's Ellipsoid.** We show in Appendix D.3 that arbitrary predictions $y_t = \tilde{f}_t(x_t)$, for $\tilde{f}_t \in \mathcal{F}_t$ in the version space, can guarantee $1/\sigma$-regret at best. Hence, selecting the correct $w_t$ is key. One natural choice of $w_t$ is the Chebyshev center of $\mathcal{F}_t$ [Elzinga and Moore, 1975], equivalent to a max-margin estimator; unfortunately it need not decrease the volume sufficiently if $\mathcal{F}_t$ is too 'pointy.' Another choice, the centroid of $\mathcal{F}_t$, ensures decrease of the *polytope*'s volume, but is #P-hard to compute [Rademacher, 2007], and does not ensure decay of the surface area. The former problem can be accomodated with a sampling scheme [Bertsimas and Vempala, 2004], but the latter is critical. In contrast, the center of the John ellipsoid controls the decay of both $\mathcal{F}_t$ and its boundary. To gain intuition as to why the decay in surface area of $\mathcal{F}_t$ is necessary, consider the case where $\mathcal{F}_t$ is simply an arc in $S^{d-1}$. In this case, as $D_t$ is the set of points orthogonal to at least one point in in $\mathcal{F}_t$, it follows that $D_t$ has positive measure even though $\mathcal{F}_t$, being a lower dimensional set, does not; thus,

it is impossible in general to get a guarantee on the size of $D_t$ only in terms of the volume of $\mathcal{F}_t$, without regard to the surface area. In this way, we see that the choice of $w_t$ as the center of the John's Ellipsoid is critical to the success of our algorithm.

**Remark 3** (*Disagreement Coefficient*). Our analysis is similar in spirit to the disagreement-coeffcient analysis of active learning [Hanneke, 2007], which also exhibits geometric decay of the disagreement region $D_t$. The key difference is that the latter applies to *any algorithm* that selects a classifier from the version space $\mathcal{F}_t$ at each time $t$. Again, as shown in Appendix D.3, no such analysis can recover a better than $1/\sigma$-regret bound. The culprit is that disagreement-coefficient arguments ensure that $D_t$ shrink only *probabilistically* under samples $x_t \sim \mu$, and this probability may shrink by a factor of $\sigma$ in the smoothed-online setting. In contrast, our choice of classifier as the center of the John's ellipsoid ensures a *deterministic* decay of the disagreement region whenever a mistake is made.

**Lower Bound.** Before we move on to the more complicated settings, we note that this regret bound is tight up to a logarithmic factor in $d$. A proof of the following proposition, based on Ville's inequality, can be found in Appendix D.

**Proposition 6.** *Suppose that we are in the situation of Theorem 5. Then there is a realizable adversary such that any classifier experiences*

$$\mathbb{E}\left[\mathrm{Reg}_T\right] \geq \Omega\left(d + \log\left(T/\sigma\right)\right). \tag{4.2}$$

### 4.1 Smoothed classification via the Perceptron algorithm

Next, we present a guarantee for the classical Perceptron algorithm Rosenblatt [1958], which requires a much weaker notion of smoothness. We say that the adversary satisfies $\sigma_{\mathrm{dir}}$ directional smoothness if, for any fixed $w \in S^{d-1}$, it holds that for all $t$, $\langle x_t, w \rangle$ is $\sigma_{\mathrm{dir}}$-smooth with respect to the Lebesgue measure on the real line. As we explain in Example 2 in Appendix G, the directional smoothness $\sigma_{\mathrm{dir}}$ can be nontrivial even when the smoothness parameter $\sigma = 0$. We now show that the perceptron satisfies the following mistake bound under directional smoothness.

**Theorem 7.** *Fix any $w^\star \in S^{d-1}$ and $b^\star \in \mathbb{R}$. And suppose that the adversary satisfies $\sigma_{\mathrm{dir}}$-directional smoothness. Then, with probability $1 - \delta$, the online Perceptron (Algorithm 9 in Appendix G) satisfies*

$$\mathrm{Reg}_T = \sum_{t=1}^{T} \mathbb{I}\{\hat{y}_t \neq y_t\} \lesssim (T/\sigma_{\mathrm{dir}})^{\frac{2}{3}} \cdot (N_{\mathrm{err}}(w^\star, b^\star))^{\frac{1}{3}} + \log(\lceil \log T \rceil / \delta),$$

*where $N_{\mathrm{err}}(w^\star, b^\star) = 1 + \sum_{i=1}^{T} \mathbb{I}\{y_t \neq \mathrm{sign}(b^\star + \langle w^\star, x_t \rangle)\}$ controls deviation from realizability.*

For simplicity, Theorem 7 is stated relative to a *fixed* $w^\star \in S^{d-1}$ and $b^\star \in \mathbb{R}$; *uniform* bounds can be derived via a covering argument, at the expense of an additive $d \log(T/\delta\sigma_{\mathrm{dir}})$ term in the error bound. Unlike other algorithms proposed in this paper, Theorem 7 accomodates possibly non-realizable adversaries. It is also slightly more computationally expedient, not requiring the computation of the center of a John's ellipsoid. In contrast, its bound is polynomial in $T$ and $1/\sigma_{\mathrm{dir}}$, rather than logarithmic in $T$ and $1/\sigma$. There are situations where Algorithm 1 performs exponentially better than the Perceptron approach: suppose $x_t$ is uniform on an $\varepsilon$-ball whose center is chosen by the adversary. Then we have $\sigma = \varepsilon^{-d}$ and so Theorem 5 implies that the John ellipsoid approach gives regret that scales as $O(d \log(d/\varepsilon) + \log(T))$, whereas $\sigma_{\mathrm{dir}} \approx 1/\varepsilon$ and so Theorem 7 only ensures regret that is polynomial in $\varepsilon$. For further comparison, consult Remark 4 in Appendix G.

## 5 Beyond the Linear Case

While the results of Section 4 are technically interesting and have broad applications, they are limited to the specific linear setting. In this section, we show how our results can be extended, first to the more general affine setting, where the decision boundaries do not have to go through the origin, and then to a more general regime where we do not require linear decision boundaries.

### 5.1 Affine Classification

Our first generalization of Theorem 5 is to the setting where we allow our decision boundaries to be offset. Thus instead of assuming realizability with respect to $\mathcal{F}_{\mathrm{lin}}^d$, we will assume that the adversary

isrealizable with respect to

$$\mathcal{F}_{\text{aff}}^d = \left\{ x \mapsto \text{sign}(\langle w, x \rangle + b) | w \in \mathcal{B}_1^d \text{ and } b \in \mathbb{R} \right\}. \tag{5.1}$$

We have the following result:

**Corollary 8.** *Let $\mu$ be the uniform measure on $\mathcal{B}_1^d$ and suppose that we are in the smoothed online learning setting, where the adversary samples $x_t$ from a distribution that is $\sigma$-smooth with respect to $\mu$. Suppose that the adversary is realizable with respect to the function class $\mathcal{F}_{\text{aff}}^d$ defined in (5.1). Then Algorithm 3 in Appendix E.1 is a computationally efficient algorithm for choosing $f_t \in \mathcal{F}_{\text{aff}}^d$ such that for all $T$, with probability at least $1 - \delta$, it holds that*

$$\text{Reg}_T \leq 268 d \log(d) + 34 \log\left(T/(\sigma\delta)\right) + 56.$$

As $\mathcal{F}_{\text{lin}}^d \subset \mathcal{F}_{\text{aff}}^d$, the lower bound of Proposition 6 holds and Corollary 8 is tight up to a factor logarithmic in dimension. The proof is given in Appendix E.1 and proceeds by reducing to the linear setting of Theorem 5 by imbedding the problem into an online learning problem with contexts $\tilde{x}_t \in \mathbb{R}^{d+1}$, carefully randomized so as to preserve their smoothness with respect to $\mu_{d+1}$.

## 5.2 Linear Classification Under a Feature map

One limitation of the above discussion has been the assumption of linearity, which can be overly strong in many cases. In this section, we weaken this assumption in two ways. First, we show that if we transform the features with a well-behaved function, then we may still apply our above machinery. Second, we will show that our approach actually generalizes to polynomial decision boundaries through an elegant reduction. In both cases, the key technical challenge is to show that our transformed features remain smooth with respect to the uniform measure on a ball. Note that it is immediate that $\phi(x_t)$ is smooth with respect to $\phi_\star \mu_d$; in order to apply our results, however, we require smoothness with respect to the uniform measure. As it is not true that $\phi_\star \mu_d$ is smooth with respect to $\mu_d$ for general $\phi$, we require additional assumptions. We have the following result:

**Theorem 9.** *Let $\phi : \mathcal{B}_1^d \to \mathcal{B}_1^d$ be a function such that each coordinate function, $\phi_i : \mathbb{R} \to \mathbb{R}$ satisfies $\phi_i'(u) \geq \alpha$ for some $\alpha > 0$. If we run Algorithm 4 in Appendix E.2 then, for all $T$, with probability at least $1 - \delta$, it holds that*

$$\text{Reg}_T \leq 136 d \log\left(d/\alpha\right) + 34 \log\left(T/(\sigma\delta)\right) + 56.$$

Algorithm 4, the algorithm that achieves the above regret bound, is actually quite simple as it just runs Algorithm 1 on the data sequence $(\phi(x_t), y_t)$. A proof of a more general result, which applies to a larger class of maps $\phi$, is available in Appendix E.2. Even in the setting of Theorem 9, though, standard transformations like the sigmoid already apply.

We now turn to the more challenging case of polynomial features. We have the following result:

**Theorem 10.** *Let $\phi : \mathcal{B}_1^d \to \mathcal{B}_1^m$ be an L-Lipschitz function whose coordinates are polynomials of degree at most $\ell$ in the coordinates of $x \in \mathcal{B}_1^d$. Suppose that we are in the smoothed online learning setting where the $x_t$ are $\sigma$-smooth with respect to $\mu_d$ and the $y_t$ are realizable with respect to $\mathcal{F}_{\text{lin}}^m \circ \phi$. Suppose further that the Jacobian of $\phi$ satisfies for some $\alpha > 0$,*

$$\det\left(\mathbb{E}_{x \sim \mu_d}\left[D\phi(x)D\phi(x)^T\right]\right) \geq \alpha^2.$$

*Then Algorithm 5 in Appendix E.3 is a computationally efficient algorithm such that for all $T$, with probability at least $1 - \delta$,*

$$\text{Reg}_T \lesssim m \log(m) + \log\left(\frac{1}{\alpha}\right) + \ell^2 m^2 d \log^2\left(\frac{d\ell TL}{\sigma\delta}\right).$$

Algorithm 5 is a bit more complicated than simply applying Algorithm 1 to $(\phi(x_t), y_t)$ because if $d \leq m$, then $\phi(x_t)$ can never be smooth with respect to $\mu_m$ by dimension constraints. To escape this difficulty, we define a "meta-point," $\overline{x}_\tau$, which is the average of $\phi(x_t)$ for multiple different $t$. To understand why this might fix the problem, consider the identity imbedding of $S^{d-1} \subset \mathcal{B}_1^d$: if we sample $x$ uniformly on $S^{d-1}$, then the law of $x$ will not even be absolutely continuous with respect to $\mu_d$ but if we sample two points $x, x' \sim S^{d-1}$ then their average *is* absolutely continuous with respect to $\mu_d$. We note that the conditions on $\phi$ are fairly mild due to the logarithmic dependence on both the Lipschitz constant and the lower bound on the determinant, which is typically no less than exponentially small in $d$ and $m$.

*Proof Sketch of Theorem 10.* Algorithm 5 proceeds initially in a similar way to Algorithm 1: we maintain a version space $\mathcal{F}_t \subset \mathcal{F}_{\text{lin}}^m$ that gets updated every round and, when we change $w_t$, we set it to be the center of the John ellipsoid of the version space. In contradistinction to the earlier algorithm, however, we do not update $w_t$ every time we make a mistake. Instead, for some parameter $p$, we wait until we have misclassified a label $p$ times, i.e., we guessed $-1$ but $y_t$ was 1 $p$ times (or the reverse) and construct $\overline{x}_\tau$ to be the average of the $\phi(x_t)$ for each of these $p$ mistakes. Using a novel anti-concentration bound for determinants of certain random matrices (Proposition 41) as well as some techniques from geometric measure theory (Proposition 40), we show that $\overline{x}_\tau$ is smooth with respect to $\mu_m$. We then apply the abstract decay lemma (Lemma 3) in much the same way as we did in the proof of Theorem 5. The details are in Appendix E.3. ∎

# 6 Beyond Binary Classification

In the previous sections, we restricted our focus to binary classification; in this section we expand our scope to a $K$-class setting and then further extend to a regression setting. Our results for the regression setting, combined with the reduction of Foster and Rakhlin [2020], are applied to the setting of contextual bandits in Appendix A.

## 6.1 Multi-Class Classification

We first generalize our results to multi-class classification. The targets are $y_t \in [K]$ some fixed $K$ and classifications are assigned by maximum inner-product:[3]

$$\mathcal{F}_{K\text{-lin}}^d = \{x \mapsto f_{\mathbf{w}}(x) = \arg\max_{1 \leq i \leq K} \langle w^i, x \rangle \mid \mathbf{w} = (w^1, \dots, w^K) \in \left(\mathcal{B}_1^d\right)^K\}. \tag{6.1}$$

Our algorithm is a direct reduction to binary classification. For each $i < j$, we maintain an instance $\mathcal{A}_{\text{bin}}^{(i,j)}$ of Algorithm 1 which makes binary predictions $\hat{y}_t^{(i,j)}$ of $y_t^{(i,j)} = \text{sign}(\langle w_\star^i - w_\star^j, x_t \rangle)$. We then set our $K$-class prediction $\hat{y}_t$ as the first index $i$ for which $\hat{y}_t^{(i,j)} = 1$ for all $j > i$. The key insight is that, even though the learner does not recieve feedback on *all* $\hat{y}_t^{(i,j)}$ in this way, we can always assign a mistake $\hat{y}_t = y_t$ to an error $y_t^{(i,j)} \neq \hat{y}_t^{(i,j)}$ for *some* $i < j$. Formal pseudocode is given Algorithm 6 and a proof of the following regret bound is given Appendix F.1.

**Theorem 11.** *Suppose we are in the realizable, smoothed, online learning setting where the adversary is realizable with respect to the $\mathcal{F}_{K\text{-lin}}^d$ in (6.1). Then, then for all $T$, with probability at least $1 - \delta$, the regret of Algorithm 6 is at most*

$$\text{Reg}_T \leq 136K^2 d \log(d) + 91K^2 \log\left(TK^2/(\sigma\delta)\right). \tag{6.2}$$

The efficiency of the above algorithm follows from the efficiency of the binary classifers $\mathcal{A}_{\text{bin}}^{(i,j)}$. We conjecture that the dependence on $K^2$ is an artifact of our reduction to $\binom{K}{2}$ base classifiers.

## 6.2 Piecewise Regression

This section extends $K$-class classification to piecewise affine regression. We now suppose that the targets $y_t$ are real-valued, and realizable with respect to the following class of functions:

$$\mathcal{G}_{\mathcal{F}} = \left\{x \mapsto g_f(x) = \sum_{i=1}^K g_i(x)\mathbb{I}[f(x) = i] \middle| g_i(x) = \langle a_i, x \rangle \text{ for } a_i \in \mathbb{R}^d \text{ and } f \in \mathcal{F}_{K\text{-lin}}^d\right\}. \tag{6.3}$$

In contradistinction to the rest of the paper, where the adversary is allowed to play the $y_t$ adaptively subject only to the condition of realizability, in this section we suppose that the adversary is *semi-oblivious* in the sense that there is a ground-truth function chosen before the start of play and after learning begins, the adversary is only allowed to choose the contexts, $x_t$. This assumption is natural in the aforementioned contextual bandits application in Appendix A.

**Theorem 12.** *Adopt the semi-oblivious, smoothed online learning setting, where the adversary begins by choosing $g_{f^\star}^\star \in \mathcal{G}_{\mathcal{F}}$ from (6.3), and, at each time $t$, draws $x_t$ from a distribution that is $\sigma$-smooth*

---

[3]For simplicity, we interpret the $\arg\max$ lexicographically.

*with respect to $\mu$ and sets $y_t = g_{f^*}^*(x_t)$. Then, Algorithm 7 is an algorithm that is efficient in the number of calls to an ERM oracle over $\mathcal{G}_{\mathcal{F}}$ that satisfies for all $T$, with probability at least $1 - \delta$,*

$$\text{Reg}_T \leq 136K^2 d \log(d) + 91K^2 \log\left(TK^2/(\sigma\delta)\right) + K^2(\ell + 1). \tag{6.4}$$

In Appendix F.2 we prove a more general version of the above result that allows the regression functions on each piece to be polynomial. The intuition is to reduce $K$-piece regression to $K$-class classification, but where each of the "classes" materialize sequentially, once there are sufficiently many points observed to "determine" one of the pieces. The algorithm and proof are considerably more subtle, and are given in Appendices F.3 and F.4, respectively. We note that Algorithm 7 only requires the ERM oracle to be called on sets of size independent of $T$, making the total runtime of the algorithm logarithmic in the horizon.

## Acknowledgements

AB acknowledges support from the National Science Foundation Graduate Research Fellowship under Grant No. 1122374. We also would like to thank Sofiia Dubova for her help in translating Tarasov et al. [1988] and Michel Goemans for pointing to the closely related, English-language work of Khachiyan [1990].

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
