# Contents

# A Contextual Bandits

In this section, we apply Theorem 12 and the approach of Foster and Rakhlin [2020] to the setting of contextual bandits with contexts drawn from a smooth distribution, considered in Block et al. [2022]. Unlike in that work, however, we will realize regret bounds achievable by an oracle-efficient algorithm that are polynomially improved both in the horizon and the number of actions in the particular case of noiseless rewards that are piecewise linear.

We consider the following setting: the learner has access to the context set $\mathcal{B}_1^d$ and an action set $\mathcal{A}$ with $|\mathcal{A}| = A < \infty$. Let

$$\mathcal{G}_{\mathcal{F}}^{\mathcal{A}} = \left\{ \mathbf{g_f} = (\mathbf{g_f}^a)_{a \in \mathcal{A}} \,|\, \mathbf{g_f}^a \in \mathcal{G}_{\mathcal{F}} \right\}$$

where $\mathcal{G}_{\mathcal{F}}$ is as in Theorem 12, be a class of functions $\mathbf{g_f} : \mathcal{B}_1^d \times \mathcal{A} \to \mathbb{R}$. Before the game begins, Nature selects some $\ell^\star \in \mathcal{G}_{\mathcal{F}}^{\mathcal{A}}$ unknown to the learner. At each time $t$, Nature draws $x_t$ from a $\sigma$-smooth distribution on $\mathcal{B}_1^d$; the learner then chooses $a_t \in \mathcal{A}$, observes $\ell^\star(x_t, a_t)$ and suffers the same loss. Given $\ell^\star \in \mathcal{G}_{\mathcal{F}}^{\mathcal{A}}$, it is clear that the best policy, given a context is greedy:

$$\pi_{\ell^\star}(x) = \underset{a \in \mathcal{A}}{\operatorname{argmin}} \, \ell^\star(x, a).$$

The goal of the learner is to minimize regret, $\operatorname{Reg}_T$, to the optimal policy $\pi_{\ell^\star}$. The primary difference between our setting and that of Foster and Rakhlin [2020], Block et al. [2022], other than the fact that we are considering a particular function class $\mathcal{G}_{\mathcal{F}}^{\mathcal{A}}$, is that our losses are *noiseless*, while the prior works allow for some noise that is mean zero conditional on the history. We have the following regret bound:

**Corollary 13.** *Suppose that we are in the contextual bandit setting outlined above with $\mathcal{G}_{\mathcal{F}}$ from* (6.3) *and $\mathcal{X} \times \mathcal{A}$ identified with some subset of $\mathcal{B}_1^d$. Then there is an oracle-efficient algorithm that, for all $T$, with probability at least $1 - \delta$, achieves*

$$\operatorname{Reg}_T \leq 80 \cdot A \sqrt{T \left( K^2 d \log(d) + K^2 \log\left( ATK/(\sigma\delta) \right) \right)} + 8 \cdot \sqrt{AT \log\left(4/\delta\right)}.$$

We prove Corollary 13 in Appendix A.1 using the reduction of Foster and Rakhlin [2020, Theorem 1] and Theorem 12. Note that, in contradistinction to the corresponding bound proved as Block et al. [2022, Theorem 12], we achieve the optimal $\sqrt{T}$ regret, albeit with stronger assumptions on the setting.

## A.1 Proof of Corollary 13

In this section, we prove Corollary 13 by applying the black box reduction of Foster and Rakhlin [2020] to our Theorem 12. The key lemma is as follows:

**Lemma 14.** *Suppose that we are in the setting of Corollary 13 and that we predict $\hat{y}_t(a)$ and sample $a_t$ according to Algorithm 2. Then, for all $T$, with probability at least $1 - \delta$, we have*

$$\sum_{t=1}^{T} \mathbb{I}[\hat{y}_t(a_t) \neq \ell_t(a_t)] \leq A \left( 136 K^2 d \log(d) + 91 K^2 \log\left( \frac{4 A T^2 K^2}{\sigma\delta} \right) + K^2(\ell + 1) \right)$$

*Proof.* We begin by noting that

$$\sum_{t=1}^{T} \mathbb{I}[\hat{y}_t(a_t) \neq \ell_t(a_t)] = \sum_{a \in \mathcal{A}} \sum_{t=1}^{T} \mathbb{I}[\hat{y}_t(a) \neq \ell_t(a)] \mathbb{I}[a_t = a]$$

let

$$\mathcal{U} = \left\{ \text{for all } 1 \leq t \leq T \text{ and } a \in \mathcal{A} \text{ if } p_{t,a} \leq \frac{\delta}{2AT} \text{ then } a_t \neq a \right\}$$

A union bound implies that $\mathbb{P}(\mathcal{U}) \geq 1 - \frac{\delta}{2}$. Restricting to $\mathcal{U}$, we note that for any $B \subset \mathcal{B}_1^d$ measurable,

$$\mathbb{P}_t \left( x_t \in B | a_t = a \right) \leq \frac{\mathbb{P}_t(x_t \in B)}{p_{t,a_t}} \leq \frac{2AT\mu_d(B)}{\sigma\delta}$$

thus after restricting to $\mathcal{U}$, the distribution of $x_t$ conditioned on $a_t = a$ is $\left(\frac{\delta\sigma}{2AT}\right)$-smooth with respect to $\mu_d$. Thus for each $a$, we may apply the regret bound from Theorem 12 and, summing over $a \in \mathcal{A}$ concludes the proof. ∎

**Algorithm 2** Inverse Gap Weighting [Foster and Rakhlin, 2020] with Piecewise Regression

---
1: **Init:** $A$ instances of the Piecewise Regressor (Algorithm 7) **regressor**$(a)$ for $a \in \mathcal{A}$, learning rate $\gamma > 0$, exploration parameter $\mu > 0$.
2: **for** each time $t = 1, 2, \dots$ **do**
3:     **recieve** $x_t$
4:     **for** each action $a \in \mathcal{A}$ **do**
5:         **predict** $\hat{y}_t(a) = $ **regressor**$(a)$**.predict**$(x_t)$ `% Prediction step of Algorithm 7`
6:     Assign $b_t \leftarrow \operatorname{argmin}_{a \in \mathcal{A}} \hat{y}_t(a)$
7:     **for** each $a \neq b_t$ **do**
8:         Assign

$$ p_{t,a} \leftarrow \frac{1}{\mu + \gamma(\hat{y}_t(a) - \hat{y}_t(b_t))} \qquad \text{(\% Inverse Gap Weighting)} $$

        Assign

$$ p_{t,b_t} \leftarrow 1 - \sum_{a \neq b_t} p_{t,a} \qquad \text{(\% Inverse Gap Weighting)} $$

9:     **sample** $a_t \sim p_t$ and **play** $a_t$
10:     **observe** $\ell_t(a_t)$
11:     **update regressor**$(a)$**.update**$(x_t, a_t, \ell_t(a_t))$ `% Update step of Algorithm 7`

---

We can now prove Corollary 13:

*Proof of Corollary 13.* Note that by Lipschitzness and boundedness, twice the mistake bound is larger than the square loss regret considered in Foster and Rakhlin [2020]. Applying Foster and Rakhlin [2020, Theorem 1] concludes the proof. ∎

# B Preliminaries

In this section, we provide some key definitions and results that come up in our analysis. We divide the section by theme, with the first part collection results on probability and concentration, the second part on geometric measure theory, and the third on convex geometry.

## B.1 Probability and Concentration

We begin by stating the foundation of our regret bounds.

**Lemma 15** (Ville's Inequality [Ville, 1939]). *Let $\mathscr{F}_t$ denote a filtration and suppose that the sequence of random variables $A_t$ is a supermartingale with respect to $\mathscr{F}_t$. Suppose that*

$$\mathbb{P}\left(A_t > 0 \text{ for all } t > 0\right) = 1.$$

*Then for any $x > 0$, the following inequality holds:*

$$\mathbb{P}\left(\sup_{t>0} A_t \geq x\right) \leq \frac{\mathbb{E}\left[A_0\right]}{x}.$$

We will also require a standard Chernoff bound.

**Lemma 16** (Chernoff Bound). *Let $X_1, \ldots, X_t$ be a sequence of binary random variables such that $\mathbb{E}[X_i \mid X_1, \ldots, X_{i-1}] \geq \eta$. Then,*

$$\mathbb{P}\left[\sum_{i=1}^{t} X_i \leq t\eta/2\right] \leq \exp(-t\eta/8).$$

Finally, we will clear up any confusion about which distribution is smooth: that of contexts $x_t$ or that of samples $(x_t, y_t)$.

**Lemma 17.** *Suppose that $x \sim p$ and $(x, y) \sim \widetilde{p}$ where $p, \widetilde{p}$ are distributions. Suppose that $y \in \{\pm 1\}$. Then if $p$ is $\sigma$-smooth with respect to $\mu$ then $\widetilde{p}$ is $\left(\frac{\sigma}{2}\right)$-smooth with respect to $\mu \otimes \mathrm{Unif}(\{\pm 1\})$. Conversely, if $\widetilde{p}$ is $\sigma$-smooth with respect to $\mu \otimes \mathrm{Unif}(\{\pm 1\})$ then $p$ is $\sigma$-smooth with respect to $\mu$.*

*Proof.* The converse follows immediately from Lemma 35, proved in Appendix E.1. To prove the first statement, note that any distribution on $\{\pm 1\}$ is $\frac{1}{2}$-smooth with respect to $\mathrm{Unif}(\{\pm 1\})$. Thus, decomposing $\widetilde{p}(x, y) = p(x) \cdot \widetilde{p}(y|x)$ concludes the proof. ∎

## B.2 Geometric Measure Theory

The key definition is that of Hausdorff measure, which formally generalizes our intuitive notion of volume and surface area.

**Definition 18** (Hausdorff Measure [Federer, 2014]). *Let $\mathcal{X}$ be a metric space. For any $k \in \mathbb{R}_+$, we define the $k$-dimensional Hausdorff measure of a set $A \subset \mathcal{X}$ to be*

$$2^{-k} \omega_k \lim_{\varepsilon \downarrow 0} \mathrm{vol}_k^\varepsilon(A),$$

*where*

$$\mathrm{vol}_k^\varepsilon(A) := \inf \left\{ \sum_{i=1}^{\infty} (\mathrm{diam}\, U_i)^k \,\Big|\, A \subset \bigcup_{i=1}^{\infty} U_i \text{ and } \mathrm{diam}\, U_i < \varepsilon \right\}$$

*and $\mathrm{diam}\, U_i$ is the diameter of the set $U_i$, i.e., the maximal distance between any two points contained in $U_i$. We define the Hausdorff dimension $\dim(A) = \inf\{k > 0 \mid \mathrm{vol}_k(A) > 0\}$. As is common, when we integrate with respect to the Hausdorff measure, we denote the measure in the integral as $d\mathcal{H}^k$ instead of $d\,\mathrm{vol}_k$.*

Note that when $\mathcal{X} = \mathbb{R}^d$ then $\mathrm{vol}_d$ exactly coincides with the Lebesgue measure [Federer, 2014]. The following is an immediate consequence of the definition:

**Lemma 19.** *For a given set $A \subset \mathcal{X}$, let $N(A, \varepsilon)$ denote the minimal number of balls of radius $\varepsilon$ required to cover $A$. Then*

$$\mathrm{vol}_k(A) \leq \omega_k \varepsilon^k N(A, \varepsilon).$$

*Proof.* It is immediate from the definition that $\mathrm{vol}_k^\varepsilon$ is monotone nonincreasing as $\varepsilon \downarrow 0$. The result follows by letting $U_i$ be the set of balls of radius $\varepsilon$ covering $A$. ∎

We also use the co-area formula:

**Theorem 20** (Co-area Formula [Federer, 2014]). *Let* $\phi : \mathbb{R}^n \to \mathbb{R}^m$ *be a Lipschitz function with* $n \geq m$. *Then, for* $A \subset \mathbb{R}^m$,

$$\int_{\phi^{-1}(A)} \sqrt{\det(D\phi(x)D\phi(x)^T)} d\mathcal{H}^m(x) = \int_A \mathrm{vol}_{n-m}(\phi^{-1}(y)) d\mathcal{H}^m(y).$$

This in turn implies the projection formula:

**Corollary 21** ([Federer, 2014]). *Let* $\phi : \mathcal{X} \to \mathcal{Y}$ *denote a* 1-*Lipchitz map between* $m$-*dimensional sets* $\mathcal{X}, \mathcal{Y}$. *Then* $\mathrm{vol}_m(\phi(\mathcal{X})) \leq \mathrm{vol}_m(\mathcal{X})$.

*Proof.* By Theorem 20,

$$\mathrm{vol}_m(\phi(\mathcal{X})) = \int_{\phi(\mathcal{X})} d\mathcal{H}^m(y) \leq \frac{\sup_x \sqrt{\det(D\phi(x)D\phi(x)^T)}}{\inf_y \mathrm{vol}_0(\phi^{-1}(y))} \int_{\mathcal{X}} d\mathcal{H}^m(x) \leq \mathrm{vol}_m(\mathcal{X}),$$

where the last inequality holds because the Lipschitz assumption bounds the largest singular value of $D\phi$ and for any $y \in \phi(\mathcal{X})$, there is at least one point $x \in \phi^{-1}(y)$. ∎

### B.3 Convex Geometry

We first define a polytope:

**Definition 22.** *We say that a set* $A \subset \mathbb{R}^d$ *is a polytope if it is the intersection of a finite number of halfspaces. If* $A$ *is the intersection of* $K$ *halfspaces, we say that it has* $K$ *faces.*

We now define an ellipsoid:

**Definition 23.** *Let* $A \in \mathbb{R}^{d \times d}$ *be a positive definite matrix and let* $a \in \mathbb{R}^d$ *be a point. We define an ellipsoid to be*

$$\mathcal{E}(A, a) = \left\{ w \in \mathbb{R}^d | (w - a)^T A^{-1} (w - a) \leq 1 \right\}.$$

*Note that the volume of an ellipsoid is given by* $\mathrm{vol}_d(\mathcal{E}(A, a)) = \omega_d \sqrt{\det(A)}$.

We now define the John ellipsoid associated with a convex body:

**Theorem 24** (John Ellipsoid [John, 1948, Ball et al., 1997]). *Let* $A \subset \mathbb{R}^d$ *be a convex body, i.e., a convex set with nonempty interior. Then there is a unique ellipsoid* $\mathcal{E}_A$ *that has maximal volume subject to the condition that the ellipsoid is contained in* $A$. *Furthermore,* $A \subset d \cdot \mathcal{E}_A$.

We require the following general fact about ellipsoids:

**Lemma 25** (Corollary 15 from Rivin [2007]). *Suppose* $\mathcal{E} = \mathcal{E}(A, a)$ *is an ellipsoid and* $A$ *has eigenvalues given by* $q = (q_1, \ldots, q_d)$. *Then,*

$$\mathrm{vol}_{d-1}(\partial \mathcal{E}) \leq ||q|| \cdot \sqrt{d} \cdot \mathrm{vol}_d(\mathcal{E}). \tag{B.1}$$

Finally, we have the following result about cutting planes through the center of the John ellipsoid:

**Lemma 26** (Tarasov et al. [1988], Khachiyan [1990]). *Let* $A \subset \mathbb{R}^d$ *be a polytope with John ellipsoid* $\mathcal{E}_A$ *with center* $a$. *Let* $A'$ *be the intersection of* $A$ *and a halfspace going through* $a$, *i.e., there is some* $w \in \mathbb{R}^d$ *such that*

$$A' = A \cap \left\{ w \in \mathbb{R}^d | \langle w, x \rangle \geq \langle a, x \rangle \right\}.$$

*If* $\mathcal{E}_{A'}$ *is the John ellipsoid of* $A'$, *then*

$$\mathrm{vol}_d(\mathcal{E}_{A'}) \leq \frac{8}{9} \mathrm{vol}_d(\mathcal{E}_A).$$

# C  Technical Workhorses

This appendix proves the technical workhorses, the abstract decay lemma (Lemma 3), and the main geometric lemma, Lemma 4.

## C.1  Proof of Lemma 3: The Abstract Decay Lemma

We prove a slightly more general form of the lemma, with a weaker assumption on the sequence of $z$:

**Lemma 27.** *Suppose that a sequence $(\ell_t, z_t)$ satisfies $(R, c)$-geometric decay with respect to some a $\mu$ on $\mathcal{Z}$, and define a sequence of stopping times $t_m$ where $t_m = t$ if $t$ is the $m^{th}$ time that $\ell_s(z_s) = 1$. Let $m_t$ denote the maximal $m$ such that $t_m < t$ and thus $t_{m_t}$ is last time before $t$ that $\ell_s = 1$. Suppose that for all $t$, the distribution of $z_t$ conditional on $t_{m_t}$ is $\sigma$-smooth with respect to $\mu$. Then for all $T \in \mathbb{N}$, with probability at least $1 - \delta$,*

$$\sum_{t=1}^{T} \ell_t(z_t) \leq 4 \frac{\log\left(\frac{2TR}{\sigma\delta}\right)}{\log\left(\frac{1}{c}\right)} + \frac{e-1}{1-\sqrt{c}}. \tag{C.1}$$

*Proof.* Fix a sequence of positive integers $h_k$ for $k \in \mathbb{N}$, whose values we tune at the end of the proof. Let $\tau_0 = 0$ and for all $m > 0$, let

$$\tau_m = \tau_{m-1} + \inf\left\{ k > 0 \,\middle|\, \sum_{t=\tau_{m-1}+(k-1)h_m}^{\tau_{m-1}+kh_m} \ell_t(z_t) = 1 \right\}$$

$$= \tau_{m-1} + \inf\left\{ k > 0 \,\middle|\, \exists t \in \tau_{m-1} + [(k-1)h_m, kh_m - 1] \text{ s.t. } \ell_t(z_t) = 1 \right\}.$$

Furthermore, let $T(m) = \sum_{k=1}^{m}(\tau_k - \tau_{k-1})h_k$ and

$$t_m = \inf\left\{ t > T(m-1) | \ell_t(z_t) = 1 \right\}. \tag{C.2}$$

In words, we consider epochs of length $h_m$, whose length can change every time we make a mistake in an epoch. We have $T(m)$ the time of the $m^{th}$ change of epoch and $\tau_m$ the number of epochs of length $h_m$ we have to go before we make a mistake; we also have $t_m$ is the time of the first mistake after the $m^{th}$ change of epoch size. Let

$$A_m = \sum_{s=t_m+1}^{T(m)-1} \ell_s(z_s). \tag{C.3}$$

be the number of mistakes in a given epoch other than the first mistake. Let $\pi_m = \min\left(\frac{R_m}{\sigma}, 1\right)$, where we abbreviate $R_m = R_{t_m}$. We first claim that with probability at least $1 - \delta$, for all $m$ it holds that:

$$A_m \leq \log\left(\frac{1}{\delta}\right) + (e-1)\sum_{k=1}^{m} \pi_k(h_k - 1). \tag{C.4}$$

To see this, let

$$B_m^\lambda = \exp\left( \lambda A_m - \left(e^\lambda - 1\right)\sum_{k=1}^{m} \pi_k h_k \right).$$

We show that $B_m^\lambda$ is a supermartingale for all $\lambda > 0$. To see this, we have

$$\mathbb{E}\left[B_m^\lambda | B_{m-1}^\lambda\right] = B_{m-1}^\lambda \mathbb{E}\left[ \exp\left( \lambda \sum_{s=t_m+1}^{T(m)-1} \mathbb{I}[\hat{y}_s \neq y_s] - \left(e^\lambda - 1\right)\pi_m(h_m - 1) \right) \middle| B_{m-1}^\lambda \right] \leq B_{m-1}^\lambda,$$

where the inequality follows because the conditional probability of a mistake for $t_m + 1 \leq T(m) - 1 \leq \pi_m$ by the assumption of smoothness conditional on a sub-sigma algebra of that generated by $t_{m-1}$ and realizability and $T(m) - 1 - (t_m + 1) \leq h_m - 1$ by construction. Thus we may apply Ville's inequality from Lemma 15 and recover (C.4).

**Claim 1.** *With probability at least $1 - \delta$, it holds for all $m$ that*

$$\tau_m - \tau_{m-1} \geq \max\left(1, \log\left(\frac{\delta}{\pi_m h_m T}\right)\right). \tag{C.5}$$

*Proof of Claim 1.* For any $\tau_{m-1} + (k-1)h_m \leq t < \tau_{m-1} + kh_m$, smoothness implies

$$\mathbb{P}\left(\ell_t(z_t) = 1 | \ell_s(z_s) = 0 \text{ for all } s < t\right) \leq h_m \pi_m. \tag{C.6}$$

where we note that the event that $\ell_s(z_s) = 0$ for $s < t$ is contained in the sigma-algebra generated by $t_{m_t}$. A union bound then implies that

$$\mathbb{P}\left[\exists t \in \tau_{m-1} + [(k-1)h_m, \, kh_m) \text{ s.t. } \ell_t(z_t) = 1 \Big| \ell_s(z_s) = 0, \forall s \in [\tau_{m-1}, (k-1)h_m)]\right] \leq h_m \pi_m.$$

Hence, letting $X_m$ be a random variable distributed geometrically with parameter $\widetilde{\pi}_m = \min(h_m \pi_m, 1)$, $\tau_m - \tau_{m-1}$ stochastically dominates $X_m$. Thus, for any $\lambda < -\log(1 - \pi_m)$,

$$\mathbb{E}\left[e^{\lambda(\tau_m - \tau_{m-1})}\right] \leq \mathbb{E}\left[e^{\lambda X_m}\right] = \frac{\widetilde{\pi}_m e^\lambda}{1 - (1 - \widetilde{\pi}_m)e^\lambda}. \tag{C.7}$$

We further note that

$$\log\left(1 - (1 - \widetilde{\pi}_m)e^{-1}\right) \geq 1 - \frac{1}{1 - (1 - \widetilde{\pi}_m)e^{-1}} = -\frac{(1 - \widetilde{\pi}_m)e^{-1}}{1 - (1 - \widetilde{\pi}_m)e^{-1}} \tag{C.8}$$

$$\geq -\frac{e^{-1}}{1 - e^{-1}}. \tag{C.9}$$

Thus, setting $\lambda = -1$, we see that with probability at least $\frac{\delta}{T}$,

$$\tau_m - \tau_{m-1} \geq 1 + \log\left(\frac{1}{\widetilde{\pi}_m}\right) - \log\left(\frac{T}{\delta}\right) - \frac{e^{-1}}{1 - e^{-1}} \geq \log\left(\frac{\delta}{h\pi_m T}\right). \tag{C.10}$$

Because $\tau_m - \tau_{m-1} > 0$ by construction, we may then take a union bound to conclude the proof of the claim. ∎

Now we note that

$$T \geq T(m) = \sum_{k=1}^{m}(\tau_k - \tau_{k-1})h_k \tag{C.11}$$

and, further, that if $m_T$ is the maximal $m$ such that the preceding display holds,

$$\text{Reg}_T \leq m_T + A_{m_T}. \tag{C.12}$$

Thus, combining (C.4) and (C.5), along with the fact that $\pi_k \leq c^k R_0/\sigma$, with probability at least $1 - 2\delta$, we have

$$T \geq \sum_{k=1}^{m_T} \log\left(\frac{\sigma\delta}{c^k R_0 h_k T}\right) h_k \tag{C.13}$$

$$\text{Reg}_T \leq m_T + \log\left(\frac{1}{\delta}\right) + (e - 1)\sum_{k=1}^{m} c^k \frac{R_0}{\sigma}(h_k - 1) \tag{C.14}$$

Now, let $h_k = 1$ for $k \leq 2\log\left(\frac{TR_0}{\sigma\delta}\right)/\log\left(\frac{1}{c}\right)$ and let $h_k = c^{-\frac{k}{2}}$ otherwise. Then we see that if (C.13) and (C.14) hold, then

$$m_T \leq 2\frac{\log T}{\log\left(\frac{1}{c}\right)} + \frac{2\log\left(\frac{TR_0}{\sigma\delta}\right)}{\log\left(\frac{1}{c}\right)} \leq -4\frac{\log\left(\frac{TR_0}{\sigma\delta}\right)}{\log c}, \tag{C.15}$$

and

$$\sum_{k=1}^{m} c^k \frac{R_0}{\sigma}(h_k - 1) \leq \sum_{j=0}^{\infty} c^{\frac{j}{2}} = \frac{1}{1 - \sqrt{c}}. \tag{C.16}$$

Thus we see that with probability at least $1 - \delta$.

$$\text{Reg}_T \leq 4\frac{\log\left(\frac{2TR_0}{\sigma\delta}\right)}{\log\left(\frac{1}{c}\right)} + \frac{e - 1}{1 - \sqrt{c}}. \tag{C.17}$$

which proves the result. ∎

We note that Lemma 3 follows immediately because $t_{m_t} < t$ almost surely and so the sigma algebra generated by $t_{m_t}$ is contained in that generated by the history up to $t - 1$ and so the smoothness assumption of Lemma 3 implies that of Lemma 27.

## C.2 Proof of Lemma 4: The Key Geometric Lemma

We begin by proving the following technical geometric lemma, which, for simplicity, considers subsets of the sphere, rather than of the ball. This ultimately suffices due to positive homogeneity of linear classifiers.

**Lemma 28.** *Let $\hat{\mathcal{F}} \subset S^{d-1}$ be a measurable subset of the $(d-1)$-dimensional sphere imbedded in $\mathbb{R}^d$. Let $D(\hat{\mathcal{F}})$ denote the set of points in $S^{d-1}$ orthogonal to at least one point in $\hat{\mathcal{F}}$, i.e.,*

$$D(\hat{\mathcal{F}}) = \left\{ x \in S^{d-1} | \text{for some } w \in \hat{\mathcal{F}}, \quad \langle w, x \rangle = 0 \right\}. \tag{C.18}$$

*Then, if $\mathrm{vol}_k$ is the $k$-dimensional Hausdorff measure on the sphere, we have*

$$\mathrm{vol}_{d-1}(D(\hat{\mathcal{F}})) \leq 2 \cdot 4^{d-1} \, \mathrm{vol}_{d-1}(\hat{\mathcal{F}}) + 4^{d+1} \, \mathrm{vol}_{d-2}(\partial \hat{\mathcal{F}}). \tag{C.19}$$

*Proof.* For a given set $A \subset S^{d-1}$, denote by $T(A, \varepsilon)$ the "tube" of radius $\varepsilon$ around $A$, i.e., the set of points in $S^{d-1}$ with distance at most $\varepsilon$ from a point in $A$.

Note that for any fixed point $w \in \hat{\mathcal{F}}$, we have $D(w)$ is just the $(d-2)$-sphere formed by intersection the linear space orthogonal to $w$ with $S^{d-1}$. If $\hat{B}_\varepsilon(w)$ denotes the $\varepsilon$-ball around $w$ in $S^{d-1}$, then we claim that $D(\hat{B}_\varepsilon(w)) \subset T(D(w), \varepsilon)$. Indeed, suppose that $v \in \hat{B}_\varepsilon(w)$ so that $\langle w', v \rangle = 0$ for some $w' \in \hat{B}_\varepsilon(w)$. Let $\alpha$ be a member of the orthogonal group such that $\alpha w' = v$ and $\langle \alpha w, w \rangle = 0$. Then $\langle v + \alpha(w - w'), w \rangle = 0$ and $\|\alpha(w - w')\| = \|w - w'\| \leq \varepsilon$, proving the claim.

Let $N(\hat{\mathcal{F}}, \varepsilon)$ denote the minimum size of an $\varepsilon$-net of $\mathcal{F}$ and let $P(\hat{\mathcal{F}}, \varepsilon)$ denote the maximum size of an $\varepsilon$-packing. By abuse of notation, we will also use $N(\hat{\mathcal{F}}, \varepsilon)$ to denote the minimal $\varepsilon$-net itself. The fact that $D(\hat{B}_\varepsilon(w)) \subset T(D(w), \varepsilon)$ implies that

$$\mathrm{vol}_{d-1}(D(\hat{\mathcal{F}})) \leq \mathrm{vol}_{d-1} \left( \bigcup_{w \in N(\mathcal{F}, \varepsilon)} T(D(w), \varepsilon) \right) \leq N(\hat{\mathcal{F}}, \varepsilon) \cdot \mathrm{vol}_{d-1}(T(D(w), \varepsilon)) \tag{C.20}$$

By packing, covering duality, we have

$$N(\hat{\mathcal{F}}, \varepsilon) \leq P\left(\hat{\mathcal{F}}, \frac{\varepsilon}{2}\right) \leq \frac{2^{d-1} \, \mathrm{vol}_{d-1}(T(\hat{\mathcal{F}}, \varepsilon))}{\mathrm{vol}_{d-1}(\hat{B}_\varepsilon(w))} \tag{C.21}$$

Now, we may apply Gray [2003, Theorem 10.20], the generalization of Steiner's formula to submanifolds of a sphere, to get

$$\mathrm{vol}_{d-1}(T(\hat{\mathcal{F}}, \varepsilon)) \leq \mathrm{vol}_{d-1}(\hat{\mathcal{F}}) + \mathrm{vol}_{d-2}(\partial \hat{\mathcal{F}}) \left( 2^{d-1} \varepsilon + 2^{d-1} \varepsilon^d \right) \tag{C.22}$$

Putting this together, we have

$$\mathrm{vol}_{d-1}(D(\hat{\mathcal{F}})) \leq 2^{d-1} \frac{\mathrm{vol}_{d-1}(T(D(w), \varepsilon))}{\mathrm{vol}_{d-1}(B_\varepsilon(w)))} \left( \mathrm{vol}_{d-1}(\hat{\mathcal{F}}) + \mathrm{vol}_{d-2}(\partial \hat{\mathcal{F}}) \left( 2^{d-1} \varepsilon + 2^{d-1} \varepsilon^d \right) \right). \tag{C.23}$$

Now we may apply Weyl's tube formula [Weyl, 1939] (see Gray [2003], Lotz [2015] for a clear exposition on the topic) to $S^{d-2}$ imbedded as the equator of $S^{d-1}$ to get that for any $\varepsilon < 1$,

$$\frac{\mathrm{vol}_{d-1}(T(D(w), \varepsilon))}{\mathrm{vol}_{d-1}(\hat{B}_\varepsilon(w))} \leq \frac{2\omega_{d-1} \left( (1+\varepsilon)^{d-1} - (1-\varepsilon)^{d-1} \right)}{\varepsilon^{d-1} \omega_{d-1}} = 2 \frac{\left( (1+\varepsilon)^{d-1} - (1-\varepsilon)^{d-1} \right)}{\varepsilon^{d-1}}. \tag{C.24}$$

As $\varepsilon \uparrow 1$, the above expression tends to $2^d$. Putting everything together, we have

$$\mathrm{vol}_{d-1}(D(\hat{\mathcal{F}})) \leq 2 \cdot 4^{d-1} \, \mathrm{vol}_{d-1}(\hat{\mathcal{F}}) + 2 \cdot 2^{2d+1} \, \mathrm{vol}_{d-2}(\partial \hat{\mathcal{F}}). \tag{C.25}$$

as desired. ∎

We now use the homogenity of the inner product to show that it suffices to consider the sphere:

**Lemma 29.** *Let $\mathcal{F} \subset \mathcal{B}_1^d$ and let $\widehat{\mathcal{F}}_t$ denote its projection to $S^{d-1}$. Suppose that $\mathcal{F}$ is such that*

$$\mathcal{F} = \left\{ r\widehat{x} \mid 0 \leq r \leq 1 \text{ and } \widehat{x} \in \widehat{\mathcal{F}} \right\}.$$

*Then,*

$$\frac{\text{vol}_d(\mathcal{F})}{\text{vol}_d(\mathcal{B}_1^d)} = \frac{\text{vol}_{d-1}(\widehat{\mathcal{F}})}{\text{vol}_{d-1}(S^{d-1})}.$$

*Proof.* Let $\widehat{\mathcal{F}}_r = r\widehat{\mathcal{F}}$. Then we see from Theorem 20 that $\text{vol}_{d-1}(\widehat{\mathcal{F}}_r) = r^{d-1} \text{vol}_{d-1}(\widehat{\mathcal{F}})$. Thus,

$$\text{vol}_d(\mathcal{F}) = \int_0^1 \text{vol}_{d-1}(\widehat{\mathcal{F}}_r) dr = \text{vol}_{d-1}(\widehat{\mathcal{F}}) \int_0^1 r^{d-1} dr.$$

In particular, this holds for $\mathcal{F} = \mathcal{B}_1^d$. Thus, we have

$$\frac{\text{vol}_d(\mathcal{F})}{\text{vol}_d(\mathcal{B}_1^d)} = \frac{\text{vol}_{d-1}(\widehat{\mathcal{F}}) \int_0^1 r^{d-1} dr}{\text{vol}_{d-1}(\widehat{\mathcal{B}_1^d}) \int_0^1 r^{d-1} dr} = \frac{\text{vol}_{d-1}(\widehat{\mathcal{F}})}{\text{vol}_{d-1}(S^{d-1})}.$$

as desired. ∎

We now put everything together:

*Proof of Lemma 4.* Let $\widehat{D}(\mathcal{F})$ be the set of $x \in D$ such that $||x|| = 1$ and let $\widehat{\mathcal{F}}$ be defined similarly. By the positive homogeneity of both $D(\mathcal{F})$ and $\mathcal{F}$, we have

$$\mu_d(D) = \frac{\text{vol}_d(D)}{\text{vol}_d(\mathcal{B}_1)} = \frac{\text{vol}_{d-1}(\widehat{D}(\mathcal{F}))}{\text{vol}_{d-1}(\partial \mathcal{B}_1)} \tag{C.26}$$

$$\mu_d(\mathcal{F}) = \frac{\text{vol}_d(\mathcal{F})}{\text{vol}_d(\mathcal{B}_1)} = \frac{\text{vol}_{d-1}(\widehat{\mathcal{F}})}{\text{vol}_{d-1}(\partial \mathcal{B}_1)} \tag{C.27}$$

where $\text{vol}_{d-1}(\cdot)$ denotes the $(d-1)$-dimensional Hausdorff measure. Thus, it suffices to compare $\text{vol}_{d-1}(\widehat{D}(\mathcal{F}))$ with $\text{vol}_{d-1}(\widehat{\mathcal{F}})$, which is the content of Lemma 28. The result follows. ∎

# D Proofs From Section 4

In this appendix, we provide proofs of Theorem 5 and Proposition 6.

## D.1 Proof of Theorem 5

In order to apply Lemma 3 to prove Theorem 5, we need to show that the loss functions satisfy $(R, c)$-geometric decay. We will show this for $R = 4^{d+1}d^{2d}$ and $c = \frac{8}{9}$ using Lemma 4 and some more convex geometry. We begin by proving the following characterization of the disagreement region, which will in turn allow us to apply Lemma 4:

**Lemma 30.** *Suppose that we are in the situation of Theorem 5. Then we have*

$$D_t \subset \left\{ x \in \mathcal{B}_1^d \,|\, \langle w, x \rangle = 0 \text{ for some } w \in \mathcal{F}_t \right\}. \tag{D.1}$$

*Proof.* Recall that the version space is defined as

$$\mathcal{F}_t = \{ w \in \mathcal{F} \,|\, \langle w, y_s x_s \rangle \geq 0 \text{ for all } s < t \}. \tag{D.2}$$

If $x \in D_t$, then there are $w, w' \in \mathcal{F}_t$ such that $\text{sign}(\langle w', x \rangle) \neq \text{sign}(\langle w'', x \rangle)$. Consider the continuous function $h(\lambda) = \langle \lambda w'' + (1 - \lambda)w', x \rangle$; by the intermediate value theorem, there is some $0 < \lambda^\star < 1$ and $w = \lambda^\star w'' + (1 - \lambda^\star)w'$ such that $\langle w, x \rangle = 0$. By convexity of $\mathcal{F}_t$, then $w \in \mathcal{F}_t$ and thus $D_t$ is contained within the set of points orthogonal to at least one point in $\mathcal{F}_t$. ∎

With Lemma 30 in hand, we will be able to apply Lemma 4 and it will suffice to control $\mu(\mathcal{F}_t)$ and $\text{vol}_{d-1}(\mathcal{F}_t)$. The next result bounds these quantities in terms of their analogues in $\mathcal{E}_t$:

**Lemma 31.** *Let $\mathcal{F} \subset \mathbb{R}^d$ be a convex body with John ellipsoid $\mathcal{E}$. Then we have*

$$\mu(\mathcal{F}) \leq d^d \mu(\mathcal{E}) \qquad\qquad \text{vol}_{d-1}(\partial \mathcal{F}) \leq 2d^d \mu(\mathcal{E}). \tag{D.3}$$

*Proof.* Note that it is a classical fact that $\mathcal{F} \subset d \cdot \mathcal{E}$ [John, 1948] and thus $\mu(\mathcal{F}) \leq d^d \mu(\mathcal{E})$. We thus only have to prove the second bound. To do this we first note that

$$\text{vol}_{d-1}(\partial \mathcal{F}) \leq \text{vol}_{d-1}(\partial(d \cdot \mathcal{E})). \tag{D.4}$$

To see that this is the case, consider $\pi : \partial(d \cdot \mathcal{E}) \to \partial \mathcal{F}$ be projection onto the convex $\mathcal{F}$. Then $\pi$ is a contraction and thus shrinks Hausdorff measure as per Corollary 21. We now apply Lemma 25 and note that because our ellipsoids are contained in the ball, $||q|| \leq 2 \cdot \sqrt{d}$, where $q$ is the vector of semi-axis lengths, i.e., the eigenvalues of the associated positive definite matrix. Thus we have

$$\text{vol}_{d-1}(\partial(d \cdot \mathcal{E})) = d^{d-1} \text{vol}_{d-1}(\partial \mathcal{E}) \leq d^{d-1} \left( 2d\mu(\mathcal{E}) \right), \tag{D.5}$$

and the result follows. ∎

We are now finally ready to apply the geometry that we have done so far to prove a slightly more general form of Theorem 5, which we will require to apply some of our reductions below.

**Proposition 32.** *Suppose we are in the situation of Theorem 5 with the added complication that at any time $t$, the adversary can choose to censor round $t$ from the learner, so the learner does not observe $y_t$ and does not suffer loss at time $t$. We further allow $x_t$ to be drawn adversarially with the condition that if $x_t$ is drawn adversarially, then the adversary always censors time $t$. Then the conclusion of Theorem 5 holds.*

*Proof.* By Lemma 3, it suffices to show that with our choice of $w_t$, the sequence

$$\ell_t = \mathbb{I} \left[ \text{sign}(\langle w_t, x_t \rangle) \neq y_t \text{ and round } t \text{ is not censored} \right]$$

satisfies $(R, c)$ geometric decay with respect to $\mu$ for come $R, c$. In particular, we need to find an $R \geq \mu(D_1)$ and a $c < 1$ such that if we make a mistake, then $\mu(D_{t+1}) \leq c\mu(D_t)$. Note that if $\ell_t = 1$ then we must have $x_t \in D_t$ and $y_t$ is not censored. By Lemma 30 it suffices to control the size of the

set of points orthogonal to at least one $w \in \mathcal{F}_t$; by Lemma 4, it in turn suffices to control $\mu(\mathcal{F}_t)$ and $\mathrm{vol}_{d-1}(\mathcal{F}_t)$. Applying Lemma 31 to the preceding logic, we have:

$$\mu(D_t) \leq 2 \cdot 4^{d-1}\mu(\mathcal{F}_t) + \frac{4^{d+1}}{\omega_d}\mathrm{vol}_{d-1}(\partial\mathcal{F}_t) \tag{D.6}$$

$$\leq 2 \cdot 4^{d-1} \cdot d^d\mu(\mathcal{E}_t) + \frac{4^{d+1}}{\omega_d} \cdot 2 \cdot d^d\mu(\mathcal{E}_t) \tag{D.7}$$

$$\leq 4^{d+1}d^{2d}\mu(\mathcal{E}_t). \tag{D.8}$$

As $\mu(\mathcal{E}_t) \leq 1$, we may choose $R = 4^{d+1}d^{2d}$ and reduce to showing that every time we make a mistake, $\mu(\mathcal{E}_{t+1}) \leq c\mu(\mathcal{E}_t)$ for some $c$.

Now, suppose that we make a mistake at time $t$, i.e., $\langle w_t, y_t x_t \rangle < 0$. Then, we have

$$\mathcal{F}_{t+1} = \mathcal{F}_t \cap \{w \in \mathcal{F}|\langle w, x_t y_t\rangle > 0\} \subset \mathcal{F}_t \cap \{w \in \mathcal{F}|\langle w, x_t y_t\rangle \geq \langle w_t, x_t y_t\rangle\}. \tag{D.9}$$

by monotonicity. Thus $\mathcal{F}_{t+1}$ is a subset of the intersection of $\mathcal{F}_t$ and a halfspace through the center of $\mathcal{E}_t$. Thus, by Lemma 26, $\mathrm{vol}(\mathcal{E}_{t+1}) \leq \frac{8}{9}\mathrm{vol}(\mathcal{E}_t)$. Thus, we may choose $c = \frac{8}{9}$ and conclude the proof. ∎

We remark that Theorem 5 trivially follows from Proposition 32 by restricting the adversary to never censor a time $t$.

## D.2  Proof of Theorem 6

We construct separate adversaries which regret $\mathbb{E}[\mathrm{Reg}_T] \geq \Omega(d)$ and $\mathbb{E}[\mathrm{Reg}_T] \geq \Omega\left(\log\left(\frac{T}{\sigma}\right)\right)$. Randomizing between the two with probability one-half gives the lower bound.

We first note that any algorithm must experience $\mathbb{E}[\mathrm{Reg}_T] \geq \frac{d+1}{2}$ against some adversary; indeed, as a generic set of $d+1$ points defines a hyperplane, a realizable adversary can choose $y_t$ as independent Rademachers for each $1 \leq d \leq d+1$ and the learner will suffer expected regret $\frac{d+1}{2}$.

We now construct an adversary in one dimension that will force $\mathbb{E}[\mathrm{Reg}_T] \geq \Omega\left(\log\left(\frac{T}{\sigma}\right)\right)$; by projecting onto some fixed direction, the higher dimensional case reduces to this setting. Thus, suppose that the $x_t$ are required to be sampled from a distribution that is $\sigma$-smooth with respect to the uniform measure on the unit interval. At each time $t$, let $D_t$ be the interval between the rightmost $x_s$ labelled $-1$ and the leftmost $x_s$ labelled $1$, let $R_t$ be its length and $w_t$ its midpoint. Fix $0 < \varepsilon < 1$ to be tuned later and let

$$\widetilde{D}_t = \left\{x \in D_t | \frac{1-\varepsilon}{2}R_t \leq |x - w_t| \leq \frac{1}{2}R_t\right\} \tag{D.10}$$

be the set of points in the disagreement region close to its boundary. We let the adversary select the distribution that picks uniformly from $\widetilde{D}_t$ with probability $\min\left(\frac{\mu(\widetilde{D}_t)}{\sigma}, 1\right)$ and with remaining probability selects 0. If $|x_t - w_t| \geq \frac{R_t}{2}$, then $y_t$ is determined by realizability. Otherwise, let $y_t$ be an independent Rademacher random variable.

Let $\pi_m = \min\left(\frac{\varepsilon R_m}{\sigma}, 1\right)$ and $t_m$ be the $m^{th}$ time that $x_t \in \widetilde{D}_t$, we see that $t_{m+1} - t_m$ is geometrically distributed with parameter $\pi_m$ and thus

$$B_m^\lambda = \exp\left(\lambda t_m - m\lambda - \sum_{k=1}^m \log\left(\frac{\pi_k}{1-(1-\pi_k)e^\lambda}\right)\right) \tag{D.11}$$

is a supermartingale for $\lambda < \min_{k \leq m}(-\log(1-\pi_k)) = -\log(1-\pi_m)$. Note that by construction, $R_{m+1} \geq (1-\varepsilon)R_m$ and thus $R_m \geq (1-\varepsilon)^m$. Setting $\lambda = \pi_m \leq -\log(1-\pi_m)$ and applying Ville's Inequality, Lemma 15, we get that with probability at least $1-\delta$, for all $m$,

$$t_m \leq m + \frac{\log\left(\frac{1}{\delta}\right)}{\pi_m} + \frac{1}{\pi_m}\sum_{k=1}^m \log\left(\frac{\pi_k}{1-(1-\pi_k)e^{\pi_k}}\right). \tag{D.12}$$

We now note that

$$\frac{1}{\pi_m} \log \left( \frac{\pi_k}{1 - (1 - \pi_k)e^{\pi_k}} \right) \leq \frac{1}{\pi_m} \left( \frac{\pi_m}{1 - (1 - \pi_m)e^{\pi_m}} - 1 \right) \tag{D.13}$$

$$= \frac{e^{\pi_m} - 1}{\pi_m} \frac{1 - \pi_m}{1 - (1 - \pi_m)e^{\pi_m}} \tag{D.14}$$

$$\leq (e - 1)\frac{2}{\pi_m^2} \tag{D.15}$$

using monotonicity, the fact that $\pi_m \leq 1$ and the following computation:

$$1 - (1 - x)e^x = 1 - (1 - x)\sum_{k=0}^{\infty} \frac{x^k}{k!} = \sum_{k=2}^{\infty} x^k \left( \frac{1}{(k-1)!} - \frac{1}{k!} \right) \geq \frac{x^2}{2}. \tag{D.16}$$

Now, using the fact that $\pi_m \geq \frac{\varepsilon}{\sigma}(1 - \varepsilon)^m$, we have

$$t_m \leq m + \frac{\sigma}{\varepsilon}(1 - \varepsilon)^{-m} \log \left( \frac{1}{\delta} \right) + 2(e - 1)m \left( \frac{\varepsilon}{\sigma} \right)^{-2} (1 - \varepsilon)^{-2m}. \tag{D.17}$$

Setting $\varepsilon = 1 - e^{-1}$, we see that there is some constant $c > 0$ such that with probability at least $1 - \delta$,

$$t_m \leq c \max \left( \sigma e^m \log \left( \frac{1}{\delta} \right), m\sigma^2 e^{2m} \right). \tag{D.18}$$

In particular, there is a universal constant $C$ such that if $m = C \log \left( \frac{T}{\sigma \log\left(\frac{1}{\delta}\right)} \right)$, then with probability at least $1 - \delta$ we have $\mathrm{Reg}_T \geq m$ because the probability of a mistake, given that $x_t \in \widetilde{D}_t$ is $\frac{1}{2}$. The result follows.

### D.3 Lower bound against naive play.

In this section, we show that it is necessary to choose the half-spaces $w_t$ intelligently in order to attain logarithmic-in-$1/\sigma$ regret.

Consider $d = 1$, so $\mu_1$ is the uniform measure on the interval $[-1, 1]$. We define $\mathcal{F}^{\mathrm{thres}} := \{x \mapsto \mathrm{sign}(x - c)\}$ as the set of (monotone) threshold classifiers. Given a function class $\mathcal{F}$, we say that a learning strategy is *consistent*, if at each $t \in [T]$, it selects an $f_t \in \mathcal{F}_t$ in the version space $\mathcal{F}_t := \{f \in \mathcal{F} : f(x_s) = y_s, \quad 1 \leq s \leq t - 1\}$. Define the left and right endpoints of the negative and positive regions

$$\tilde{x}_t := \max \left\{ -1, \max_{1 \leq s \leq t} \{x_t : y_t = -1\} \right\}, \quad \bar{x}_t := \min \left\{ 1, \min_{1 \leq s \leq t} \{x_t : y_t = 1\} \right\}.$$

For a given $\eta > 0$, we consider the strategy

$$\hat{y}_t = \begin{cases} \mathrm{sign}\left(x_t - \tilde{x}_{t-1} - \eta\right) & \tilde{x}_{t-1} + \eta < \bar{x}_t \\ \mathrm{sign}\left(x_t - \frac{1}{2}\left(\tilde{x}_{t-1} + \bar{x}_{t-1}\right)\right) & \text{otherwise.} \end{cases} \tag{D.19}$$

This is consistent with $\mathcal{F}^{\mathrm{thres}}$, since the thresholds are always chosen strictly between $\tilde{x}_{t-1}$ and $\bar{x}_{t-1}$. However, the strategy is very naive, since it defaults to setting the threshold only slightly to the right of $\tilde{x}_{t-1}$. As a consequence, we show it suffers $\Omega(1/\sigma)$ regret when $\eta$ is small.

**Proposition 33.** *Fix $\eta > 0$. For $T \geq 1$ and $\sigma \in (1/T, 1/4]$, there exists an $\mathcal{F}^{\mathrm{thres}}$-realizable, $\sigma$ smooth adversary such that the strategy in Equation (D.19) suffers expected regret linear in $1/\sigma$ for $\eta$ small:*

$$\mathbb{E}[\mathrm{Reg}_T] \geq \lfloor\frac{1}{\sigma}\rfloor \cdot \left(1 - \frac{\eta}{2\sigma}\right).$$

*Proof.* At each time $1 \leq t \leq T_0 := \lfloor 1/\sigma \rfloor$, the adversary selects

$$x_t = -1 + 2\sigma(t - 1) + 2\sigma a_t, \quad a_t \sim \mathrm{Unif}([0, 1]).$$

For times $t \geq T_0$, the adversary selects $x_t \sim \text{Unif}([-1,1])$. This adversary is clearly $\sigma$ smooth, satisfies $x_t \in -1 + 2\sigma[t-1,t]$ until $T_0$, and then plays arbitrarily. Moreover, for $\sigma \leq 1/T$, $x_t \in [-1,1]$ for all $t$. Fixing a ground-truth classifier $f^\star(x) = \text{sign}(x-1)$, we see $y_s = f^\star(x_t) = -1$ is realizable for all $t$.

Lastly, we analyze the regret of Equation (D.19); notice that under the above adversary, $\bar{x}_t = 1$, so we are always in the first case $y_t = \text{sign}(x_t - \max_{1 \leq s \leq t-1} x_s - \eta)$. Then, for any $t \leq T_0$,

$$
\begin{aligned}
\mathbb{P}\left(\hat{y}_t = y_t | \mathscr{F}_{t-1}\right) &= \mathbb{P}\left(x_t \leq \eta + \max_{1 \leq s \leq t-1} x_s | \mathscr{F}_{t-1}\right) \\
&\leq \mathbb{P}\left[x_t \leq \eta + 2(t-1)\sigma - 1 \mid \mathscr{F}_{t-1}\right] \\
&= \mathbb{P}_{a_t \sim \text{Unif}([0,1])}\left[-1 + 2\sigma(t-1) + 2\sigma a_t \leq \eta + 2(t-1)\sigma - 1\right] \\
&= \mathbb{P}_{a_t \sim \text{Unif}([0,1])}\left[a_t \leq \frac{\eta}{2\sigma}\right] = \frac{\eta}{2\sigma}.
\end{aligned}
$$

Hence,

$$
\begin{aligned}
\mathbb{E}[\text{Reg}_t] &= \sum_{t=1}^{T} \mathbb{E}[\mathbb{P}[\hat{y}_t \neq y_t \mid \mathscr{F}_{t-1}]] \\
&\geq \sum_{t=1}^{T_0} \mathbb{E}[\mathbb{P}[\hat{y}_t \neq y_t \mid \mathscr{F}_{t-1}]] \\
&= \sum_{t=1}^{T_0} 1 - \mathbb{E}[\mathbb{P}[\hat{y}_t = y_t \mid \mathscr{F}_{t-1}]] \\
&\geq \sum_{t=1}^{T_0} \left(1 - \frac{\eta}{2\sigma}\right) = T_0\left(1 - \frac{\eta}{2\sigma}\right), \quad T_0 := \lfloor 1/\sigma \rfloor.
\end{aligned}
$$

∎

# E   Proofs from Section 5

## E.1   Proof of Corollary 8

The key technical result is contained in the following lemma, which says that we can lift a $\sigma$-smooth distribution on $\mathcal{B}_1^d$ to one on $\mathcal{B}_1^{d+1}$ and only lose a factor that is exponential in dimension. Because our regret guarantees are only logarithmic in $\sigma$, this will translate into a factor that is only linear in $d$ by reducing to the setting of Theorem 5. We have the following result:

**Lemma 34.** *There exist a probability kernel* $\mathcal{K} : \mathcal{B}_1^d \to \triangle(\mathcal{B}_1^{d+1})$ *that satisfies the following two properties: first,*

$$\mathbb{P}_{\tilde{x} \sim \mathcal{K}(\cdot|x)} \left( for\ all\ \tilde{w} = (w, b) \in \mathbb{R}^d \times \mathbb{R}, \ \mathrm{sign}(\langle w, x \rangle + b) = \mathrm{sign}(\langle \tilde{w}, \tilde{x} \rangle) \right) = 1,$$

*and second, if $p$ is $\sigma$-smooth with respect to $\mu_d$, then $\mathcal{K} \circ p$ is $\sigma'$-smooth with respect to $\mu_{d+1}$, where $\sigma' = \sigma/4^{d+2}$ and $\tilde{x} \sim \mathcal{K} \circ p$ if $x \sim p$ and $\tilde{x} \sim \mathcal{K}(\cdot|x)$.*

*Proof.* For general $b$, define $\tilde{w} := (w, b)$, let $\phi(x, z) = \frac{z(x,1)}{4}$, and let

$$\tilde{x} = \phi(x_t, z_t) \qquad\qquad z_t \sim \mathrm{Unif}(1, 2). \qquad\qquad (\text{E.1})$$

Note that whenever $x \in \mathcal{B}_1^d$, $\mathbb{P}[\tilde{x}_t \in \mathcal{B}_1^{d+1} \mid x_t = x] = 1$. Moreover,

$$\mathrm{sign}(\langle w, x_t \rangle + b) = \mathrm{sign}(\langle \tilde{w}, (x_t, 1) \rangle) = \mathrm{sign}(\langle \tilde{w}, z_t(x_t, 1) \rangle) = \mathrm{sign}(\langle \tilde{w}, \tilde{x}_t \rangle).$$

Since our proposed algorithm is a function only of the *version space* $\mathcal{F}_t$, and not the *disagreement region*, it follows that we can assume without loss of generality that the learner interacts with the distribution $\tilde{p}_t$ induced by drawing $x_t \sim p_t$, $z_t \sim \mathrm{Unif}[1, 2]$, and $\tilde{x}_t = \phi(x_t, z_t)$.

To conclude, we must argue that if $x_t \sim p_t$ is $\sigma$-smooth with respect to the uniform measure $\mu_d$ on $\mathcal{B}_1^d$, then $\tilde{x}_t \sim \tilde{p}_t$ is $\sigma'$-smooth with respect to the uniform measure $\mu_{d+1}$ on $\mathcal{B}_1^{d+1}$, for an appropriate $\sigma'$.

Let $\tilde{\mu}_{d+1}$ denote the density of $\tilde{x} = \phi(x, z)$, where $z \in \mathrm{Unif}[1, 2]$ when $x \sim \mu_d$. Then,

$$\frac{\mathrm{d}\tilde{p}_t(\tilde{x})}{\mathrm{d}\mu_{d+1}(\tilde{x})} = \frac{\mathrm{d}\tilde{p}_t(\tilde{x})}{\mathrm{d}\tilde{\mu}_{d+1}(\tilde{x})} \cdot \frac{\tilde{\mu}_{d+1}(\tilde{x})}{\mu_{d+1}(\tilde{x})}. \qquad\qquad (\text{E.2})$$

To bound the first term, consider $\phi^{-1}$, the inverse of $\phi$ from $\mathcal{B}_{1/4}^d \times [\frac{1}{4}, 1/2] \to \mathcal{B}_{1/4}^d \times [1, 2]$ given by

$$\phi^{-1} : \tilde{x} = (x, z) \mapsto ((x/z), 4z)$$

Then, $\tilde{p}_t$ is the pushforward under $\phi^{-1}$ of the measure $p_t \otimes \mathrm{Unif}[\frac{1}{4}, \frac{1}{2}]$, and $\tilde{\mu}_{d+1}$ the pushforward of $\mu_d \otimes \mathrm{Unif}[\frac{1}{4}, \frac{1}{2}]$. Thus, by Lemma 35, we have that $\tilde{p}_t$ is $\sigma$-smooth with respect to $\tilde{\mu}_{d+1}$.

Now, we compute $\frac{\mathrm{d}\tilde{\mu}_{d+1}(\tilde{x})}{\mathrm{d}\mu_{d+1}(\tilde{x})}$. It suffices to show that for any set $H \subset \mathcal{B}_{1/2}^d \times [\frac{1}{4}, 1/2]$, we have

$$\tilde{\mu}_{d+1}(H) \leq C \mathrm{d}\mu_{d+1}(H), \qquad\qquad (\text{E.3})$$

for some desirable constant $C$. Let $J_{\phi^{-1}}$ denote the Jacobian of the map $\phi^{-1} : (x, u) \mapsto (x/z, 4z)$. Then, $J_{\phi^{-1}}$ is a triangular matrix with determinant $4(1/z)^d$. Thus, on $H \subset \mathcal{B}_{1/2}^d \times [\frac{1}{4}, 1/2]$, its

---
**Algorithm 3** Binary Classification with Affine Thresholds
---
1: **Initialize** $\widetilde{\mathcal{W}}_1 = \mathcal{B}_1^{d+1}$, $\widetilde{w}_1 = \mathbf{e_1} \in \widetilde{\mathcal{W}}_1$,
2: **for** $t = 1, 2, \ldots$ **do**
3:     **Recieve** $x_t$, draw $z_t \sim \mathrm{Unif}(1,2)$, and **assign**

$$\widetilde{x}_t \leftarrow \phi(x_t, z_t) = \frac{z_t(x_t, 1)}{4}$$

4:
5:     **predict**

$$\hat{y}_t = \mathrm{sign}(\langle \widetilde{w}_t, \widetilde{x}_t \rangle), \hspace{3cm} \text{(\% self.classify}(x_t))$$

6:     **Update** $\widetilde{\mathcal{W}}_{t+1} = \widetilde{\mathcal{W}}_t \cap \{\widetilde{w} \in \mathcal{B}_1^{d+1} \mid \langle \widetilde{w}, \widetilde{x}_t y_t \rangle \geq 0\}$
7:     **if** $\hat{y}_t \neq y_t$ **then**                          (% self.errorUpdate$(x_t)$)
8:        $\widetilde{w}_{t+1} \leftarrow$ **JohnEllpsoidCenter**$(\widetilde{\mathcal{W}}_{t+1})$
9:        % returns center of John Ellpsoid of given convex body
---

determinant is at most $|\det_{J_\phi}(x, z)| = 4^{d+1}$. Hence,

$$\begin{aligned}
\tilde{\mu}_{d+1}(H) &= \mathbb{P}_{(x,z) \sim \mathcal{B}_1^d \times \mathrm{Unif}[1,2]}[\phi(x,z) \in H] \\
&= \mathbb{P}_{(x,z) \sim \mathcal{B}_1^d \times \mathrm{Unif}[1,2]}[(x,z) \in \phi^{-1}(H)] \\
&= \frac{\int_{\phi^{-1}(H)} \mathrm{d}x \mathrm{d}z}{\mathrm{vol}_{d+1}(\mathcal{B}_1^d \times \mathrm{Unif}[1,2])} \\
&= \frac{\int_H |\det(J_{\phi^{-1}})(x,z)| \mathrm{d}x \mathrm{d}z}{\mathrm{vol}_{d+1}(\mathcal{B}_1^d \times \mathrm{Unif}[1,2])} \\
&\leq \frac{4^{d+1} \mathrm{vol}(H)}{\mathrm{vol}_{d+1}(\mathcal{B}_1^d \times \mathrm{Unif}[1,2])} \\
&= \mu_{d+1}(H) \cdot \frac{4^{d+1} \mathrm{vol}_{d+1}(\mathcal{B}_1^{d+1})}{\mathrm{vol}_{d+1}(\mathcal{B}_1^d \times \mathrm{Unif}[1,2])} \\
&= \mu_{d+1}(H) \cdot \frac{4^{d+1} \mathrm{vol}_{d+1}(\mathcal{B}_1^{d+1})}{\mathrm{vol}_d(\mathcal{B}_1^d)} \\
&= 4^{d+1} \frac{\sqrt{\pi}}{d + \frac{1}{2}} \leq 4^{d+2}.
\end{aligned}$$

Combining these computations with (E.2) yields

$$\frac{\mathrm{d}\tilde{p}_t(\tilde{x})}{\mathrm{d}\mu_{d+1}(\tilde{x})} = \frac{\mathrm{d}\tilde{p}_t(\tilde{x})}{\mathrm{d}\tilde{\mu}_{d+1}(\tilde{x})} \cdot \frac{\tilde{\mu}_{d+1}(\tilde{x})}{\mu_{d+1}(\tilde{x})} \leq \sigma^{-1} 4^{d+2},$$

which concludes the proof. ∎

**Lemma 35.** *Suppose that $f : \mathcal{X} \to \mathcal{Y}$ is a measurable map and suppose that $p, \mu$ are measures on $\mathcal{X}$ and $p$ is $\sigma$-smooth with respect to $\mu$. Define the pushforward measure on $\mathcal{Y}$ by taking $f_\star \mu(B) = \mu(f^{-1}(B))$ for any measurable $B \subset \mathcal{Y}$. Then $f_\star p$ is $\sigma$-smooth with respect to $f_\star \mu$.*

*Proof.* Let $B \subset \mathcal{Y}$ be measurable. Then

$$f_\star p(B) = p(f^{-1}(B)) \leq \frac{\mu(f^{-1}(B))}{\sigma} \leq \frac{f_\star \mu(B)}{\sigma}. \hspace{2cm} \text{(E.4)}$$

As this holds for any $B \subset \mathcal{Y}$, the result follows. ∎

We now describe Algorithm 3. At each time $t$, we draw $z_t \sim \mathrm{Unif}(1,2)$ independently and form $\widetilde{x}_t = \phi(x_t, z_t)$, where $\phi$ is as in (E.1). We then run the classify and update subroutines of Algorithm 1 at each time step on the new data sequence $(\widetilde{x}_t, y_t)$. We are now ready to prove Corollary 8:

**Algorithm 4** Binary Classification with Nonlinear Features

---

1: **Initialize** $\widetilde{\mathcal{W}}_1 = \mathcal{B}_1^{d+1}$, $\widetilde{w}_1 = \mathbf{e_1} \in \mathcal{W}_1$, $\phi : \mathcal{B}_1^d \to \mathcal{B}_1^d$
2: **for** $t = 1, 2, \ldots$ **do**
3:     **Recieve** $x_t$, **predict**

$$\hat{y}_t = \text{sign}(\langle \widetilde{w}_t, \phi(x_t) \rangle), \hspace{3cm} \text{(\% self.classify}(x_t)\text{)}$$

4:     **Update** $\widetilde{\mathcal{W}}_{t+1} = \widetilde{\mathcal{W}}_t \cap \{\widetilde{w} \in \mathcal{B}_1^{d+1} | \langle \widetilde{w}, \phi(x_t) y_t \rangle \geq 0\}$
5:     **if** $\hat{y}_t \neq y_t$ **then**                                    (**% self.errorUpdate**$(x_t)$)
6:         $\widetilde{w}_{t+1} \leftarrow$ **JohnEllpsoidCenter**$(\widetilde{\mathcal{W}}_{t+1})$
7:         `% returns center of John Ellpsoid of given convex body`

---

*Proof of Corollary 8.* We use Algorithm 3 to reduce the problem to the situation of Theorem 5. Indeed, by Lemma 34, the data sequence $(\widetilde{x}_t, y_t)$ satisfies the property that $\widetilde{x}_t$ is $\left(4^{-d-2}\sigma\right)$-smooth with respect to $\mu_{d+1}$ and is realizable by the function class $\mathcal{F}_{\text{lin}}^{d+1}$. The result then follows immediately from Theorem 5. ∎

### E.2 Proof of Theorem 9

We now prove begin generalizing beyond linear function classes with Theorem 9. The key technical result shows that if $\phi : \mathcal{B}_1^d \to \mathcal{B}_1^d$ is well-behaved, then $\phi_\star \mu_d$ is $\sigma$-smooth with respect to $\mu_d$, which will then allow us to apply Theorem 5.

**Lemma 36.** *Suppose that $p$ is a measure on $\mathcal{B}_1^d$ that is $\sigma$-smooth with respect to $\mu_d$ and suppose that $\phi : \mathcal{B}_1^d \to \mathcal{B}_1^d$ is a function satisfying the following two properties:*

- *There is some $c > 0$ such that $|\det(D\phi(x))| > c$ for all $x \in \mathcal{B}_1^d$.*

- *There is some $N \in \mathbb{N}$ such that for every $x \in \mathcal{B}_1^d$, it holds that $\left|\phi^{-1}(x)\right| \leq N$, where $\phi^{-1}(x) = \left\{y \in \mathcal{B}_1^d | \phi(y) = x\right\}$.*

*Then, $\phi_\star p$ is $\left(\frac{c}{N}\sigma\right)$-smooth with respect to $\mu_d$.*

*Proof.* By Lemma 35, we have that $\phi_\star p$ is $\sigma$-smooth with respect to $\phi_\star \mu_d$. Thus, as

$$\frac{\mathrm{d}\phi_\star p}{\mathrm{d}\mu_d} = \frac{\mathrm{d}\phi_\star p}{\mathrm{d}\phi_\star \mu_d} \cdot \frac{\mathrm{d}\phi_\star \mu_d}{\mu d}$$

it suffices to bound the latter factor. By the area formula [Federer, 2014], we have for any $B \subset \mathcal{B}_1^d$ that

$$\begin{aligned}
\phi_\star \mu_d(B) &= \int_{\phi^{-1}(B)} \mathrm{d}\mu_d(x) \\
&= \int_{\phi^{-1}(B)} \frac{|\det(D\phi(x))|}{|\det(D\phi(x))|} \mathrm{d}\mu_d(x) \\
&\leq \frac{1}{c} \int_{\phi^{-1}(B)} |\det(D\phi(x))| \, \mathrm{d}\mu_d(x) \\
&= \frac{1}{c} \int_B \left|\phi^{-1}(y)\right| \mathrm{d}\mu_d(y) \leq \frac{N}{c}\mu_d(y)
\end{aligned}$$

Thus we see that for any $B$,

$$\frac{\phi_\star \mu_d(B)}{\mu_d(B)} \leq \frac{N}{c}$$

and so the result follows. ∎

As a corollary, we generalize Theorem 5 to adversaries that are now realizable to a class linear in some new set of features:

**Corollary 37.** *Let $\phi$ be a map as in Lemma 36 and suppose that we are in the smoothed online learning setting with an adversary realizable with respecto to $\mathcal{F} \circ \phi = \{x \mapsto f(\phi(x)) | f \in \mathcal{F}\}$. If we run Algorithm 4 on the data $(\phi(x_t), y_t)$, then for all $T$, with probability at least $1 - \delta$, it holds that*

$$\text{Reg}_T \le 136 d \log(d) + 34 \log\left(\frac{N}{c}\right) + 34 \log\left(\frac{T}{\sigma\delta}\right) + 56$$

*Proof.* The statement follows immediately from applying Theorem 5 to the data sequence $(\phi(x_t), y_t)$ and using Lemma 36 to bound the smoothness. ∎

Finally, we prove the simpler result stated in Section 5:

*Proof of Theorem 9.* By Corollary 37, it suffices to bound $N$ and $c$ in Lemma 36. Suppose that $\phi(x) = (\phi_1(x_1), \ldots, \phi_d(x_d))$ as in the statement of the result. Then we see that $\mathrm{D}\phi(x)$ is diagonal with $\phi_i'(x_i)$ as the $i^{th}$ element of the diagonal and thus

$$\det(\mathrm{D}\phi(x)) = \prod_{i=1}^{d} |\phi_i'(x_i)| \ge \alpha^d$$

where the final inequality follows from the assumption. Note that if $\phi_i' > 0$ for all $i$, then $\phi$ is strictly increasing coordinate wise and thus we may take $N = 1$. The result follows. ∎

### E.3 Proof of Theorem 10

In this section, we show that our techniques extend to polynomial decision boundaries. Morally, we proceed on similar lines as to the proof of Theorem 9 outlined in the previous section, but there are a number of new technical subtleties that appear in this analysis that were not present before. The most salient difference between the maps considered above and that which is required for a polynomial decision boundary is that polynomial features require imbedding our problem into a higher dimensional space; while Lemma 35 ensures that the pushforward of the law of each $x_t$ is smooth with respect to the pushforward of $\mu_d$, our analysis is very specific to the dominating measure being uniform on the ball, which can never happen if we are pushing $\mu_d$ forward into a higher dimensional space. In order to resolve this difficulty, we will present a reduction that allows us to combine multiple points into one 'meta-point,' whose law will be smooth with respect to the uniform measure on the higher dimensional ball. We will then be able to reduce to a similar setting as considered in Theorem 5 and deduce a similar regret bound. We prove the following result:

**Proposition 38.** *Suppose that $\phi : \mathcal{B}_1^d \to \mathcal{B}_1^m$ satisfies the following properties:*

- *$\phi$ is L-Lipschitz.*

- *$\phi$ is polynomial in the sense that each of the coordinates of $\phi$ is a polynomial in the coordinates of $x \in \mathcal{B}_1^d$ with degree at most $\ell$.*

- *There is an $\alpha > 0$ such that the Jacobian $\mathrm{D}\phi$ satisfies:*

$$\det\left(\mathbb{E}_{x \sim \mu_d}\left[\mathrm{D}\phi(x)\mathrm{D}\phi(x)^T\right]\right) \ge \alpha^2$$

*Suppose further that the $x_t \in \mathcal{B}_1^d$ are generated in a $\sigma$-smooth manner and the $y_t$ are realizable with respect to $\mathcal{F}_{\text{lin}}^m \circ \phi$. Then there is a universal constant $C$ such that for all $T$, if we set*

$$p = Cm\ell \log\left(\frac{L\ell T}{\delta}\right)$$

*and run Algorithm 5, then with probability at least $1 - \delta$,*

$$\text{Reg}_T \le C\left(m \log(m) + \log\left(\frac{1}{\alpha}\right) + \ell^2 m^2 d \log^2\left(\frac{d\ell T L}{\sigma\delta}\right)\right).$$

*Proof.* Consider the sequence of stopping times $\rho_\tau$, where $\rho_0 = 0$ and, for $\tau > 0$,

$$\rho_\tau = \inf\left\{ t > \rho_{\tau-1} \,\middle|\, \max\left( \sum_{s=\rho_{\tau-1}}^{t} \mathbb{I}[y_t = 1 \text{ and } \hat{y}_t = -1], \sum_{s=\rho_{\tau-1}}^{t} \mathbb{I}[y_t = -1 \text{ and } \hat{y}_t = 1] \right) \geq p \right\}.$$

for some $p$ to be determined. Furthermore, let

$$\ell_t = \mathbb{I}\left[t \in (2p+1)\mathbb{N} \text{ and } t - 2p \leq \rho_\tau \leq t \text{ for some } \tau\right]$$

We begin by claiming that the following inequality holds:

$$\text{Reg}_T \leq (2p+1)\left(1 + \sum_{t=1}^{T} \ell_t\right) \leq (2p+1)\left(1 + \sum_{t'=1}^{\left\lfloor \frac{T}{2p+1} \right\rfloor} \ell_{(2p+1)t'}\right). \tag{E.5}$$

Indeed, we note that the sum is equal to $\tau_T$, the maximal $\tau$ such that $\rho_\tau \leq T$ and the pigeonhole principle tells us that we suffer at most $2p+1$ mistakes in the interval $\rho_{\tau-1} \leq t \leq \rho_\tau$. There are at most $2p$ mistakes in the interval $\rho_{\tau_T} \leq t \leq T$ and so the first inequality holds. The second inequality follows from noting that $\ell_t = 1$ implies that $t = (2p+1)t'$ for some $t'$. For each $1 \leq \tau \leq \tau_T$, we let

$$\overline{x}_\tau = \frac{1}{p} \sum_{\substack{\rho_{\tau-1} < t \leq \rho_\tau \\ y_t = y_\tau \text{ and } \hat{y}_t \neq y_t}} \phi(x_t)$$

Now, fix $\beta, \gamma > 0$ to be set later and let

$$\mathcal{U}_1 = \{\text{for all } t \text{ such that } \mathbb{P}_t(\hat{y}_t \neq y_t) < \gamma, \text{ it holds that } \hat{y}_t = y_t\}$$
$$\mathcal{U}_2 = \{\text{for all } t \text{ such that } \mathbb{P}_t(y_t = y) < \beta \text{ for some } y, \text{ it holds that } y_t \neq y\}.$$

We claim that for some $p$, there is a sequence of $\overline{x}'_\tau \in \mathcal{B}_1^m$ such that if

$$\mathcal{U} = \{\widetilde{x}_\tau = \widetilde{x}'_\tau \text{ for all } \tau\}$$
$$\mathcal{U} = \mathcal{U}_1 \cap \mathcal{U}_2 \cap \mathcal{U}_3$$

then first, $\mathbb{P}(\mathcal{U}) \geq 1 - 2T\beta - T\gamma - \frac{\delta}{4}$ and second, the $\overline{x}'_\tau$ are $\sigma'$-smooth with respect to $\mu_m$. This claim, and the dependence of $\sigma'$ on the relevant parameters is the subject of Proposition 39 below. For now, we will take it as given. Now, recalling the disagreement region and version space notation $D_t, \mathcal{F}_t$, as from Appendix D, we note that if $\ell_{t'} = 1$ and $\tau$ is maximal subject to $\rho_\tau \leq (2p+1)t'$, then we must have $\overline{x}_\tau \in D_{\rho_{\tau-1}}$. To see this, note that $w_t = w_{\rho_{\tau-1}}$ for $\rho_{\tau-1} < t \leq \rho_\tau$ and thus $w_{\rho_{\tau-1}}$ is such that $\langle w_{\rho_{\tau-1}}, y_s\phi(x_s)\rangle < 0$ for each such $s$. By linearity, we have

$$\left\langle w_{\rho_{\tau-1}}, \overline{y}_\tau \overline{x}_\tau \right\rangle = \frac{1}{p} \sum_{i=1}^{p} \left\langle w_{\rho_{\tau-1}}, y_{\tau_i}\phi(x_{\tau_i})\right\rangle < 0$$

Realizabilty implies that there is some $w$ such that $\langle w, y_s\phi(x_s)\rangle \geq 0$ for all $s$, and so linearity implies that, for that $w$,

$$\langle w, \overline{y}_\tau \overline{x}_\tau \rangle = \frac{1}{p} \sum_{i=1}^{p} \langle w, \overline{y}_\tau \overline{x}_\tau \rangle \geq 0$$

Thus we see that $\overline{x}_\tau \in D_{\rho_{\tau-1}}$. We now note that for any $t'$,

$$\mathbb{P}_{\rho_{\tau-1}}(\ell_{(2p+1)t'} = 1) \leq \frac{(2p+1)\mu_m(D_{\rho_{\tau-1}})}{\sigma'}$$

where $\sigma'$ is as in Proposition 39. Applying now Lemmas 4, 30 and 31 in the same way as in the proof of Theorem 5, we place ourselves now into the situation of Lemma 27, the generalized version of the master reduction Lemma 3. Thus, we have that for all $T$, with probability at least $1 - \delta$,

$$\sum_{t'=1}^{\left\lfloor \frac{T}{2p+1} \right\rfloor} \ell_{(2p+1)t'} \leq 4 \frac{\log\left(\frac{2T(2p+1)4^{m+1}m^{2m}}{(2p+1)\sigma'\delta}\right)}{\log\left(\frac{9}{8}\right)} + \frac{e-1}{1 - \sqrt{\frac{8}{9}}}$$

$$\leq C\left(1 + m\log(m) + \log\left(\frac{T}{\delta}\right) + \log\left(\frac{1}{\sigma'}\right)\right)$$

---

**Algorithm 5** Binary Classification with Polynomial Features

---

1: **Initialize** $\widetilde{\mathcal{W}}_1 = \mathcal{B}_1^m$, $\widetilde{w}_1 = \mathbf{e_1} \in \mathcal{W}_1$, $\phi : \mathcal{B}_1^d \to \mathcal{B}_1^m$, $\mathcal{M}_1, \mathcal{M}_{-1} = \{\}$, $p \in \mathbb{N}$
2: **for** $t = 1, 2, \ldots$ **do**
3:     **Recieve** $x_t$, **predict**

$$\hat{y}_t = \text{sign}(\langle \widetilde{w}_t, \phi(x_t) \rangle), \qquad\qquad (\text{\% self.classify}(x_t))$$

4:     **Update** $\widetilde{\mathcal{W}}_{t+1} = \widetilde{\mathcal{W}}_t \cap \{\widetilde{w} \in \mathcal{B}_1^{d+1} | \langle \widetilde{w}, \phi(x_t)y_t \rangle \geq 0\}$
5:     **if** $\hat{y}_t \neq y_t$ **then**                                      (**% self.errorUpdate**($x_t$))
6:         **Update** $\mathcal{M}_{y_t} \leftarrow \mathcal{M}_{y_t} \cup \{x_t\}$
7:     **if** $\max(|\mathcal{M}_1|, |\mathcal{M}_{-1}|) = p$ **then**
8:         **Update**
9:         $\widetilde{w}_{t+1} \leftarrow$ **JohnEllpsoidCenter**($\widetilde{\mathcal{W}}_{t+1}$)
10:          **% returns center of John Ellpsoid of given convex body**
11:         **Reset** $\mathcal{M}_1, \mathcal{M}_{-1} \leftarrow \{\}$

---

If we set

$$p = Cm\ell \log\left(\frac{L\ell T}{\delta}\right)$$

then plugging in the penultimate display into (E.5), taking $\gamma = \frac{\delta}{4T}$ and $\beta = \frac{\delta}{8T}$, and plugging in the bounds from Proposition 39 concludes the proof.

$\blacksquare$

We note that Theorem 10 follows immediately from Proposition 38. The key difficulty in the proof of Proposition 38, that we left until now, is the smoothness of the $\overline{x}_\tau$. We state this fact, and provide a quantitative bound on the smoothness parameter, in the next proposition:

**Proposition 39.** *Let $\phi : \mathcal{B}_1^d \to \mathcal{B}_1^m$ be an L-Lipschitz function whose coordinates are polynomials in the coordinates of $x \in \mathcal{B}_1^d$, with degree at most $\ell$. Suppose $\phi$ is such that*

$$\det\left(\mathbb{E}_{x \sim \mu_d}\left[D\phi(x)D\phi(x)^T\right]\right) \geq \alpha^2.$$

*Fix any $p \in \mathbb{N}$ such that $pd \geq m$. Suppose that $(x_1, y_1), \ldots, (x_T, y_T)$ is a data sequence satisfying the following four properties:*

- *The distribution of $x_t$ conditional on the history is $\sigma$-smooth with respect to $\mu_d$.*

- *For $y \in \{\pm 1\}$, for any $t$, $\mathbb{P}_t(y_t = y) \geq \beta$.*

- *The $y_t$ are realizable with respect to $\mathcal{F}_{\text{lin}}^m \circ \phi$.*

- *For any $t$, and any choice of $\hat{y}_t$ by the learner, $\mathbb{P}_t(\hat{y}_t \neq y_t) \geq \gamma$.*

*where $\mathbb{P}_t$ is the conditional probability of the history up to time $t$. Now, consider the set of stopping times $\rho_\tau$ with $\rho_0 = 0$ and*

$$\rho_\tau = \inf\left\{t > \rho_{\tau-1} | \max\left(\sum_{s=\rho_{\tau-1}}^{t} \mathbb{I}[y_t = 1 \text{ and } \hat{y}_t = -1], \sum_{s=\rho_{\tau-1}}^{t} \mathbb{I}[y_t = -1 \text{ and } \hat{y}_t = 1]\right) \geq p\right\}$$

*Let $\overline{y}_\tau = y_{\rho_\tau}$ and let*

$$\overline{x}_\tau = \frac{1}{p}\sum_{k=1}^{p} \phi(x_{\tau_k})$$

*where $\tau_1, \ldots, \tau_p$ are the $p$ times $\rho_{\tau-1} < t \leq \rho_\tau$ satifying $y_t = \overline{y}_\tau$ and $\hat{y}_t \neq y_t$. There is a universal constant $C$ such that if*

$$p \geq Cm\ell \log\left(\frac{L\ell T}{\delta}\right)$$

then there is a data sequence a sequence $\overline{x}'_t \in \mathcal{B}_1^m$ satisfying the following three properties. First, the sequence $(\overline{x}'_t, \overline{y}_t)$ is realizable with respect to $\mathcal{F}_{\text{lin}}^m$. Second, with probability at least $1 - \delta$, for all $t$, $\overline{x}_t = \overline{x}'_t$. Third, the $\overline{x}'_t$ is $\sigma'$-smooth with respect to $\mu_m$, where

$$\sigma' = c \cdot \alpha \cdot p^{-\frac{m}{2}} \left(\frac{\beta\gamma\sigma}{T}\right)^{2\ell mp} \ell^{-m(\ell+pd)} d^{-pd} \tag{E.6}$$

*wisth $c$ a universal constant*

Intuitively, we wait until we have misclassified a class $p$ times and then form a 'meta-point' $(\widetilde{x}_\tau, \widetilde{y}_\tau)$ that will allow us to reduce to the setting of Theorem 5. The meta-point will be constructed by averaging samples $x_t$ in order to ensure smoothness with respect to $\mu_m$.

We will show that Proposition 39 follows from three results that we will prove below. First, we show that if $\phi$ is well-behaved, and the $x_t$ are $\sigma$-smooth with respect to $\mu_d$, then the $\widetilde{x}_t$ are $\sigma'$-smooth with respect to $\mu_m$.

**Proposition 40.** *Suppose that $\phi : \mathcal{B}_1^d \to \mathcal{B}_1^m$ is a smooth map between Euclidean balls of dimensions $d$ and $m$ with Jacobian $D\phi$. Consider the function $f : (\mathbb{R}^d)^p \to \mathbb{R}^m$ defined as*

$$\psi(x_1, \ldots, x_p) = \frac{1}{p}\sum_{i=1}^{p} \phi(x_i) \tag{E.7}$$

*Suppose that that the following three conditions are satisfied:*

- *There is some $V \subset (\mathcal{B}_1^d)^{\times p}$ and $c > 0$ such that for $\mu_d^{\otimes p}$-almost every $x \in V$, $\det(D\psi(x)D\psi(x)^T) \geq \alpha^2$.*

- *For some $\ell \geq 2$*

$$\sup_{x \in \mathcal{B}_1^d} \max_{\substack{|\nu|=\ell+1 \\ 1 \leq i \leq m}} |\partial^\nu \phi_i(x)| \leq 2^{-(1+\ell)} \tag{E.8}$$

  *In particular, this holds if $\phi(x)$ is a polynomial of degree at most $\ell$.*

- *Finally, suppose that the joint distribution of $(x_1, \ldots, x_p)$ is $\sigma^p$-smooth with respect to $\mu_d^{\otimes p}$.*

*If $pd \geq m$, then the law of $\psi(x_1, \ldots, x_p)$, conditioned on $(x_1, \ldots, x_p) \in V$ is $\sigma'$-smooth with respect to $\mu_m$, the uniform measure on $\mathcal{B}_1^m$, where*

$$\sigma' = \frac{\alpha\sigma^p \cdot \mathbb{P}\left((x_1, \ldots, x_p) \in V\right)}{\ell^{2m+mpd} d^{pd}} \tag{E.9}$$

Second, we show that if $\phi$ is a polynomial, then it is well-behaved in the sense of Proposition 40 with high probability, by proving the more general small-ball type estimate below:

**Proposition 41.** *There exists a univesal constant $C$ such that the following holds. Let $\Psi : \mathcal{R}^d \to \mathbb{S}_+^D$ be any function whose image is contained in the set of PSD matrices and whose entries are polynomials of degree at most $\ell$, and let $x_1, \ldots, x_p$ be a sequence of random-variables such that, for each $t$, $x_t \mid x_1, \ldots, x_{t-1}$ is $\sigma$-smooth with respect to a common log-concave measure $\mu$, and $\mathbb{P}_{x \sim \mu}[\lambda_{\max}(\Psi(x)) \leq B] = 1$. Define*

$$\Lambda := \mathbb{E}_{x \sim \mu}[\Psi(x)]$$

*Suppose that $p \geq 16\log(1/\delta) + \frac{D}{4}\log(24B) + \frac{1}{4}(\log\det(\Lambda) + D\ell\log(C\ell))$. Then,*

$$\mathbb{P}\left[\det\left(\frac{1}{p}\sum_{i=1}^{p} \Psi(x_i)\right) \leq \left(\frac{\sigma}{C\ell}\right)^{\ell D}\det(\Lambda)\right] \leq \delta.$$

Finally, we will show that if the probabilities corresponding to each label are well-controlled, then the laws of $x_{\tau_k}$ are smooth with respect to $\mu_d$:

**Proposition 42.** *Let $(x_{\tau_1}, y_{\tau_1}), \ldots, (x_{\tau_p}, y_{\tau_p})$ be the sequence of points defined in Proposition 39, arising from a sequence of $(x_t, yt)$ with the $x_t$ being $\sigma$-smooth conditional on the history and for each $t$, and $y \in \{\pm 1\}$ it holds that $\mathbb{P}(y_t = y) \geq \beta$ and that $\mathbb{P}(\hat{y}_t \neq y_t) \geq \gamma$. Then, for each $i$, it holds that the law of $x_{\tau_i}$ conditional on the history up to time $\tau_{i-1}$ is $(\beta\gamma\sigma/T)$-smooth with respect to $\mu_d$. In particular, the law of $(x_{\tau_1}, \ldots, x_{\tau_p})$, conditional on the sigma-algebra generated by $\rho_{\tau-1}$, is $(\beta\gamma\sigma/T)^p$-smooth with respect to $\mu_d^{\otimes p}$.*

*Proof.* Note that a peeling argument and induction show that the second statement follows immediately from the first. For any $\tau, i$, denote probability conditioned on the history up to time $\tau_{i-1}$ by $\mathbb{P}_{\tau_{i-1}}$. Let $B \subset \mathcal{B}_1^d$ be measurable. Then we compute that $\mathbb{P}_{\tau_{i-1}}(x_{\tau_i} \in B)$ can be given by:

$$\sum_{1 \le t \le T} \mathbb{P}_{\tau_{i-1}}(t = \tau_i)\mathbb{P}_{\tau_{i-1}}(x_t \in B | \tau_i = t)$$

$$\le \sum_{1 \le t \le T} \mathbb{P}_{\tau_{i-1}}(\tau_i > t - 1)\mathbb{P}_{\tau_{i-1}}(x_t \in B | y_t \ne \hat{y}_t \text{ and } y_t = \widetilde{y}_\tau)$$

$$\le \sum_{1 \le t \le T} \mathbb{P}_{\tau_{i-1}}(\tau_i > t - 1)\frac{\mathbb{P}_{\tau_{i-1}}(x_t \in B | y_t = \widetilde{y}_\tau)}{\gamma}$$

$$\le \sum_{1 \le t \le T} \mathbb{P}_{\tau_{i-1}}(\tau_i > t - 1)\frac{\mathbb{P}_{\tau_{i-1}}(x_t \in B | y_t = 1, \widetilde{y}_\tau = 1)\mathbb{P}_{\tau_{i-1}}(\widetilde{y}_\tau = 1) + \mathbb{P}_{\tau_{i-1}}(x_t \in B | y_t = -1, \widetilde{y}_\tau = -1)\mathbb{P}_{\tau_{i-1}}(\widetilde{y}_\tau = 1)}{\gamma}$$

$$= \sum_{1 \le t \le T} \mathbb{P}_{\tau_{i-1}}(\tau_i > t - 1)\frac{\mathbb{P}_{\tau_{i-1}}(x_t \in B)}{\beta\gamma}$$

$$\le \frac{T\mu_d(B)}{\beta\gamma\sigma}$$

Thus, the result follows. ∎

Propositions 40 and 41 will be shown below but for now we will take them as given. We can now prove the key proposition:

*Proof of Proposition 39.* Let

$$\mathcal{E} = \left\{ \det\left(\frac{1}{p}\sum_{i=1}^p (\mathrm{D}\phi\mathrm{D}\phi^T)(x_{\tau_i})\right) > \left(\frac{\beta\gamma\sigma}{T}\right)^{(2p+1)\ell m}(C\ell)^{-\ell m}\alpha^2 \text{ for all } \tau \right\}$$

and note that by the fact that $\tau \le T$

$$p \ge Cm\log\left(\frac{\ell L\alpha T}{\delta}\right),$$

applying a union bound to Proposition 41 and using Proposition 42 to ensure that the hypothesis holds, shows that $\mathbb{P}(\mathcal{U}) \ge 1 - \delta$. On $\mathcal{U}$ we will let $\overline{x}'_\tau = \overline{x}_\tau$ and on $\mathcal{U}^c$, we will draw $\overline{x}'_\tau$ from $\mu_m$, conditioned on $(\overline{x}'_\tau, \overline{y}_\tau)$ being realizable with respect to $\mathcal{F}_{\text{lin}}^m$. Note that we have realizability by construction on $\mathcal{U}^c$. On $\mathcal{U}$, we have that $\overline{x}'_\tau = \overline{x}_\tau$ and note that convexity implies that if $y_{\tau_1} = \cdots = y_{\tau_p}$, then any realizable adversary must classify $\overline{x}_\tau$ as $\overline{y}_\tau$. Indeed, if $w \in \mathcal{F}_{\text{lin}}^m$ is in the version space, and $\overline{y}_\tau = 1$, then

$$\left\langle w, \frac{1}{p}\sum_{i=1}^p \phi(x_{\tau_i})\right\rangle = \frac{1}{p}\sum_{i=1}^p \langle w, \phi(x_{\tau_i})\rangle > 0$$

and similarly if $\overline{y}_\tau = -1$. Thus, realizability holds. As we have already seen that

$$\{\text{there exists } \tau \text{ such that } \widetilde{x}'_\tau \ne \overline{x}_\tau\} \subset \mathcal{E}^c$$

and $\mathbb{P}(\mathcal{U}^c) \le \delta$, it suffices to show smoothness of $\overline{x}'_\tau$. On $\mathcal{U}^c$, the construction implies that $\overline{x}'_\tau$ are smooth, so we now restrict to the event $\mathcal{E}$. We first compute the Jacobian of $\psi$:

$$\mathrm{D}\psi(x_1, \ldots, x_p) = \frac{1}{p}\begin{bmatrix} \mathrm{D}\phi(x_1) & \mathrm{D}\phi(x_2) & \cdots & \mathrm{D}\phi(x_p)\end{bmatrix}$$

and thus

$$\mathrm{D}\psi\mathrm{D}\psi^T = \frac{1}{p^2}\sum_{i=1}^p D\psi(x_i)D\psi(x_i)^T$$

which in turn implies:

$$\det(\mathrm{D}\psi\mathrm{D}\psi^T) = p^{-m}\det\left(\frac{1}{p}\sum_{i=1}^p D\psi(x_i)D\psi(x_i)^T\right).$$

Thus, under $\mathcal{U}$, we have that

$$\det((\mathrm{D}\psi\mathrm{D}\psi^T)(x_{\tau_1},\ldots,x_{\tau_p})) \geq p^{-m}\left(\frac{\beta\gamma\sigma}{T}\right)^{(2p+1)\ell m}(C\ell)^{-\ell m}\alpha^2 = \widetilde{\alpha}^2$$

We may now use Proposition 42 to get that $(x_{\tau_1},\ldots,x_{\tau_p})$ has a law that is $(\beta\gamma\sigma/T)^p$ smooth with respect to $\mu_d^{\otimes p}$ and apply Proposition 40 to get that, conditional on $\mathcal{E}$, the law of $\psi(x_{\tau_1},\ldots,x_{\tau_p})$ is $\sigma'$-smooth with respect to $\mu_m$, where

$$\sigma' = \frac{\alpha'\left(\frac{\beta\gamma\sigma}{2T}\right)^p}{\ell^{2m+mpd}d^{pd}}$$

where we let $V = \mathcal{U}$ and note that $\mathbb{P}((x_{\tau_1},\ldots,x_{\tau_p}) \in \mathcal{U}) \geq \frac{1}{2}$. The result follows. ∎

### E.3.1 Proof of Propossition 40

We will proceed by using the co-area formula [Federer, 2014]. For a given $x \in (B_1^d)^n$, let

$$J(\psi)(x) = \sqrt{\det(\mathrm{D}\psi(x)\mathrm{D}\psi(x)^T)} \tag{E.10}$$

and let $\mathcal{H}^j$ denote the $j$-dimensional Hausdorff measure. Then the co-area formula tells us that for any $B \subset V$, we have

$$\int_{\psi^{-1}(B)} J(\psi)(x)\mathrm{d}\mathcal{H}^{dp}(x) = \int_B \mathcal{H}^{dp-m}(\psi^{-1}(y))\mathrm{d}\mathcal{H}^m(y) \tag{E.11}$$

We make use of the following lemma:

**Lemma 43.** *Suppose that $\phi$ is as in Proposition 40. Then for all $y$,*
$$\mathcal{H}^{dp-m}(\psi^{-1}(y)) \leq \ell^{2m+mpd}$$

Before proving the lemma, we require a preliminary result from Yomdin [1984]; we reprove it here in order to keep track of the constant.

**Lemma 44.** *Fix $k \in \mathbb{N}$ and suppose that $Y \subset \mathcal{B}_1^k$ is a hypersurface and let $\mathcal{B}_r^k \subset \mathcal{B}_1^k$. Suppose that any line passing through $\mathcal{B}_r^d$ intersects $Y$ in at most $\ell$ points. Then,*

$$\mathrm{vol}_{k-1}(Y) \leq \frac{\ell 2\pi^{\frac{k}{2}}}{\Lambda\left(\frac{k}{2}\right)}r^{-k}. \tag{E.12}$$

*Proof.* Because $\mathcal{B}_r^k$ is convex, we may consider the map $\pi : Y \to \partial\mathcal{B}_r$ of projection to the boundary. Recentering so that $\mathcal{B}_r^k$ has center at the origin, we have $\pi(y) = r\frac{y}{||y||}$. By the co-area formula introduced as (E.11), we have

$$\int_Y J(\pi)(y)\mathrm{d}\mathcal{H}^{k-1}(y) = \int_{\partial\mathcal{B}_r^k}\mathcal{H}^0(\pi^{-1}(z))\mathrm{d}\mathcal{H}^{k-1}(z) \tag{E.13}$$

Now, note that $J(\pi(y)) \geq r^k$ and thus we have

$$\mathrm{vol}_{k-1}(Y) = \int_Y \mathrm{d}\mathcal{H}^{k-1}(y) \leq \frac{\int_{\partial\mathcal{B}_r^k}\mathcal{H}^0(\pi^{-1}(z))\mathrm{d}\mathcal{H}^{k-1}(z)}{r^k} \leq \frac{\ell 2\pi^{\frac{k}{2}}}{\Lambda\left(\frac{k}{2}\right)r^k} \tag{E.14}$$

where the last inequality comes from combining the fact that by assumption $\mathrm{vol}_0(\psi^{-1}(z)) \leq \ell$ and the expression for the surface area of $S^{d-1}$. ∎

*Proof of Lemma 43.* We apply Yomdin [1984, Theorem 3 (iii)] iteratively on the coordinates of $\psi$. In particular, we apply Lemma 44 in order to keep track of the explicit constant in Yomdin [1984, Lemma 7] and apply Yomdin [1984, Lemma 4] to show that we may choose $r = (20 \cdot \ell^2)^{-1}$ in Lemma 44. Thus we have for any $y \in \mathcal{B}_1^m$,

$$\mathrm{vol}_{pd-m}(f^{-1}(y)) \leq \prod_{j=0}^{m-1}\left(\frac{\ell 2\pi^{\frac{pd-j}{2}}}{\Lambda\left(\frac{pd-j}{2}\right)}(20 \cdot \ell^2)^{pd-j}\right) \tag{E.15}$$

$$\leq \ell^{2m+mpd} \tag{E.16}$$

using the fact that $pd \geq m$. The result follows. ∎

Returning now to the proof of Proposition 40, we see for a given set $B \subset V$, that

$$\mathbb{P}\left(\psi(x_1, \ldots, x_p) \in B \mid (x_1, \ldots, x_p) \in V\right) \leq \frac{\sigma^{-p} \operatorname{vol}_{dp}(\psi^{-1}(B))}{\omega_d^p \mathbb{P}(V)} \tag{E.17}$$

$$\leq \frac{\sigma^{-p} \ell^{2m+mpd}}{\omega_d^p \mathbb{P}(V)} \operatorname{vol}_m(B) \tag{E.18}$$

$$\leq \frac{\sigma^{-p} \ell^{2m+mpd} \omega_m}{\omega_p^d \mathbb{P}(V)} \mu_m(B) \tag{E.19}$$

where the first inequality follows from the definition of smoothness, the second inequality follows by (E.11) and the above claims, and the last inequality follows by the definition of $\mu_m$. The result follows by using the fact that

$$\omega_m = \frac{\pi^{\frac{m}{2}}}{\Lambda\left(\frac{m}{2}+1\right)} \tag{E.20}$$

and bounding $\left(\frac{m}{4}\right)^{\frac{m}{4}} \leq \Lambda\left(\frac{m}{2}+1\right) \leq \left(\frac{m}{2}\right)^{\frac{m}{2}}$.

### E.3.2 Proof of Proposition 41

We first introduce a small-ball estimate for sums of PSD random variables, in the spirit of Simchowitz et al. [2018].

**Lemma 45.** *Let $X_1, X_2, \ldots, X_p$ be i.i.d. of positive semi-definite, $\mathbb{R}^{D \times D}$-valued random variables, and suppose there exists $B, \eta > 0$ and $\Lambda \in \mathbb{S}_{++}^D$ for which, for all $t \in [p]$,*

$$\mathbb{P}[\lambda_{\max}(X_t) > B] = 0$$

$$\mathbb{P}_{X_t}[v^\top X v \geq v^\top \Lambda v \mid X_1, \ldots, X_{t-1}] \geq \eta, \quad \forall v \in \mathbb{R}^D.$$

*Then, if $p \geq 8\eta^{-1} \log(1/\delta) + \frac{D}{4} \log(\frac{12B}{\eta}) + \frac{1}{4} \log \det(\Lambda^{-1})$,*

$$\mathbb{P}\left[\frac{1}{p} \sum_{i=1}^p X_i \nsucceq \frac{\eta}{4} \cdot \Lambda\right] \leq \delta.$$

*In particular,*

$$\mathbb{P}\left[\det\left(\frac{1}{p} \sum_{i=1}^p X_i\right) \leq \left(\frac{\eta}{4}\right)^D \det(\Lambda)\right] \leq \delta.$$

*Proof.* The proof follows along the lines of Simchowitz et al. [2018], sharpened slightly for the less general setting. Let $\Sigma = \sum_{i=1}^p X_i$. By a Chernoff bound (Lemma 16), for any $v \in \mathbb{S}^{D-1}$, $\mathbb{P}[v^\top \Sigma v \leq \eta p v^\top \Lambda v / 2] = \mathbb{P}[\sum_{i=1}^p v^\top X_i v \leq \eta p v^\top \Lambda v / 2] \leq \mathbb{P}[\sum_{i=1}^p \mathbb{I}\{v^\top X_i v \leq \eta v^\top \Lambda v\} \leq \eta p/2] \leq \exp(-\eta p/8)$, where we use that $X_i \succeq 0$. Hence, for any finite subset $\mathcal{T} \subset \mathbb{S}^{D-1}$,

$$\mathbb{P}[v^\top \Sigma v \geq v^\top \Lambda v \cdot \frac{\eta p}{2}, \quad \forall v \in \mathcal{T}] \geq 1 - \exp(-\eta p/8 + \log |\mathcal{T}|).$$

To conclude, we show that there exists a finite set $\mathcal{T}$ of size at most $\exp(\frac{D}{2} \log(\frac{12B}{\eta}) + \frac{1}{2} \log \det(\Lambda))$ such that, if $v^\top \Sigma v \geq v^\top \Lambda v \cdot \eta T/2$ for all $\mathcal{T}$, then $\Sigma \succeq \frac{\eta T}{4}$.

We take $\mathcal{T}$ to be an $\varepsilon = \sqrt{\eta/4B}$-net of the set $\mathcal{S}_\Lambda := \{v \in \mathcal{R}^d : v^\top \Lambda v = 1\}$. Then, if $v^\top \Sigma v \geq v^\top \Lambda v \cdot \eta p/2 = \eta p/2$ for all $\tilde{v} \in \mathcal{S}_\Lambda$, it holds

$$\tilde{v}^\top \Sigma \tilde{v} \geq \frac{1}{2} v^\top \Sigma v - (\tilde{v} - v)^\top \Sigma (\tilde{v} - v) \geq \frac{1}{2} \eta T - B p \|\tilde{v} - v\|^2 \geq \frac{\eta p}{4},$$

which means that $\Sigma \succeq \frac{\eta p}{4} \Lambda$. Define the ellipsoid $\mathcal{E}_\Lambda := \{v \in \mathcal{R}^d : v^\top \Lambda v \leq 1\} \supset \mathcal{S}_\Lambda$. Note that since $\lambda_{\max}(\Lambda) \leq B$, $\mathcal{E}_\Lambda \supset \{v : \|v\|^2 \leq 1/B\} \supset 2\varepsilon \mathcal{B}_1^D$, since $\varepsilon = \sqrt{\eta/4B} \leq \frac{1}{2\sqrt{B}}$.

$$|\mathcal{T}| \leq \frac{\operatorname{vol}(\frac{\varepsilon}{2} \mathcal{B}_1^D + \mathcal{S}_\Lambda)}{\operatorname{vol}(\frac{\varepsilon}{2} \mathcal{B}_1^D)} \leq \frac{\operatorname{vol}(\frac{\varepsilon}{2} \mathcal{B}_1^D + \mathcal{E}_\Lambda)}{\operatorname{vol}(\frac{\varepsilon}{2} \mathcal{B}_1^D)} \leq \frac{\operatorname{vol}(\frac{5}{4} \mathcal{E}_\Lambda)}{\operatorname{vol}\left(\frac{\varepsilon}{2} \mathcal{B}_1^D\right)} = \left(\frac{8}{5\varepsilon}\right)^D \det(\Lambda^{-1/2}).$$

Hence, we can take

$$\log|\mathcal{T}| \le \frac{D}{2}\log(\frac{64}{25\varepsilon^2}) + \frac{1}{2}\log\det(\Lambda) \le \frac{D}{2}\log(\frac{12B}{\eta}) + \frac{1}{2}\log\det(\Lambda).$$

∎

**Lemma 46.** *Let $\Psi : \mathcal{R}^d \to \mathbb{S}_+^D$ be any function whose image are PSD matrices whose entries are polynomials of degree at most $\ell$. Let $\rho$ be any distribution which is $\sigma$-smooth with respect to a log-concave measure $\mu$. Then, there exist a universal constant $C$ such that, for any $v \in \mathcal{R}^D \setminus 0$,*

$$\mathbb{P}_{x\sim\rho}[v^\top \Psi(x)v \le \sigma^\ell(C\ell)^{-\ell}v^\top \mathbb{E}_{x\sim\mu}\Psi(x)v] \le \frac{1}{2}.$$

*Proof.* Consider the polynomial function $f_v(x) = v^\top \Psi(x)v$. This is a polynomial of degree $\ell$ in $x$, and nonnegative. By Carbery and Wright [2001, Theorem 8], with $q = \ell$, we have

$$\mathbb{E}_{x\sim\mu}[f_v(x)]^{1/\ell} \cdot \alpha^{-1/\ell} \cdot \mathbb{P}_{x\sim\mu}[f_v(x) \le \alpha] \le C'\ell,$$

where $C'$ is a universal constant. Reparametrizing $\alpha \leftarrow \alpha \cdot \mathbb{E}_{x\sim\mu}[f_v(x)]$, we have

$$\mathbb{P}_{x\sim\mu}[f_v(x) \le \alpha\,\mathbb{E}_{x\sim\mu}[f_v(x)]] \le C'\ell\alpha^{1/\ell}.$$

To conclude, take $\alpha = (2C'\ell/\sigma)^{-\ell}$, we get

$$\mathbb{P}_{x\sim\mu}[f_v(x) \le (2C'\ell/\sigma)^{-\ell}\,\mathbb{E}_{x\sim\mu}[f_v(x)]] \le \frac{\sigma}{2}$$

Hence,

$$\mathbb{P}_{x\sim\rho}[f_v(x) \le (2C'\ell/\sigma)^{-\ell}\,\mathbb{E}_{x\sim\mu}[f_v(x)]] \le \frac{1}{2}$$

Taking $C = 2C'$ and substituting $f_v(x) = v^\top \Psi(x)v$ concludes. ∎

*Proof of Proposition 41.* Let $\rho_t$ denote the conditional distribution of $x_t \mid x_1, \ldots, x_{t-1}$. By assumption, $\rho_t$ is $\sigma$-smooth with respect to the log-concave measure $\mu$, so

$$\mathbb{P}[v^\top \Psi(x_t)v \le (C\ell)^{-\ell}v^\top \mathbb{E}_{x\sim\mu}\Psi(x)v \mid x_1, \ldots, x_{t-1}] \le \frac{1}{2}$$

Hence, we can apply Lemma 45 with $\eta \leftarrow 1/2$, $B \leftarrow B$, and $\Lambda \leftarrow (C\ell)^{-\ell}\Lambda$. Using that $\det((C\ell)^{-\ell}\Lambda) = (C\ell)^{-D\ell}\det(\Lambda)$ concludes. ∎

# F Proofs from Section 6

In this section, we prove the extensions of our results to the multipiece setting.

## F.1 Proof of Theorem 11

**Algorithm Description.** Algorithm 6 gives our algorithm for $K$-class classification. We maintain $\binom{K}{2}$ instances of the binary classification algorithm, Algorithm 1. That is, each $\mathcal{A}_{\text{bin}}$ maintains a $w_t^{(i,j)}$ at each time $t$, and

$$\mathcal{A}_{\text{bin}}^{(i,j)}.\textbf{classify}(x) = \text{sign}(\langle w_t^{(i,j)}, w \rangle).$$

To gain intuition, recall that we assume the ground-truth classifier to

$$f^\star(x) = \arg\max_{i \in [K]} \langle x, w_\star^i \rangle, \tag{F.1}$$

where the argmax is taken lexicographically. Hence, $f^\star(x)$ admits the following equivalent representation:

$$\begin{aligned}
f^\star(x) &= \min_{i \in [K]} \{i : \langle x, w_\star^i \rangle \geq \max_{j > i} \langle x, w_\star^j \rangle\} \\
&= \min_{i \in [K]} \{i : \text{sign}(\langle x, w_\star^i - w_\star^j \rangle \geq, \ \forall i > i\} \tag{F.2}
\end{aligned}$$

Hence, $f^\star(x)$ can be thought of running a lexicographic tournament, picking out the first index $i$ which 'wins' over all lesser indices $k$. This is what motivates the selection of $\hat{y}$ in Appendix F.1 of Algorithm 6.

*Proof of Theorem 11.* We reduce to the generalized, "censored" variation of our linear classification setting, depicted in Proposition 32. For pairs $i < j$, define

$$w_\star^{(i,j)} := w_\star^i - w_\star^j \quad y_t^{(i,j)} := \text{sign}(\langle w_\star^{(i,j)}, x_t \rangle), \quad \hat{y}_t^{(i,j)} := \mathcal{A}_{\text{bin}}^{(i,j)}.\textbf{classify}(x_t)$$

Note that

$$y_t^{(i,j)} = 1 \ \ \forall j > i \text{ whenever } i = y_t. \tag{F.3}$$

For simplicty, lets assume that $w^{\star,(i,j)} \neq 0$ for $i < j$. We address the edgecase where this term may be zero at the end. Further, let $i_t < j_t$ denote the indices select in Equation (F.4). Then, since the algorithm always selects such a pair $(i_t, j_t)$ whenever a mistake is made (and defining, say $(i_t, j_t) = (0,0)$ to indicate no mistake),

$$\sum_{t=1}^T \mathbb{I}\{\hat{y}_t \neq y\} = \sum_{i < j} \sum_{t=1}^T \mathbb{I}\{(i_t, j_t) = (i,j)\}.$$

The following claim reduces to binary-losses.

**Claim 2.** *For the indices $i_t < j_t$ selected in Equation (F.4), and any $1 \leq i < j \leq K$,*

$$\mathbb{I}\{(i_t, j_t) = (i,j)\} = \mathbb{I}\{y_t^{(i,j)} \neq \hat{y}_t^{(i,j)}\}\mathbb{I}\{(i_t, j_t) = (i,j)\}.$$

*Moreover, when $(i_t, j_t) = (i,j)$, $y_t^{(i,j)} = \text{sign}(\hat{y}_t - y_t)$, and thus can be determined by learner.*

*Proof.* Indeed, at a round where $\mathbb{I}\{(i_t, j_t) = (i,j)\}$, we have $\hat{y}_t \neq y_t$. We have two cases

- When $y_t < \hat{y}_t$, then Equation (F.4) selects $i = y_t$ and $j$ as some index for which $\hat{y}_t^{(i,j)} = -1$, such an index must exist by the choice of $\hat{y}_t$ in Appendix F.1 (otherwise, either $\hat{y}_t < y_t$, or else $y_t$ would be correctly selected as the true class). On the other hand, $y_t^{(i,j)} := \text{sign}(\langle w_\star^{(i,j)}, x_t \rangle) = 1 = \text{sign}(\hat{y}_t - y_t)$ by Equation (F.3). Thus, $y_t^{(i,j)} \neq \hat{y}_t^{(i,j)}$

- If $\hat{y}_t < y_t$, then from Appendix F.1 it must be the case that $\hat{y}_t^{(i,j)} = 1$ for $i = \hat{y}_t$ and $j = y_t$ being the indices selected in Equation (F.4). But by the reverse of Equation (F.3), $y_t^{(i,j)} = -1 = \mathrm{sign}(\hat{y}_t - y_t)$. Hence, $\hat{y}_t^{(i,j)} \neq y_t^{(i,j)}$.

∎

Hence, we may write

$$\sum_{t=1}^{T} \mathbb{I}\{\hat{y}_t \neq y\} = \sum_{i<j} \sum_{t=1}^{T} \ell_t^{(i,j)}, \quad \ell_t^{(i,j)} := \mathbb{I}\{y_t^{(i,j)} \neq \hat{y}_t^{(i,j)}\} \mathbb{I}\{(i_t, j_t) = (i,j)\}.$$

We now claim that the losses $\ell_t^{(i,j)} := \mathbb{I}\{y_t^{(i,j)} \neq \hat{y}_t^{(i,j)}\} \mathbb{I}\{(i_t, j_t) = (i,j)\}$ precisely corresponding to the censored binary setting of Proposition 32. Indeed, consider a setting where $x_1, x_2, \dots$ are selected by the $\sigma$-smooth adversary, and the label is $\hat{y}_t^{(i,j)}$ defined above. $\mathcal{A}_{\mathrm{bin}}^{(i,j)}$ does not always see $\hat{y}_t^{(i,j)}$, but whenever $\ell_t^{(i,j)} = 1$, Claim 2 shows that the learner does indeed observe the true value $\hat{y}_t^{(i,j)}$. Thus, by Proposition 32, it holds for any fixed $i < j$ that with probability $1 - \delta$,

$$\sum_{t=1}^{T} \ell_t^{(i,j)} \leq 136 d \log(d) + 34 \log\left(\frac{T}{\sigma\delta}\right) + 56$$

Union bounding over all $\binom{K}{2} \leq K^2$ pairs $i < j$ and summing, we conclude that with probability $1 - \delta$,

$$\mathrm{Reg}_T = \sum_{i>j} \sum_{t=1}^{T} \ell_t^{(i,j)} \leq 136 K^2 d \log(d) + 34 K^2 \log\left(\frac{TK^2}{\sigma\delta}\right) + 56 K^2.$$

$$\leq 136 K^2 d \log(d) + 90 K^2 \log\left(\frac{TK^2}{\sigma\delta}\right)$$

**Modification for non-unique ground truth classifiers.** Here, we can modify $\mathcal{A}_{\mathrm{bin}}^{(i,j)}$ with the following rule: predict $\hat{y}_t^{(i,j)} = 1$ until there is an time $t$ for which $(i_t, j_t) = (i,j)$, and then reinitialize $\mathcal{A}_{\mathrm{bin}}^{(i,j)}$ to have $w_t^{(i,j)} = e_1$, as in Algorithm 1.

Consider an $i < j$ with $w_\star^i = w_\star^j$. We claim $(i_t, j_t) \neq (i,j)$ for any $t$. Now, suppose there is a time $t$ that $(i_t, j_t) = (i,j)$, let $\tau$ denote the first time $t$ for which this is true. Then, $\hat{y}_t^{(i,j)} = 1$. But in addition $y_t \neq j$ for any $t$ because we assume the $\arg\max$ in Equation (F.1) is broken lexicograophically. Thus, from Equation (F.4), it must be that $y_\tau = i$, and that $j$ is such that $\hat{y}_t^{(i,j)} = -1$; this gives a contradiction.

Now consider $i < j$ with $w_\star^i \neq w_\star^j$. Then our modification of $\hat{y}_t^{(i,j)}$ only increases $\sum_{t=1}^{T} \ell_t^{(i,j)}$ by at most 1. This adds at most $\binom{K}{2} < K^2$ to the total regret (modifying the constant of 90 to 91).

∎

### F.2 Formal Guarantees for Piecewise Regression

We will prove a slightly more general version of Theorem 12 and then derive the result in Section 6 as a corollary. First, we will define what kinds of regression classes our result will apply to:

**Definition 47.** *Let $\mathcal{G} : \mathcal{X} \to \mathbb{R}$ be a function class. We say that $\mathcal{G}$ is $\ell$-determined with respect to some measure $\mu$ on $\mathcal{X}$ if the following two conditions hold:*

- *The values on $\ell$ points in general position uniquely determine the function, i.e.,*

$$\mathbb{P}\left(\text{there exist } g \neq g' \in \mathcal{G} \text{ such that } g(x_i) = g'(x_i) \text{ for } 1 \leq i \leq \ell \text{ and } x_i \sim \mu\right) = 0 \quad \text{(F.5)}$$

- *Two functions intersect only on measure zero sets, i.e., for all $g, g' \in \mathcal{G}$,*

$$\mu\left(\{x \in \mathcal{X} : g(x) = g'(x)\}\right) = 0 \quad \text{(F.6)}$$

**Algorithm 6** $K$-class linear classification

---

1: **Initialize** Binary classifiers $\mathcal{A}_{\text{bin}}^{(i,j)}$, $i < j$
2: **for** $t = 1, 2, \ldots$ **do**
3:     **recieve** $x_t$
4:     **for** $i < j$ **do** $\hat{y}_t^{(i,j)} = \mathcal{A}_{\text{bin}}^{(i,j)}.\textbf{classify}(x_t)$
5:     **predict** $\hat{y}_t = \min\{i \in [K] : \hat{y}_t^{(i,j)} = 1, \ i < j \le K\}$         (% self.$\textbf{classify}(x_t)$)
6:     **if** $\hat{y}_t \ne y_t$ **then**                                            (% self.$\textbf{errorUpdate}(x_t)$)
7:         Define

$$(i,j) = \begin{cases} i = y_t, \ j \in \{j > i : \hat{y}_t^{(i,j)} = -1\} & \text{if } y_t < \hat{y}_j \\ i = \hat{y}_t, \ j = y_t & \text{if } \hat{y}_j < y_t \end{cases} \tag{F.4}$$

8:         Update $\mathcal{A}_{\text{bin}}^{(i,j)}.\textbf{errorUpdate}(x_t)$

---

Note that linear classes in $\mathbb{R}^d$ are trivially $d$-determined with respect to the Lebesgue measure, and thus with respect to any measure absolutely continuous with respect to the Lebesgue measure. Polynomial classes are also $\ell$-determined with respect to the Lebesgue measure for some $\ell$ depending on $d$ and the degree of the polynomials. We observe that our definition of an $\ell$-determined function class is an offline analogue to the notion of eluder dimension from Russo and Van Roy [2013].

Now, for a given function class $\mathcal{G} : \mathcal{X} \to \mathbb{R}$, we denote by

$$\mathcal{G}_{\mathcal{F}} = \left\{ x \mapsto \mathbf{g}_f(x) = \sum_{i=1}^K g_i(x) \mathbb{I}[f(x) = i] \Big| g_i \in \mathcal{G} \text{ and } f \in \mathcal{F} \right\} \tag{F.7}$$

where $\mathcal{F}$ is the set of $K$-class linear classifiers from Theorem 11. We will continue to suppose that the $x_t$ are drawn from distributions that are $\sigma$-smooth with respect to $\mu$ and that the labels $y_t$ are realizable with respect to $\mathcal{G}_{\mathcal{F}}$.

**Assumption 1** (Oblivious, realizable smoothed sequential setting). *We suppose smoothed online learning setting and the adversary is realizable with respect to $\mathcal{G}_{\mathcal{F}}$ and oblivious in the sense that before the learning process begins, the adversary chooses $\mathbf{g}^\star = (g_1^\star, \ldots, g_K^\star) \in \mathcal{G}^K$ and $f^\star \in \mathcal{F}$ and lets $y_t = (\mathbf{g}^\star)_{f^\star}(x_t)$ for all $t$. We assume further that $\mathbf{g}^\star$ has* unique *entries: $g_i^\star \ne g_j^\star$ for $i \le j$.*

Lastly, we assume we have access to the following ERM oracle.

**Definition 48** (ERM Oracle). *Given $\mathcal{U} = \{(x_1, y_1), \ldots, (x_n, y_m)\}$, where $(x_i, y_i) \in \mathcal{B}_1^d \times \mathcal{Y}$, $\textbf{ERM}(\mathcal{U}, \mathcal{G}, K)$ returns a $n \le K$, and $g_1, \ldots, g_n$ and partition $C_1, \ldots, C_n$ of $\mathcal{U}$ such that, for all $(x, y) \in C_i$, $g_i(x) = y$. By post-processing, we may also assume that $g_i$ are distinct*[4]

**Proposition 49** (General $\ell$-Determined Regression). *Suppose that we are in the semi-oblivious, smoothed online learning setting, where the adversary begins by choosing $g_{f^\star}^\star$, $\mathcal{G}_{\mathcal{F}}$ from (F.7), and, at each time $t$, draws $x_t$ from a distribution that is $\sigma$-smooth with respect to $\mu$ and sets $y_t = g_{f^\star}^\star(x_t)$. Suppose further that $\mathcal{G}$ is $\ell$-determined, in the sence of Definition 47. Then, Algorithm 7 satisfies for all $T$, with probability at least $1 - \delta$,*

$$\text{Reg}_T \le 136 K^2 d \log(d) + 91 K^2 \log\left(\frac{TK^2}{\sigma\delta}\right) + K^2(\ell + 1) \tag{F.8}$$

*Moreover, the per-time step computational complexity of Algorithm 7 is polynomial in $d$ and the complexity of the* **ERM** *oracle Definition 48, applied to a data set $\mathcal{U}$ of size no more that $|\mathcal{U}| \le K(\ell + 1)$.*

### F.3 Algorithm for Piecewise Regression

Algorithm 7 proceeeds at follows. We let $N_t$ denote the number of clusters about which we are certain, $\mathcal{U}_t$ denote the set of points which cannot be assigned to a cluster. We maintain a supervised

---

[4]Note that the ERM Oracle need not cluster with respect to the classifiers (even thought it can certainly be implemented this way). Hence, one can can merge cluster to ensure $g_i$ is distinct.

---

**Algorithm 7** General Piecewise Regression

---

1: **Init:** $K$-class supervised linear classifier $\mathcal{A}$ (instance of Algorithm 8)
  ERM-oracle **ERM** (see )
2: **for** each time $t = 1, 2, \dots$ **do**
3:   **recieve** $x_t$
4:   **predict** $\hat{y}_t = \hat{g}_k(x_t)$ for $k = \hat{k}_t$, where $\hat{k}_t := \mathcal{A}.\textbf{classify}(x_t, N_t)$ `% ` $\hat{y}_t = 0$ `if ` $N_t = 0$
5:   **observe** $y_t$.
6:   **if** $\exists k_t^\star \in [N_t]$ with $\hat{g}_k(x_t) = y_t$ **then**
  `% update classification`
7:     **if** $\hat{k}_t \neq k_t^\star$ **then**, $\mathcal{A}.\textbf{errorUpdate}(x_t, N_t)$
8:     **maintain** $N_{t+1} \leftarrow N_t, \mathcal{U}_{t+1} \leftarrow \mathcal{U}_t$
9:   **else**`% update clustering`
10:     $(C_{1:n}, g_{1:n}) \leftarrow \textbf{ERM}(\tilde{\mathcal{U}}_t, \mathcal{G}, K)$, $\tilde{\mathcal{U}}_t = \mathcal{U}_t \cup \{(x_t, y_t)\}$
  `% Initialize ` $\tilde{N} = N_t$
11:     **for** each $i : |C_i| \geq \ell + 1$ **do**
12:       $\tilde{N} = \tilde{N} + 1, \hat{g}_{\tilde{N}} \leftarrow g_i,$
13:       $N_{t+1} \leftarrow \tilde{N}, \ \mathcal{U}_{t+1} \leftarrow \tilde{\mathcal{U}}_t \setminus \bigcup_{i:|C_i| \geq \ell+1} \{(x, y) \in \tilde{\mathcal{U}}_t : g_i(x) = y\}$

---

---

**Algorithm 8** $K$-class linear classification with supervision

---

1: **Initialize** Binary classifiers $\mathcal{A}_{\text{bin}}^{(i,j)}, i < j$
2: **for** $t = 1, 2, \dots$ **do**
 `% guarantee ` $y_t \leq M_t$
3:   **Recieve** $(x_t, M_t)$ and **predict**         (`% self`.**classify**$(x_t, M_t)$)

$$\hat{y}_t = \min\{i \in [M_t] : \mathcal{A}_{\text{bin}}^{(i,j)}.\textbf{classify}(x_t) = 1, \ i < j \leq M_t\},$$

4:   **Observe** $y_t$
5:   **if** $\hat{y}_t \neq y_t$ **then**            (`% self`.**errorUpdate**$(x_t, M_t)$)
6:     Define

$$(i, j) = \begin{cases} i = y_t, \ j \in \{k > i : \langle w_1^{(i,k)}, x_t \rangle < 0\} & \text{if } y_t < \hat{y}_j \\ i = \hat{y}_t, \ j = y_t & \text{if } \hat{y}_j < y_t \end{cases} \tag{F.9}$$

7:     Update $\mathcal{A}_{\text{bin}}^{(i,j)}.\textbf{errorUpdate}(x_t)$

---

$K$-class linear classifier, $\mathcal{A}$, described in Algorithm 8. It is similar in spirit to Algorithm 6, except it takes in "side information" $M_t$ on which it only predicts from the first $M_t$ classes. Lastly, we maintain a growing sequence of regressors $\hat{g}_1, \hat{g}_2, \dots \in \mathcal{G}$ such that $\hat{g}_i$ does not change once assigned, and $\hat{g}_i$ is defined for all $i \leq N_t$.

At each time $t$, we call $\hat{k}_t = \mathcal{A}.\textbf{classify}(x_t, N_t)$ to guess the cluster of $x_t$, only among cluster $i \leq N_t$ about which we are certain. Then, we predict $\hat{y}_t = \hat{g}_{\hat{k}_t}(x_t)$. The idea is that, for $k = \hat{k}_t \leq N_t$, we are sure that $\hat{g}_k$ is the true predictor if $x_t$ is in cluster $k$. We then observe $y_t$. If $y_t$ was correctly predicted by one of that $\hat{g}_i$ for which $i \leq N_t$, but not the $\hat{k}_t$ we guessed, then we update our classifier $\mathcal{A}$. Otherwise, we call the ERM oracle to determine if we can find new cluster(s) to add, appending to our sequence of predictors $\hat{g}$'s, and growing our number of certain clusters $N_t$. Note that we never maintain an *explicit* clustering of our points, but only cluster retroactively based on whether $\hat{g}_i(x_t) = y_t$ for some $i$, as a means to recover the classification label.

### F.4 Proof of Proposition 49

#### F.4.1 Guarantee for ERM procedure

**Lemma 50.** *Let $I \subset [T]$ be any subset of time. Then with probability one, it holds that for any partition $C_1, \ldots, C_n$ of $(x_s, y_s)_{s \in [I]}$ and any $g_1, \ldots, g_n$ distinct functions such that, for all $(x, y) \in C_n$, $\tilde{g}_i(x) = y$, then for any index $i$ for which $|C_i| \geq \ell + 1$,*

- $f^\star(x) = f^\star(x')$ *for all* $(x, y), (x', y') \in C_i$
- $\tilde{g}_i = g^\star_{f^\star(x)}$, *representative* $x \in C_i$

*Proof.* Let $I_1, \ldots, I_m$ denote the times in each cluster $C_1, \ldots, C_n$. Without loss of generality, suppose $I_1$ is a cluster for which $|I_1| \geq \ell + 1$ (we may handle all simultaneously via a finite union bound.)

**Item 1.** Suppose in fact that there exists $s, s' \in I_1$ with $f^\star(x_s) \neq f^\star(x_{s'})$. We first argue then that $g_1 \neq g^\star_k$ for all $k \in [K]$. Indeed, by smoothness and the second condition of Definition 47, it holds that with probability 1, $g^\star_k(x_{\tilde{s}}) \neq g^\star_{k'}(x_{\tilde{s}})$ for all $1 \leq \tilde{s} \leq T$ and $k \neq k'$. Set $i_1 = f^\star(x_s)$ and $i_2 = f^\star(x_{s'})$. Thus, if $g_i = g^\star_k$, the fact $g_i(x_s) = \mathbf{g}^\star_{f^\star}(x_s)$ and $g_i(x_{s'}) = \mathbf{g}^\star_{f^\star}(x_{s'})$ would require both $g^\star_k(x_s) = \mathbf{g}^\star_{f^\star}(x_s) = g^\star_{i_1}(x_s)$ and $g^\star_k(x_{s'}) = \mathbf{g}^\star_{f^\star}(x_{s'}) = g^\star_{i_2}(x_{s'})$. Thus, on the aforementioned probability one event, we would have both $k = i_1$ and $k = i_2$, which contradicts the supposition $i_1 \neq i_2$.

Next, let $S \subset [T]$ denote a set of indices. Denote $s_{\max} = \max\{s \in S\}$, and define the events

$$\mathcal{A}_S(g') := \{\exists g \in \mathcal{G} \setminus \{g'\} : g'(x_{s_{\max}}) = y_{s_{\max}}, \quad \forall s \in S, g(x_s) = y_s\}.$$

By the above observation that $g_i \neq g^\star_k$ for any $k$, we see that if there exists $s, s' \in I_1$ with $f^\star(s) \neq f^\star(s')$ with $|I_1| \geq \ell + 1$, then one of the events $\mathcal{A}_S(g^\star_k)$ must occur for some $|S| \geq \ell + 1$ and $k \in [K]$. Since there are only finitely many such events, it suffices to show that for any *fixed* $S$ and $k$, $\mathbb{P}[\mathcal{A}_S(g^\star_k)] = 0$.

Hence, fix $S$ and $k$. For a given $S$ with max element $s_{\max}$, let $\mathscr{F}_{-1}$ denote history generated by $(x_1, y_1), \ldots, (x_{s_{\max}-1}, y_{s_{\max}-1})$. Define the $\mathcal{A}_{-1} := \{\exists g \in \mathcal{G} \setminus \{g^\star_k\} : s \in S, g(x_s) = y_s, s \in S \setminus \{s_{\max}\}\}$. Then, $\mathcal{A}_{-1}$ is $\mathscr{F}_{-1}$ measurable and $\mathcal{A}_{-1}$ contains $\mathcal{A}_S(g')$. Hence,

$$\mathbb{P}[\mathcal{A}_S(g^\star_k)] = \mathbb{E}[\mathbb{P}[\mathcal{A}_S(g^\star_k) \mid \mathscr{F}_{-1}]]$$
$$= \mathbb{E}[\mathbb{I}\{\mathcal{A}_{-1}\} \cdot \mathbb{P}[\mathcal{A}_S(g^\star_k) \mid \mathscr{F}_{-1}]].$$

By the first condition of Definition 47, $\mathcal{A}_{-1}$ coincides with the event $\mathcal{A}'_{-1} := \{\exists \text{ a unique } g \in \mathcal{G} \setminus \{g^\star_k\} : s \in S, g(x_s) = y_s, s \in S \setminus \{s_{\max}\}\}$ almost surely. Hence,

$$\mathbb{P}[\mathcal{A}_S(g^\star_k)] = \mathbb{E}[\mathbb{I}\{\mathcal{A}'_{-1}\} \cdot \mathbb{P}[\mathcal{A}_S(g^\star_k) \mid \mathscr{F}_{S-1}]].$$

Lastly, when $\mathcal{A}'_{-1}$ holds, let $\hat{g} \neq g^\star_k$ denote the unique $g \neq g^\star_k$ consistent with examples $s \in S \setminus \{s_{\max}\}$. Since $\hat{g}$ is determined by $\mathscr{F}_{S-1}$, we have

$$\mathbb{P}[\mathcal{A}_S(g^\star_k) \mid \mathscr{F}_{S-1}] \leq \mathbb{P}[\hat{g}(x_{s_{\max}}) \neq g^\star_k(s_{\max})] = 0,$$

where we use that $\hat{g}$ is *fixed*, that $\hat{g} \neq g'$, and the second condition of Definition 47. The bound follows.

**Item 2.** For any fixed set of indices $\tilde{I}$ with $|\tilde{I}| \geq \ell + 1 \geq \ell$, the first condition of $\ell$-determination (Definition 47) ensures then that $\mathbb{P}[\exists g_1 \neq g^\star_j : g_1(x_s) = g^\star_j(x_s), \forall x \in \tilde{I}] = 0$. The bound follows by union boundig over all $\tilde{I} \subset [T]$ and $j \in [K]$.

∎

#### F.4.2 Distinctness of clustering

**Claim 3** ($\mathcal{U}_t$ is Uncertain Set). *Fix a time $t$. Then, for any $(x, y) \in \mathcal{U}_t$ and any $i \leq N_t$, $\hat{g}_i(x) \neq y$.*

*Proof.* This is true vacuously at time $t = 1$, when $\mathcal{U}_t = \emptyset$. Suppose it holds at time $t$, we prove it for time $t + 1$. If $x_t$ is such that there exists an $n \leq N_t$ with $\hat{g}_i(x_t) = y_t$, then $\mathcal{U}_{t+1}$ does not change from $\mathcal{U}_t$. Otherwise, if $\hat{g}_n(x_t) \neq y_t$ for all $n \leq N_t$,

$$\mathcal{U}_{t+1} \leftarrow \tilde{U}_t \setminus \bigcup_{i:|C_i| \geq \ell+1} \{(x, y) \in \tilde{\mathcal{U}}_t : g_i(x) = y\}, \quad \tilde{U}_t := \mathcal{U}_t \cup \{(x_t, y_t)\} \tag{F.10}$$

where $g_i$ and $C_i$ are the clustering from the ERM oracle. By the inductive hypothesis and fact that $\hat{g}_n(x_t) \neq y_t$ for all $n \leq N_t$, it follows that $\hat{g}_n(x) \neq y$ for all $(x, y) \in \mathcal{U}_t \cup \{(x_t, y_t)\} = \tilde{\mathcal{U}}_t \supseteq \mathcal{U}_{t+1}$. Now, if there is some $n : N_t < n \leq N_t$ for which $\hat{g}_n(x) = y$, then that $\hat{g}_n$ was added during the ERM step at round $t$: i.e. $\hat{g}_n = g_i$ for some $i$ such that $|C_i| \geq \ell + 1$. But then $(x, y)$ is removed form $\mathcal{U}_{t+1}$ by Equation (F.10). ∎

**Claim 4.** *Fix a time $t$. Then, for any $i, j \leq N_t$, $\hat{g}_i \neq \hat{g}_j$.*

*Proof.* This is trivially true at time $t = 1$. Suppose this is true at time $t$, we establish the claim for time $t + 1$. If $x_t$ is such that there exists an $i \leq N_t$ with $\hat{g}_i(x_t) = y_t$, then $N_{t+1} = N_t$ and so the set of $\hat{g}_i$'s under consideration remains unchanged.

On the other hand, suppose there is no $i \leq N_t$ with $\hat{g}_i(x_t) = y_t$. Then, all possible new $\hat{g}_j$'s for $N_t < j \leq N_{t+1}$ are correct on some subset of points of $\mathcal{U}_t \cup (x_t, y_t)$. But by the previous claim (Claim 3) and the assumption that, for $i \leq N_t$ with $\hat{g}_i(x_t) = y_t$, no element of $\mathcal{U}_t \cup (x_t, y_t)$ is correctly predicted by any $\hat{g}_i$ for $i \leq N_t$. Thus, none of the new $\hat{g}_j$'s can equal an $\hat{g}_i$ for $i \leq N_t$. Moreover, by the definition the ERM oracle, Definition 48, all newly added $\hat{g}_j$'s are distinct. ∎

### F.4.3 Key summary of Algorithm 7

We now summarize the results with the following lemma.

**Lemma 51.** *With probability $1$, there exists a permutation $\pi$ such that*

- *For each time $t$ and $i \in [N_t]$, $\hat{g}_i = g^\star_{\pi(i)}$*

- *For each time $t$ and $i \in [N_t]$, $\hat{g}_i(x) = y$ if and only if $f^\star(x) = \pi(i)$*

- *If $(x, y) \in \mathcal{U}_t$, then $\pi^{-1}(f^\star(x)) > N_t$.*

- *Whenever $\hat{y}_t \neq y_t$, either $\pi^{-1}(f^\star(x)) > N_t$, or $\hat{k}_t := \mathcal{A}.\text{classify}(x_t, N_t)$ has $\pi(\hat{k}_t) \neq f^\star(x_t)$.*

*Proof.* For $n = N_T$, let $\hat{g}_1, \ldots, \hat{g}_n \in \mathcal{G}$ denote the functions constructed by our algorithm. Since each new $\hat{g}_i$ is added from a cluster with at least $\ell + 1$ points, applying Lemma 50 (with a union bound over index sets $I$ ensures that $\hat{g}_i = g^\star_j$) for some $j \in [K]$. This gives us a mapping $\pi : [n] \to [K]$. $\pi$ must be injective, since $g^\star_j$ are distinct by assumption, and $\hat{g}_i$ are unique by Claim 4 (in particular, $n \leq K$). Thus, $\pi$ can be extednded to a permutation from $[K] \to [K]$. By construction, the first item is satisfied.

The second item is a consequence of uniquenessof that the previous point, uniqueness of $\hat{g}_i$'s, and the second point of Definition 47 , since we only need to union bound over finitely many times $t \in [T]$ and pairs $g^\star_i, g^\star_j$'s. The third item follows similarly, by invoking Claim 3.

For the last point, suppose $\pi^{-1}(f^\star(x_t)) \leq N_t$. Then, by the previous point, $(x_t, y_t) \notin \mathcal{U}_t$. Thus, the algorithm classifies $\hat{y}_t = \hat{g}_{\hat{k}_t}(x_t)$ where $\hat{k}_t \in [N_t]$. But by the first point of the lemma, $\hat{g}_{\hat{k}_t} = g^\star_{\pi(\hat{k}_t)}$. So if $\pi(\hat{k}_t) = f^\star(x_t)$, then we woudl have $\hat{g}_{\hat{k}_t}(x_t) = g^\star_{f^\star(x_t)}(x_t) = y_t$, a contradiction. ∎

### F.4.4 Proof of Proposition 12

Let $\pi$ denote the permutation ensured by Lemma 51. We may assume without loss of generality that $\pi$ is the identity permutation (by permuting $\mathbf{g}^\star$). Let $k_t = f^\star(x_t)$. Recalling also that $\hat{k}_t \leq N_t$, the

fourth point of Lemma 51 ensures.

$$\mathbb{I}\{\hat{y}_t \leq y_t\} \leq \mathbb{I}\{k_t > N_t\} + \sum_{t=1}^{T} \mathbb{I}\{\hat{k}_t \neq k_t, k_t \leq N_t\}$$

First, we bound the contribution of $\mathbb{I}\{k_t > N_t\}$:

**Claim 5.** $\sum_{t=1}^{T} \mathbb{I}\{k_t = k, \quad k > N_t\} \leq K(\ell + 1)$. *Thus,* $\sum_{t=1}^{T} \mathbb{I}\{k_t > N_t\} \leq K^2(\ell + 1)$.

*Proof.* Suppose $k > N_t$, and let $S_{t,k} := \{s \leq t : f^\star(x_s) = k\}$, and define $\tau_k = \max\{t \in [T] : k_t > N_t, k_t = k\}$. Then

$$\sum_{t=1}^{T} \mathbb{I}\{k_t > N_t, k_t = k\} = |S_{\tau_k, k}|.$$

We claim that $|S_{\tau_k, k}| \leq K(\ell + 1)$. Indeed, suppose $|S_{\tau_k, k}| > K(\ell + 1)$. Then, for some $t < \tau_k$, $|S_{t,k}| = K(\ell + 1)$ and $k_t = k$ and $k > N_t$. By Lemma 51, $g^\star_{k_t}(x_t) \neq \hat{g}_i(x_t)$ for any $i \leq N_t$. Hence, our algorithm executes Appendix F.3. By the pidgeon-hole principle, there must be at least one cluster $C_i : |C_i| \geq \ell + 1$ which contains at least one $s \in S_{t,k}$. Hence, the update rule ensures that $i \leq N_{t+1}$ for which $\hat{g}_i(x_s) = y_s$. But again, by Lemma 51 (and taking the permutation to be the identity), we have $f^\star(x_s) = i$. In other words, $i = k_s = k$, i.e. $k_s \leq N_{t+1} \leq N_{\tau_k}$. This constradictions the definition of $\tau_k$. ∎

Summarizing our argument thus far, the following holds with probability one

$$\sum_{t=1}^{T} \mathbb{I}\{\hat{y}_t \leq y_t\} \leq K^2(\ell + 1) + \sum_{t=1}^{T} \mathbb{I}\{\hat{k}_t \neq k_t, k_t \leq N_t\} \tag{F.11}$$

Finally, by mirroring the proof of Theorem 11, we upper bound

$$\sum_{t=1}^{T} \mathbb{I}\{\hat{k}_t \neq k_t, k_t \leq N_t\} \leq 136K^2 d \log(d) + 91K^2 \log\left(\frac{TK^2}{\sigma\delta}\right). \tag{F.12}$$

The key difference between the above bound and that of Theorem 11 is that we only see when get feedback $k_t \leq N_t$, but at the same time, we only suffer a loss when $k_t \leq N_t$. Hence, the bound follows from a near-identical argument, calling the general censored version of our binary classification Proposition 32, modified to add the event $\{k_t \leq N_t\}$ to the censoring. Combining Equations (F.11) and (F.12) concludes.

∎

# G    Non-realizable mistake bounds for the Perceptron.

For simplicity, we consider regret with respect to a fixed $b^\star \in \mathcal{R}, w^\star \in \mathcal{B}_1^d$, and define

$$y_t^\star = y^\star(x_t), \quad y^\star(x) := \text{sign}(b^\star + \langle x_t, w^\star \rangle).$$

Again, we normalize $x_t$ that $\max_t \|x_t\| \leq 1$. We further assume that

$$\|b^\star\|^2 + \|w^\star\|^2 = 1, \quad \|w^\star\| \geq 1/2.$$

We show in Lemma 58 at the end of this section that this is without loss of generality. We define

$$\hat{w}^\star = w^\star / \|w^\star\|.$$

Unlike with our cutting-plane methods, we allow the adversary to deviate from a realizable clasiffier. Specifically, for each time $t$, the adversary selects $x_t \sim p_t$, and may instead choose to play some $y_t \neq y_t^\star$. We define,

$$N_{\text{err}} := 1 + |\{t : y_t \neq y_t^\star\}|,$$

and obtain non-vacuous mistake bounds provided $N_{\text{err}}$ is sublinear in $T$.

Informally, our total mistake bound for the Perceptron is polynomial in the smoothness along the direction of the optimal classifier $\hat{w}^\star$. This is formalized in the following definition:

**Definition 52** (Directional $\sigma_{\text{dir}}$-smoothness). *We say that the adversary*

- *is $(\sigma_{\text{dir}}, \hat{w}^\star)$ directionally-smooth if $\langle x_t, \hat{w}^\star \rangle$ has density at most $1/\sigma_{\text{dir}}$ with respect to the Lebesgue measure on the real line.*

- *is, more generally, $(\sigma_{\text{dir}}, \alpha, \hat{w}^\star)$ directional-Tsybakov-smooth if $\sup_{a \in \mathbb{R}} \mathbb{P}_{x_t \sim p_t}[\langle x_t, \hat{w}^\star \rangle \in [a, a + \eta]] \leq \eta^{1-\alpha}/\sigma_{\text{dir}}$.*

*Note that a $(\sigma_{\text{dir}}, \alpha, \hat{w}^\star)$-Tsybakov adversary is is $(\sigma_{\text{dir}}, \hat{w}^\star)$-smooth.*

Note that Definition 52 is a slightly weaker condition than the one consider in Theorem 7 in the body, as it only requires directional smoothness along $\hat{w}^\star$ (not uniformly). As noted in the body, directional smoothness can differ substantially from general smoothness. We provide two examples.

**Example 1** (Additive $d$-Ball Noise). *Suppore that at each time $t$, the adversary selects $x_t = \hat{x}_t + e_t$, where $\|\hat{x}_t\| \leq 1/2$, and $e_t \sim r\mathcal{B}_1^d$ for $r \leq 1/2$ (and, for simplicity, $d > 1$). Then, the adversary is $\sigma$-smooth for $\sigma = \text{vol}_d(r\mathcal{B}_1^d)/\text{vol}_d(\mathcal{B}_1^d) = r^d$. However, if $u \sim \mu_d$ is drawn uniformly from the sphere, then the density $p_1(\cdot)$ of its first coordinate $u_1$ with respect to the Lebesgue measure is*

$$p_1(u_1) = \frac{\text{vol}_{d-1}(\sqrt{1-u_1^2}\mathcal{B}_1^{d-1})}{\text{vol}_d(\mathcal{B}_1^d)} \leq \frac{\text{vol}_{d-1}(\mathcal{B}_1^{d-1})}{\text{vol}_d(\mathcal{B}_1^d)} = \frac{(d-1)}{2\sqrt{\pi}}.$$

*Hence, by rotational symmetry, we see that for any $w^\star$, the adversary is $(\sigma_{\text{dir}}, \hat{w}^\star)$ directionally-smooth for $\sigma_{\text{dir}} = \frac{2\sqrt{\pi}r}{(d-1)}$. Notice that the directional smoothness is now only polynomial in $d$, rather than exponential in it.*

**Example 2** (Additive Noise in a Random-Direction). *Again consider the additive noise setting where at each time $t$, the adversary selects $x_t = \hat{x}_t + e_t$, where $\|\hat{x}_t\| \leq 1/2$. However, suppose $e_t$ is selected as follows: before the game, the adversary selects a direction $\hat{e} \sim \mu_d$, and plays $e_t = a_t\hat{e}$, where $a_t$ is drawn uniformly on the interval $[-r/2, r/2]$. Note that this adversary need not be $\sigma$-smooth with respect to $\mu_d$ for any $\sigma > 0$, because after the adversary commits to $\hat{e}$, her smoothing is restricted to a line segment. Still, with constant probability, $\langle \hat{e}, \hat{w}^\star \rangle \geq c/d$ for some constant $c > 0$. Hence, with constant probability, the adversary is $(\sigma_{\text{dir}}, \hat{w}^\star)$-directionally smooth for $\sigma_{\text{dir}} = cr/d$.*

We now state our guarantee for the classical Perceptron algorithm Rosenblatt [1958]

**Theorem 53.** *Suppose that the adversary is $(\sigma_{\text{dir}}, \alpha, \hat{w}^\star)$-Tsybakov, and define $\rho := \frac{2}{3-\alpha} \in [\frac{2}{3}, 1)$. Then, with probability $1 - \delta$, the Perceptron algorithm (Algorithm 9) satisfies*

$$\sum_{t=1}^{T} \mathbb{I}\{\hat{y}_t \neq y_t\} \lesssim (T/\sigma_{\text{dir}})^\rho \cdot (N_{\text{err}})^{1-\rho} + \log(\lceil \log T \rceil/\delta).$$

---

**Algorithm 9** Online Perceptron

---

1: **Initialize** $w_1 = \mathbf{e_1} \in \mathcal{B}_1^d$
2: **for** $t = 1, 2, \ldots$ **do**
3:     **Recieve** $x_t$ and **predict**

$$\hat{y}_t = \text{sign}(\langle w_t, x_t \rangle), \qquad\qquad (\text{\% self.classify}(x_t))$$

4:     **if** $\hat{y}_t \neq y_t$ **then**                                        (\text{\% self.errorUpdate}(x_t))
5:         $w_{t+1} \leftarrow w_t + y_t x_t$
6:

---

**Remark 4.** *Recall Example 1, which shows that directional smoothness $\sigma_{\text{dir}}$ may scale as $\sim r/d$ when the (standard) smoothness scales as $\sigma = r^d$. Applying Theorem 53 with $\alpha = 0$ and thus, $\rho = 2/3$, our $(T/\sigma_{\text{dir}})^{2/3} \sim (Td/r)^{2/3}$-mistake bound interpolates between the $\log(T/\sigma) \sim d\log(1/r) + \log(T)$ bounds attained in this paper, and the $\text{poly}(1/\sigma) \sim (1/r)^{\Omega(d)}$-regret enjoyed by previous computationally efficient algorithms. In addition, we achieve a robustness to sublinearly-in-$T$ mistakes, which prior approaches do not.*

In fact, a more general result holds, in terms of a direction-wise anti-concentration of the adversaries distributions.

**Theorem 54** (Guarantee under Tsyabkov Smoothness). *Define the anti-concentration function*

$$\mathsf{p}_\mu(\eta; v) := \sup_t \sup_{a \in \mathbb{R}} \mathbb{P}_{x_t \sim p_t}[\langle x_t, v \rangle \in [a, a + \eta] \mid \mathscr{F}_{t-1}].$$

*For any fixed $\gamma \in (0, R)$, with least $1 - \delta$, the number of mistakes made by the Perceptron (Algorithm 9) is at most*

$$\sum_{t=1}^T \mathbb{I}\{\hat{y}_t \neq y_t\} \lesssim \frac{N_{\text{err}}}{\gamma^2} + T\mathsf{p}_\mu(R\gamma, \hat{w}^\star) + \log(1/\delta),$$

### G.1   Proofs for the Perceptron

We begin by stating the standard guarantee for the Perceptron algorithm due to Freund and Schapire [1999]. To emphasizes its generality, we use $\bar{x}_i$ to denote its inputs, which we allow to have non-normalized radius $R$.

**Theorem 55.** *Let $(\bar{x}_i, y_i)_{i=1}^T \in \mathbb{R}^n \times \mathbb{R}$ be a sequence of labeled exampled with $\|\bar{x}_i\| \leq R$. Fix $\bar{w} \in \mathbb{S}^{n-1}$, $\gamma > 0$, and define the margin errors*

$$d_i := d_i(\bar{w}, \gamma) = \max\{0, \gamma - y_i \cdot \langle \bar{x}_i, \bar{w} \rangle\}$$

*Then, the number of mistakes make by the online Perceptron is at most*

$$\frac{(R+D)^2}{\gamma^2}, \quad D = \sqrt{\sum_{i=1}^T d_i^2}$$

The following corollary explicitly bounds the term $D^2$,

**Corollary 56.** *Fix a $\bar{w} \in \mathbb{S}^{n-1}, \gamma \in (0, R)$. Let*

$$N_1 := |\{i : \text{sign}(y_i \cdot \langle \bar{x}_i, \bar{w} \rangle) < 0\}|$$
$$N_2 := |\{0 \leq y_i \cdot \langle \bar{x}_i, \bar{w} \rangle \in [0, \gamma]\}|$$

*Then, the number of mistakes make by the online Perceptron is at most*

$$(8N_1 + 4)\frac{R^2}{\gamma^2} + 2N_2$$

*Proof of Corollary 56.* Let $S_1 := \{i : \text{sign}(y_i \cdot \langle \bar{x}_i, \bar{w} \rangle) \neq 1\}$ and $S_2 := \{i : \text{sign}(y_i \cdot \langle \bar{x}_i, \bar{w} \rangle) = 1, \quad y_i \cdot \langle \bar{x}_i, \bar{w} \rangle \leq \gamma\}$. Note that $N_1 := |S_1|$ and $N_2 := |S_2|$. Moreover, $S_1 \cap S_2 = \emptyset$, and if

$i \notin (S_1 \cup S_2)$, $d_i := \max\{0, \gamma - y_i \cdot \langle \bar{x}_i, \bar{w} \rangle\} = 0$. Hence,

$$D^2 = \sum_{i \in S_1} d_i^2 + \sum_{j \in S_2} d_j^2$$

$$\leq N_1 \max_{i \in S_1} d_i^2 + N_2 d_j^2.$$

For $i \in S_1$, $d_i^2 \leq (\gamma + |\langle \bar{x}_i, \bar{w} \rangle|)^2 \leq (\gamma + R)^2 \leq 4R^2$, where we used $\gamma \leq R$. For $j \in S_2$, $d_j \in [0, \gamma]$, so $d_j^2 \leq \gamma^2$. Thus, $D^2 \leq 4R^2 N_1 + N_2$. Thus,

$$\frac{(R+D)^2}{\gamma^2} \leq \frac{2R^2 + 2D^2}{\gamma^2} \leq (8N_1 + 4)\frac{R^2}{\gamma^2} + 2N_2.$$

∎

We now return to our specific setting, re-adopting $x_i$ (not $\bar{x}_i$) for features. We bound the probability that a given point $x_i$ does not lie within a margin $\gamma$.

**Lemma 57.** *Consider the $p_\mu$ function from eq. (% self.classify($x_t$)). Then, for any interval $I_0 \subset \mathbb{R}$,*

$$\mathbb{P}[y_t^\star \cdot (b^\star + \langle x_t, w^\star \rangle)) \in I_0 \mid \mathcal{F}_{t-1}] \leq 2p_\mu(2|I_0|, \hat{w}^\star),$$

*In particular, $\mathbb{P}[y_t^\star \cdot (b^\star + \langle x_t, w^\star \rangle) \leq \gamma\} \mid \mathcal{F}_{t-1}] \leq 2p_\mu(2\gamma, \hat{w}^\star)$.*

*Proof of Lemma 57.* Note that the ground truth label $y_i$ may depend on $x_i$. We circumvent this with a union bound. Let $I_0$ be any inverval.

$$\mathbb{P}[y_t^\star \cdot (b^\star + \langle x_t, w^\star \rangle) \in I_0] \leq \sum_{y \in \{-1, +1\}} \mathbb{P}[\langle x_t, w^\star \rangle \in yI_0]$$

$$\leq \sum_{y \in \{-1, +1\}} \mathbb{P}[b^\star + \langle x_t, \hat{w}^\star \rangle \in (b^\star + yI_0)/\|w^\star\|]$$

$$\leq 2p_\mu(\|w^\star\|^{-1}|I_0|, \hat{w}^\star) \leq 2p_\mu(2|I_0|, \hat{w}^\star),$$

where we recall our assumption $\|w^\star\| \geq 1/2$. ∎

We may now prove Proof of Theorem 54.

*Proof of Theorem 54.* We apply Corollary 56 with $\bar{w} = (w^\star, b^\star)$ and $\bar{x}_i = (x_i, 1)$. Note then that $\|\bar{x}_i\|^2 = 1 + \|x_i\|^2 \leq 2$, so we may take $R = \sqrt{2}$. Define

$$N_1 := |\{i : \text{sign}(y_i \cdot \langle \bar{x}_i, \|\bar{w}\| \rangle) < 0\}|$$

$$N_2 := |\{i : \text{sign}(y_i \cdot \langle \bar{x}_i, \bar{w} \rangle) = 1, y_i \cdot \langle \bar{x}_i, \bar{w} \rangle \in [0, \gamma]\}|$$

it suffices to bound $N_1$ and $N_2$. Sice $y_t^\star \cdot \langle \bar{x}_t, \bar{w} \rangle) \geq 0$, we see that each

$$N_2 = \sum_{t=1}^{T} Z_t, \quad Z_t := \mathbb{I}\{y_t \cdot (b^\star + \langle x_t, w^\star \rangle) \in [0, \gamma]\}.$$

Set $t_\gamma := 2p_\mu(2\gamma, \hat{w}^\star) + 8\log(1/\delta)/m$. By Lemma 57,

$$\mathbb{E}[Z_t \mid \mathcal{F}_{t-1}] \leq 2p_\mu(2\gamma, \hat{w}^\star) \leq t_\gamma$$

Hence, by Lemma 16,

$$\mathbb{P}[N_2 \geq 2Tt_\gamma] = \mathbb{P}[\sum_{t=1}^{T} Z_t \geq 2Tt_\gamma] \leq \exp(-Tt_\gamma/8) \leq \delta.$$

Thus, from Corollary 56, appliyed to the vectors $(x_t, 1)$, the number of mistakes is at most

$$(8N_1 + 4)\frac{R^2}{\gamma^2} + 2N_2 \leq \frac{16(N_1 + 1)}{\gamma^2} + 2N_2 \qquad (R^2 = 2)$$

$$\leq \frac{16(N_1 + 1)}{\gamma^2} + 4Tt_\gamma \qquad (\text{w.p. } 1 - \delta)$$

$$= (8N_1 + 4)\frac{R^2}{\gamma^2} + 8Tp_\mu(2\gamma, \hat{w}^\star) + 32\log(1/\delta).$$

∎

*Proof of Theorem 53.* Fix any $N \in \mathbb{N}$. Under the Tsybakov smoothness of the adversary,

$$\frac{N}{\gamma_N^2} + m\mathsf{p}_\mu(2\gamma_N, \hat{w}^\star) \leq \frac{N}{\gamma_N^2} + \sigma_{\text{dir}}^{-1}T(2\gamma_N)^{1-\alpha} \lesssim \frac{N}{\gamma_N^2} + \sigma_{\text{dir}}^{-1}T(\gamma_N)^{1-\alpha} \tag{G.1}$$

Balance both terms by setting $\gamma_N^{3-\alpha} = (N)/(\sigma_{\text{dir}}^{-1}T)$. Then, for $\rho = \frac{2}{3-\alpha}$, this choice of $\gamma$ ensures

$$\frac{N}{\gamma_N^2} = (N)^{1-\rho}(T/\sigma_{\text{dir}})^\rho$$

Since $\gamma_N^{3-\alpha}$ balanced the terms in Equation (G.1), we have

$$\frac{N}{\gamma_N^2} + T\mathsf{p}_\mu(2\gamma_N, \hat{w}^\star) \lesssim (N)^{1-\rho}(T/\sigma_{\text{dir}})^\rho \tag{G.2}$$

For $k \in \mathbb{N}$, let $\mathcal{E}_k := \{2^{k-1} \leq N_{\text{err}} \leq 2^k\}$. Then, $\mathbb{P}[\bigcup_{k=1}^{\lceil \log T \rceil}] = 1$. Moreover, by applying Theorem 54 with $\gamma_{2^k}$ for each $k$, we have with probability $1 - \delta$, if $\mathcal{E}_k$ holds, then applying Equation (G.2) with $N = 2^k$,

$$\#\text{mistakes} \lesssim N_{\text{err}}\frac{1}{\gamma_{2^k}^2} + T\mathsf{p}_\mu(2\gamma_{(2^k)}, \hat{w}^\star) + \log(1/\delta)$$

$$\lesssim (T/\sigma_{\text{dir}})^\rho((2^k))^{1-\rho} \cdot (\kappa^\star)^{\frac{2-2\alpha}{3-\alpha}} + \log(1/\delta)$$

$$\lesssim (T/\sigma_{\text{dir}})^\rho(N_{\text{err}})^{1-\rho} + \log(1/\delta),$$

where in the last line, we use $N_{\text{err}} \geq 2^k/2$ on $\mathcal{E}_k$. Taking a union bound over $k \in [\lceil \log T \rceil]$, with probability $1 - \delta$,

$$\#\text{mistakes} \lesssim (T/\sigma_{\text{dir}})^\rho(N_{\text{err}})^{1-\rho} + \log(\lceil \log T \rceil/\delta).$$

∎

## G.2   Lower bound on $1/\|w^\star\|$

**Lemma 58.** *There exists $(\tilde{w}, \tilde{b})$ for which $\mathbb{P}[y^\star(x_t) = \text{sign}(\tilde{b} + \langle x_t, \tilde{w}\rangle), \forall t \geq 1] = 1$, and which satisfy $|\tilde{b}| + |\tilde{w}|^2 = 1$, and $\|\tilde{w}\| \geq 1/2$.*

*Proof.* We consider two cases.

- Case 1: $y^\star(x)$ is not constant on $\mathcal{B}_1^d$. Let $(\tilde{b}, \tilde{w})$ be equal to $\alpha(b^\star, w^\star)$, where $\alpha$ is chosen so that $\tilde{b}^2 + \|\tilde{w}\|^2 = 1$. By positive homegenity of sign, $y^\star(x) = \text{sign}(\tilde{b} + \langle x_t, \tilde{w}\rangle)$. Since $y^\star(x)$ is not constant on $\mathcal{B}_1^d$, we must have $\|\tilde{w}\| \geq |\tilde{b}|$. This means $\|\tilde{w}\|^2 \geq \tilde{b}^2 = 1 - \|\tilde{w}\|^2$. Hence, $\|\tilde{w}\|^2 \geq 1/2$.

- Case 2: Since $y^\star(x) \equiv y^\star$ is constant on $\mathcal{B}_d^1$. For some $\varepsilon$ small, set $\tilde{b} = \sqrt{1/2 + \varepsilon}y^\star$, and set $\tilde{w} = e_1\sqrt{(1 - \tilde{b}^2)} = \sqrt{(1/2 - \varepsilon)}e_1$, where $e_1$ is the first cannonical basis vector. By construction, $\tilde{b}^2 + \|\tilde{w}\|^2 = 1$, and $y^\star(\tilde{b} + \langle x, \tilde{w}\rangle) \geq \sqrt{1/2 + \varepsilon} - \sqrt{1/2 + \varepsilon} > 0$. To conclude, we take $\varepsilon = 1/4$ (though any $\varepsilon$ arbitrarily close to zero would work as well).

∎