# OpenReview forum: "Efficient and Near-Optimal Smoothed Online Learning for Generalized Linear Functions"
_NeurIPS.cc/2022/Conference — NeurIPS 2022 Accept_

### Official Review · Reviewer_iruc · 2022-07-09

**Rating:** 7
**Confidence:** 3
**Soundness:** 3 good
**Presentation:** 3 good
**Contribution:** 3 good

**Summary:**

This paper studies the smoothed online learning setting, where the adversary's action at each step must be drawn from a distribution with density bounded by $1/\sigma$. This model bridges the offline/batch setting and the usual online setting. Previous upper bounds for this smoothed setting, however, have an exponential gap in the dependence on $T/\sigma$ between computationally bounded and unbounded learners.

This work gives efficient algorithms that achieve $\log(T/\sigma)$-type regret, which nearly matches the optimal regret when computation is unbounded. All the following results hold under the realizable case: For linear classification, Theorem 5 gives a high-probability regret bound of $O(d\log d + \log(T/\sigma))$, nearly matching an $\Omega(d + \log(T/\sigma))$ lower bound in Proposition 6. This is further generalized to affine classification (Corollary 8), linear classification under a feature map (Theorems 9 and 10), and multi-class classification (Theorems 11 and 12).

For linear classification, the algorithm (Algorithm 1) is quite intuitive: maintain the (convex) set of hypotheses that are consistent with all previous examples, and predict with the center of its John ellipsoid. (The authors show that for other natural choices like the Chebyshev center or centroid, the analysis would not work.)

The analysis builds on the following two technical tools:
- A decay analysis (Lemma 3), showing that it suffices to argue that whenever a mistake is made, the probability of making a mistake (at the next step) shrinks by a factor.
- A geometric result (Lemma 4) that proves such a decay. While its statement is intuitive, the proof of Lemma 4 is quite technical and nontrivial.

**Questions:**

1. The paragraph on Lines 248-255 says that other choices of $w_t$ cannot guarantee the decay in both volume and surface area. Are there hard instances on which these choices provably incur a large regret? In other words, to what extent is the surface area decay necessary?

2. What is the main difficulty in extending the results to the agnostic setting?

**Limitations:**

The main limitation is the results only hold for certain hypothesis classes and only under the realizable setting. This is made clear by the authors when summarizing the contribution.

**Strengths And Weaknesses:**

The paper studies a sequential learning setting that receives much attention recently, and makes good progress in closing the computation-information gap in the optimal regret. While the proposed algorithm is simple, its analysis seems highly nontrivial even in the linear classification setting. I found the paper enjoyable to read, and thought that the authors did a good job in including both intuition and technical details of their proofs in the main paper, despite the technical difficulty in them.

One weakness is that the results are specific to the realizable setting, and it is unlikely that the current apporach/techniques could be easily extended to agnostic case. Nevertheless, given that previously no efficient algorithms are known to achieve the $\log(T/\sigma)$-type regret, I found the current results sufficiently significant.

---

> ### Author Response · Authors · 2022-08-01
> **Clarification of point regarding realizability assumption as well as necessity of decay of surface area**
>
> We thank the reviewer for their thoughtful feedback. Regarding the realizability assumption, we agree that this is somewhat limiting and are actively thinking of ways to eliminate this. The main challenge is that if our learning setting is not realizable, we cannot simply eliminate hypotheses that disagree with the data thus far. One way to extend the techniques to the agnostic setting with an L* type bound would be to first assume an upper bound on the number of mistakes allowed to be made and then modify the notion of disagreement set accordingly. Aggregating over the possible number of upper bounds on mistakes would then give an L* bound; the problem with this approach is the computational difficulty, although with a clever aggregation procedure, it may well be possible to achieve desired computational efficiency.
>
> Note that the limitation to certain hypothesis classes is necessary in light of Corollary 12 from Smoothed Online Learning is as Easy as Statistical Learning by Block et al 2022; no proper, computationally efficient algorithm can achieve regret scaling logarithmically in sigma in general, despite this scaling being the correct, information theoretic rate. Our paper shows that there exist some classes where this lower bound can be broken; a more complete characterization of when this can happen is an interesting avenue for future research.
>
> Regarding the first question, on to what extent surface area decay is necessary, we omitted this discussion due to space constraints, but are happy to include it in the camera ready version as this is a great question. In fact, surface area decay is necessary to apply our techniques, which is not at all obvious and is central to our choice of the John ellipsoid. Note that our Lemma 4 bounds the measure of the disagreement region by the sum of two terms: one involving the measure of the version space and one involving the surface area of the version space. Both of these terms are necessary in general. As an example, suppose that we are in the setting of linear classification and the version space is an arc of some great circle in the d-dimensional unit sphere. Then the disagreement region is given by the set of points in the ball that are orthogonal to at least one element of the version space. But this set has positive measure even though the version space, as a lower dimensional set, has measure 0! From an intuitive, geometric perspective, what is happening is that each point in the version space is contributing a (d-1)-dimensional set of points to the disagreement region; thus, even version spaces that are (d-1)-dimensional (and thus have trivial measure) can still contribute to nontrivial disagreement regions.

---

> > ### Comment · Reviewer_iruc · 2022-08-09
> > **Thank you for your reply!**
> >
> > Thank you for the detailed answer! I would encourage the authors to include some discussion on the necessity of the assumptions (in particular, the impossibility result of [Block et al., 2022]) in the camera-ready version. I maintain an overall evaluation of rating 7.

---

### Official Review · Reviewer_pTSs · 2022-07-11

**Rating:** 6
**Confidence:** 3
**Soundness:** 3 good
**Presentation:** 2 fair
**Contribution:** 3 good

**Summary:**

This paper focuses on the statistical analysis of smoothed online learning for generalized linear functions. It provides the statistical \log(T/sigma) regret for K-wise linear classification and extends the results to settings where the true classifier is linear in an over-parameterized polynomial featurization of the contexts, as well as to a realizable piecewise-regression setting assuming access to an appropriate ERM oracle. This work develops a characterization of the geometry of the disagreement region induced by generalized linear classifiers and a general anti-concentration bound for the determinant of certain matrix averages.

**Questions:**

This paper investigates the statistical analysis of smoothed online learning for generalized linear functions. The paper can be improved from multiple points of view.
1)Presentation. The presentation could be improved. The logic of this paper is hard to follow.
2)Quality. The introduction of related work can be further specified, and the comparison work can be further refined and improved. This paper adopts the smoothed online learning setting of Rakhlin et al. [2011], Haghtalab et al. [2021], Block et al. [2022]. It would be better if there is an improvement of the algorithms.
3)Experiments. This paper mainly provides new theoretical analysis on the existing algorithms. It would be better if there are experiments to verify the results of theoretical analysis.
4)Grammar, typos, and writing. It is suggested to further polish the writing.
References are generally not used as the beginning of a sentence, such as: Rakhlin et al. [2011] proposed smoothed adversaries and proved regret bounds for linear thresholds in ...
Tenses in this paper are inconsistent. For example:
In line 73, Smoothed analysis was first proposed in Spielman and Teng [2004] as a way to explain the success of the simplex algorithm...
In line 101, Hanneke [2007] controls the label complexity of active learning...
The equation at the end of the sentence should be punctuated. For example, equation (2.2).

**Limitations:**

This paper describes the scope of application of the proposed methods in detail, and does not involve negative social impact.

**Strengths And Weaknesses:**

This paper has the following contributions: This paper studies the statistical property of certain classes of realizable smoothed online classification problems, for example affine thresholds, affine thresholds in nonlinear features, K-class affine classification, and piecewise affine regression, which have the statistically optimal regret bound. Then, this paper obtains a fast algorithm that achieves optimal regret dependence on the horizon, up to logarithmic factors, by applying the obtained results to noiseless contextual bandits. Finally, a complementary approach based on the perceptron algorithm is provided, which has a polynomial regret in a “directional smoothness” parameter. This paper adopts the smoothed online learning setting of Rakhlin et al. [2011], Haghtalab et al. [2021], Block et al. [2022]. The related work can be elaborated in more detail. It would be better to provide related experimental results to prove the rationality of the theoretical analysis.
The research field of this paper is significant. The organization and presentation of this paper can be further improved. There is no conclusion section.

---

> ### Author Response · Authors · 2022-08-01
> **Response to feedback on improving on pre-existing algorithms**
>
> We wish to thank the reviewer for their careful reading and thoughtful feedback.
>
> Regarding the first question on presentation, our desire was to first discuss our two main technical tools and then show how those tools can be applied in a number of situations to provide efficient, optimal algorithms in the smoothed online learning setting. Due to space constraints, we did not include a conclusion, but we are happy to add one in in the camera ready version.
>
> Regarding the second question on quality, we are happy to expand on our related work section. Note that our paper does in fact provide "an improvement of the algorithms." In earlier work, there was no computationally efficient algorithm that provides optimal information theoretic regret, depending logarithmically on sigma. In fact, there are lower bounds, such as Corollary 12 from Block et al 2022 showing that, in general, there can be no computationally efficient (proper) algorithm achieving this regret. In our paper, we provide a new algorithm for specific function classes demonstrating that this computational lower bound can be broken for certain natural classes, leading to a an exponential improvement on existing algorithms with regret scaling logarithmically in sigma as opposed to polynomially. In the camera ready version, we emphasize this point more strongly.
>
> Regarding experiments, our paper is theoretical, much like many of the cited papers in the related work section. Empirical validation of the approach is an interesting task for future work.
>
> Regarding grammar, thank you for pointing out some of the typos; we will correct these for the camera ready version.

---

### Official Review · Reviewer_XCuH · 2022-07-12

**Rating:** 8
**Confidence:** 2
**Soundness:** 4 excellent
**Presentation:** 4 excellent
**Contribution:** 4 excellent

**Summary:**

This paper considers the problem of smooth online learning in the realizable setting. In the smooth online learning problem, the adversary is limited to choosing a distribution over contexts at each round which is “smooth” wrt to a dominating measure. The player knows the dominating measure.

For smooth online learning algorithms, there exists a statistical computational gap, and the main goal of this paper is to close the gap for the special case of the linear function classes in high dimensions. The main contribution of the paper is designing novel algorithms for this problem which attain the optimal statistical guarantees on the regret while computationally-efficient.

The main idea behind the proof is presented in Section 3. The authors show that if, with respect to the dominating measure, the error probability of the player decays, then with respect to any other measure that is smooth wrt to the dominating measure , we have a control over the regret. Then, as usual, to control the probability of error, the authors use the fact that a control over the size of the disagreement region can control the probability of error. They have a very nice result which provides a bound on the size of the disagreement region based on the volume of the version space in Lemma 4.

In the rest of the paper, they use their techniques to obtain optimal regret bounds for Linear Classification, Perceptron algorithm, Linear Classification Under a Feature map, and Multi-Class Classification.


**Questions:**

1- What is the definition of overly strong measures in Remark 2?

2- Are your results robust to the change of the dominating measure? How can we design algorithms that are agnostic to the dominating measure?

3- It seems that the algorithms designed in the paper are agnostic to the value of sigma. Is there any improvement if we assume the player knows the value of sigma?

5- As mentioned in some cases, the sigma may depend on the dimension. Do we have a general approach to make sure sigma is O(1) in dimension?

6- Your results strongly rely on the fact that the linear function class has a natural parameterization as vectors in dimension d. How can the results be extended to the classes without natural parametrization.



**Limitations:**

yes.

**Strengths And Weaknesses:**

Overall, I think this paper is really strong. Also, the paper is very well-written, and the proof sketches are helpful in understanding the gist of the techniques. Moreover, the techniques used by the authors are also very interesting and novel.
My main concern about this paper is the “robustness” of the results and the techniques to the dominating measures. It seems in most of the results in the paper, it is assumed that the dominating measure is uniform on the unit ball of radius one in dimension d.

I have a few questions which I will discuss in the next part.

---

> ### Author Response · Authors · 2022-08-01
> **Answering questions on the dominating measure assumptions**
>
> We wish to thank the reviewer for their careful reading and thoughtful feedback.
>
> Regarding the reviewer's first question on the definition of an overly strong measure in Remark 2, we note that what we claim not to be overly strong is the assumption that our adversary is smooth with respect to the Lebesgue measure. In the camera ready version, we will clarify that sentence. To expand on that point, we note that the logarithmic dependence on sigma means that we also inherit smoothness guarantees for any measure which is sigma' smooth with respect to the Lebesgue measure on the ball, at the expense of an additional log(1/sigma') factor. For example, if the adversary is sigma-smooth with respect to the uniform measure on any convex body with volume exp(-O(poly(d))), we suffer at most an additional poly(d) regret.
>
> On the reviewer's second question, regarding robustness to changing the dominating measure, we note that algorithms that can achieve regret logarithmic in sigma in general must have some control over the dominating measure, as per Proposition 4 from the paper Smoothed Online Learning is as Easy as Statistical Learning by Block et al 2022. A better understanding of when a learner can adapt to an unknown dominating measure is an interesting future direction for research.
>
> On the third question, regarding whether knowledge of sigma can improve regret, note that Proposition 6 from our paper, providing a lower bound on the regret, still holds even when sigma is known. We will better clarify this in the camera ready version.
>
> Regarding when sigma can be taken independent of the dimension, we note that the parameter sigma is inherent to the problem. What we mean by sigma being dependent on the dimension is that one natural way to motivate smoothed online learning is to suppose that the adversary chooses a worst case point and then this is perturbed by some small amount of uniform noise; in this case, sigma is dimension dependent. We will better clarify this in the camera ready version.
>
> As noted, our results do strongly rely on the natural parameterization of the function class. Understanding how to extend our results to more general classes is a very interesting topic for further research. Note that Corollary 12 from Smoothed Online Learning is as Easy as Statistical Learning by Block et al 2022 tells us that our approach cannot work for general function classes as we cannot hope to get an efficient algorithm with regret scaling logarithmically in sigma in general. One of our paper's key points is the observation that this lower bound can be broken by specific function classes, with a more general characterization of when logarithmic regret is possible hopefully to follow.

---

> > ### Comment · Reviewer_XCuH · 2022-08-07
> > **Thanks**
> >
> > I would like to thanks the authors for their detailed response. It would be great to include these directions as open problems in the camera ready version of your paper.

---

### Official Review · Reviewer_5Ree · 2022-07-15

**Rating:** 8
**Confidence:** 3
**Soundness:** 3 good
**Presentation:** 4 excellent
**Contribution:** 4 excellent

**Summary:**

The authors study the problem of the gap that exists between the bounds for computationally feasible bounds for online learning poly(T/sigma) and exponential-complexity algorithms with a log(T/sigma). The consider a subset of this problem where the adversary or nature is allowed to select contexts with density bounded w.r.t. a known measure mu. They also provide theorems that characterize the determinant of matrix averages.

**Questions:**

None

**Strengths And Weaknesses:**

The paper gives a computationally efficient algorithm that closes the computational gap for a more restricted problem in online learning: where nature is restricted to select from a distribution bounded by 1/sigma from a know measure mu. This is a novel result that is interesting.

The bigger strengths of the paper are the interesting use of the John ellipsoid in their base algorithm and their use of Ville's inequality to analyze and bound the stopping time. Their theoretical results and techniques are of independent interest in addition to the their computationally efficient algorithm.

---

> ### Author Response · Authors · 2022-08-01
> **Thank you for the feedback.**
>
> We wish to thank the reviewer for their careful reading and thoughtful feedback.

---

> > ### Comment · Reviewer_5Ree · 2022-08-10
> > **Post-rebuttal comments**
> >
> > Just confirming my original review.

---

### Meta-Review · Area_Chair_e9uN · 2022-08-25

**Recommendation:** Accept
**Confidence:** Certain

**Metareview:**

The reviewers agree that this is a solid contribution. Please do revise the paper according to the reviewers comments and the discussion.

**Award:**

No

---

### Decision · Program_Chairs · 2022-09-14

Accept